# On the Variability of Concept Activation Vectors

**Julia Wenkmann** [1]   **Damien Garreau** [1]

## Abstract

One of the most pressing challenges in artificial intelligence is to make models more transparent to their users. Recently, explainable artificial intelligence has come up with numerous methods to tackle this challenge. A promising avenue is to use concept-based explanations, that is, high-level concepts instead of plain feature importance scores. Among this class of methods, Concept Activation Vectors (CAVs, Kim et al., 2018) stand out as one of the main protagonists. One interesting aspect of CAVs is that their computation requires sampling random examples from the train set. Therefore, the actual vectors obtained may vary depending on the randomness of this sampling. In this paper, we propose a fine-grained theoretical analysis of CAV construction in order to quantify their variability. Our results, confirmed by experiments on several real-life datasets of four different modalities, point to an universal result: the variance of CAVs declines roughly as $1/N$, where $N$ is the number of random examples. Based on this, we give practical recommendations for a resource-efficient application of the method.

## 1. Introduction

As deep learning systems are increasingly deployed in high-stakes domains, from medical diagnosis to autonomous vehicles, the inability to understand why a model makes its predictions poses serious risks. A classifier might achieve high accuracy while relying on spurious correlations, or fail catastrophically on edge cases. Without interpretable explanations, practitioners cannot verify that models are learning the right patterns, regulators cannot assess safety, and users cannot calibrate their trust appropriately. Concept-based explanations offer a promising path forward by moving beyond

[1]University of Würzburg, Center for Artificial Intelligence and Data Science (CAIDAS), Würzburg, Germany. Correspondence to: Julia Wenkmann <julia.wenkmann@uni-wuerzburg.de>.

*Proceedings of the 43rd International Conference on Machine Learning*, Seoul, South Korea. PMLR 306, 2026. Copyright 2026 by the author(s).

low-level feature attributions to human-understandable abstractions (Poeta et al., 2025). Among these methods, Testing with Concept Activation Vectors (TCAV), introduced by Kim et al. (2018) has become a foundational framework that underpins much of the subsequent work in concept-based explainability. TCAV quantifies the influence of human-understandable concepts (*e.g.*, "stripes") on a specific class prediction (*e.g.*, "zebra"). Central to the method are CAVs, which associate to a concept's direction in a latent layer by training a linear classifier to distinguish concept examples from random ones. The resulting TCAV score measures the sensitivity of a prediction to this concept direction, as described in detail in Section 3.

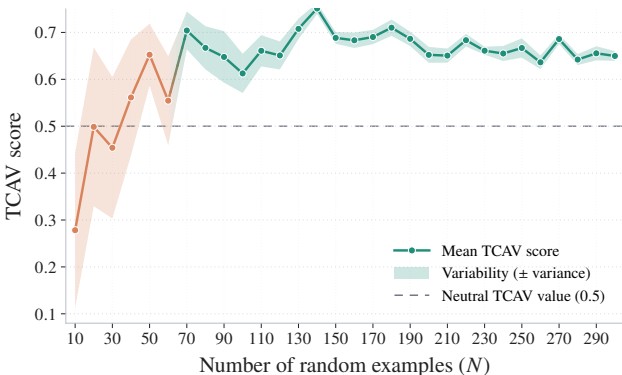

*Figure 1.* Example visualization of the TCAV score as a function of the number of random examples $N$ on the tabular dataset described in Section 5, computed from the `hidden_layers.0` layer for the `sex=Female` concept. The line shows the mean TCAV score across independently sampled random sets, and the shaded ribbon shows empirical run-to-run variability. The dashed line marks the neutral TCAV value of 0.5, which indicates no consistent directional influence of the concept and is the value expected by chance from a random direction.

However, TCAV's reliability is compromised by an under-examined source of instability: the method requires randomly sampling reference examples to train the linear classifier that defines each CAV. Different random samples yield different CAVs, and consequently different TCAV scores even when analyzing the same model with the same concept. This variability is not merely a theoretical concern. Sensitivity to data choices is a recognized issue for CAVs (Ramaswamy et al., 2023; Soni et al., 2020). In real-world settings, practitioners may obtain a TCAV score of 0.7 in

one run and $0.5$ in another, potentially leading to contradictory conclusions about whether a concept positively or negatively influences a class (see Figure 1). Kim et al. (2018) acknowledged this issue and recommended running the method multiple times and reporting average scores. Yet this pragmatic fix raises immediate questions: *How many runs are enough? How many random samples per run?* Without theoretical grounding, practitioners must resort to *ad-hoc* choices that may be either wasteful or insufficient.

In this work, we conduct the first theoretical analysis of CAV variability. Based on this, we derive recommendations for a resource-efficient application of the method. Our contributions are as follows:

1. **Asymptotic normality of CAVs.** We prove that penalized logistic regression CAVs are asymptotically normal in the infinitely imbalanced regime (Thm. 4.1), with covariance trace decaying as $\mathcal{O}(1/N)$. We establish analogous results for hinge loss (Thm. 2), Difference of Means (Thm. 3), and SVMs (Thm. 4).

2. **Variance transfer to downstream quantities.** We show that variance of the derived measure of concept sensitivity inherits the same $\mathcal{O}(1/N)$ scaling, but TCAV scores may retain $\Theta(1)$ variance due to points with zero concept sensitivity (Section 4.4).

3. **Multi-run averaging analysis.** We analyze the variance reduction from averaging over $s$ independent runs, establishing $\mathcal{O}(1/s)$ scaling.

4. **Practical guidelines.** Based on our theoretical and empirical findings across four data modalities (images, tabular, text, and multimodal), we provide concrete recommendations for resource-efficient, stable TCAV.

The code for all experiments is publicly available at https://github.com/juliawenkmann/variability-of-tcav.

## 2. Related Work

As mentioned earlier, TCAV is a very influential method among concept-based explainability approaches and has been extended and modified several times (Poeta et al., 2025). We therefore focus on TCAV, as it serves as a natural starting point for studying stability in this family of methods.

**Influence of the (T)CAV method.** Many subsequent works use both CAVs and TCAV scores to represent concepts in latent space and analyze class-concept relationships. For example, STCE (Ji et al., 2023) transfers the TCAV method to video data and thus allows a temporal view of concepts. Other methods, such as ACE (Ghorbani et al., 2019), ICE (Zhang et al., 2021) and COCOX (Akula et al., 2020),

adapt the calculation of the TCAV score, but in an unsupervised setting. Although TCAV provides mainly global class-concept relationships, CAVLI (Shukla et al., 2023) and Visual-TCAV (De Santis et al., 2024) adapts TCAV to produce local explanations. Specifically, CAVLI combines TCAV with LIME (Ribeiro et al., 2016) for instance-level interpretations, and Visual-TCAV adds saliency maps to localize concepts in the input, leveraging an Integrated Gradients (Sundararajan et al., 2017) approach. Beyond interpretability, CAVs have also been utilized for model steering, where concept vectors are added to or subtracted from internal activations during inference to shift model output toward desired behaviors. This has been applied to guide text generation (Subramani et al., 2022) and more broadly to control model traits like honesty or sentiment via representation engineering (Zou et al., 2023).

Subsequent work has also focused on improving the accuracy of CAVs and accommodating non-linearly separable concepts, which can not be fully captured by CAVs. Methods such as *Concept Activation Regions* (CARs) (Crabbé & van der Schaar, 2022) and *Concept Gradient* (CG) (Bai et al., 2023) generalize CAVs to capture more complex concept boundaries, representing concepts through kernel-based regions or non-linear functions. Soni et al. (2020) improve CAV robustness in two ways. First, *Adversarial* CAV introduces small adversarial perturbations to concept samples. This leads to more stable concept vectors. Second, *Orthogonal Adversarial* CAV applies a Gram–Schmidt-like orthogonalization to further separate concept and non-concept subspaces, thus improving CAV separability. Pattern-based CAV*s*, introduced by Pahde et al. (2025), address the issue of noise when learning CAVs. Instead of learning to separate positive and negative examples, they find a direction in activation space that best correlates with a concept's intensity. This yields concept representations that are more precise and robust against noise.

**Prior Analysis of Robustness and Consistency in XAI.** An important part of evaluating XAI methods is examining their consistency, *i.e.*, the extent to which an explanation method provides deterministic explanations for the same inputs to be explained (Alvarez-Melis & Jaakkola, 2018). Such an analysis has already been carried out for some established XAI methods, such as LIME. Garreau & Mardaoui (2021) show that the randomness and instability observed in LIME explanations are a direct result of not using enough samples to fit the linear model. When the number of generated samples is very large, the explanations converge to an explicit "limit explanation." Visani et al. (2022) take a different approach by introducing two new indices to measure the instability and reliability of LIME, which give practitioners a tool to assess the trustworthiness of LIME's outputs.

**Prior Analysis of CAVs.** Although CAVs are widely used, their fundamental properties and limitations have been comparatively underexplored, with only a few works systematically addressing this gap (Nicolson et al., 2025). At the representation level, Nicolson et al. (2025) show that CAVs suffer from three distinct issues: they are *inconsistent across layers* (the same concept yields different directions depending on the layer probed), *entangled* with semantically related concepts, and *spatially dependent*, meaning that the direction learned for a concept can vary with where in the image the concept appears. At the data level, Ramaswamy et al. (2023) demonstrate that the choice of probe dataset has a significant impact on the resulting explanations: CAVs learned for the same concept on different probe sets can have low cosine similarity, calling into question how far an explanation generalises beyond the dataset it was estimated on. Raman et al. (2024) further highlight a dependence on the spatial location of the concept in the input. Together, these works establish that CAVs are sensitive to layer choice, probe dataset, and spatial structure.

We focus on a different issue: the variance introduced by randomly sampling reference examples.

## 3. Preliminaries and Notation

In this section, we describe the concept activation vectors and TCAV values used by Kim et al. (2018), as well as generalizations introduced by Schmalwasser et al. (2025) and Crabbé & van der Schaar (2022). Hereby we introduce our notation.

### 3.1. Generating Concept Activation Vectors

To formalize concept-based analysis, consider a model *e.g.* a neural network $f: \mathcal{X} \to \mathbb{R}^K$, that classifies inputs into one of $K$ classes. At a chosen layer $\ell$, we decomposed $f$ as $f := g_\ell \circ h_\ell$. Here $h_\ell: \mathcal{X} \to \mathbb{R}^d$ maps inputs to the latent representation at layer $\ell$, and $g_\ell: \mathbb{R}^d \to \mathbb{R}^K$ transforms these embeddings into class logits. In other words, for a fixed class $k \in \{1, \ldots, K\}$, the function $g_{\ell,k}: \mathbb{R}^d \to \mathbb{R}$ maps a latent embedding $v \in \mathbb{R}^d$ to the logit (or score) of class $k$. This is illustrated in Figure 2. The CAV in layer $\ell$ is computed by collecting:

1. *Positive examples* of concept $C$, *i.e.*, a set of $n$ inputs $\{x_i\}_{i=1}^n \subset \mathcal{X}$ that clearly exhibit the concept $C$.

2. *Random examples*, *i.e.*, a set of $N$ inputs $\{z_j\}_{j=1}^N \subset \mathcal{X}$ selected uniformly from the training set.

*Remark* 1. Positive examples can be sourced from annotated datasets such as the Broden dataset (Bau et al., 2017), or be custom-curated, as done e.g. by Kohlikyan et al. (2020). While Kim et al. (2018) also propose using examples from contrary concepts as references, we use randomly

selected samples throughout this work, as they are often more accessible.

We then compute the latent embeddings of the concepts and the random examples. Because the presence of a concept $C$ is generally not directly readable from raw embeddings, TCAV trains a linear classifier to discriminate between positive $\{h_\ell(x_i)\}_{i=1}^n$ and negative $\{h_\ell(z_j)\}_{j=1}^N$ embeddings. With a slight abuse of notation, we use $\{x_i\}_{i=1}^n$ and $\{z_j\}_{j=1}^N$ to refer to the latent representations $\{h_\ell(x_i)\}_{i=1}^n$ and $\{h_\ell(z_j)\}_{j=1}^N$ throughout the paper. The normal vector to the decision boundary learned by the classifier, oriented towards the concept examples, is called the **Concept Activation Vector** $v_C^\ell$ associated to concept $C$. Note that the CAV depends on both the random sampling and the stochastic nature of the classifier training process. Several different classifiers can be employed:

- The original TCAV implementation defaults to a linear Support Vector Machine (`SGDClassifier` with hinge loss (Pedregosa et al., 2011)), as detailed in Appendix B.2.

- A simpler approach, `Difference of Means`, proposed by Martin (2019), is adopted in Visual-TCAV (De Santis et al., 2024). It computes the CAV as the difference between the mean embeddings of the concept and random samples. De Santis et al. (2024) report that this estimator yields better CAV quality. Moreover, this method also yields an average speed-up of $46.4\times$ according to Schmalwasser et al. (2025). We discuss this case in Appendix B.3.

- Crabbé & van der Schaar (2022) propose an alternative that does not require concepts to be linearly separable. They use `Kernel SVMs` to learn Concept Activation *Regions* rather than Concept Activation *Vectors*. We assess the variability of this method using the same metric, extended to the induced Hilbert spaces. See Appendix B.4.

Despite the differences between these methods of computing CAVs *i.e.* CARs, they all exhibit the same asymptotic normality behavior, as we demonstrate experimentally in Appendix A and theoretically in Appendix B. However, in the main part of this paper, we focus on `LogisticRegression` with binary cross-entropy loss as it presents a more amenable mathematical analysis and is the prominent classifier available in the official TCAV repository (Kokhlikyan et al., 2020; Kim et al., 2025).

### 3.2. Calculating TCAV Scores

Once the CAV $v_C^\ell$ has been determined for the concept $C$ in the layer $\ell$, TCAV then evaluates the relevance of this concept for a class $k$ by comparing the gradient of $g_{\ell,k}$ with

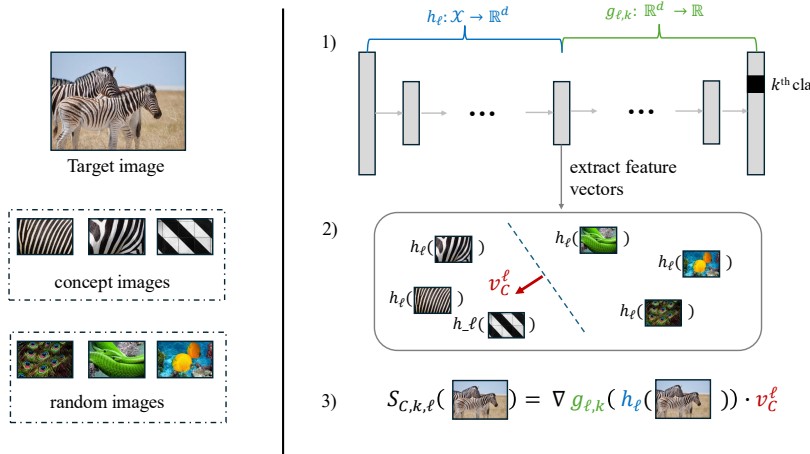

*Figure 2.* An overview of the first part of the TCAV operating procedure: 1) TCAV extracts the activation vectors at a specific layer $\ell$. 2) The CAV $v_C^\ell$, shown by the red arrow, is learned by training a binary linear classifier to differentiate between concept examples and random examples. 3) TCAV calculates the directional derivative $S_{C,k,\ell}(x)$ to measure how sensitive the model's class prediction is to the Concept Activation Vector $v_C^\ell$.

the respective CAV $v_C^\ell$ using the dot product (denoted by $\cdot$). Specifically, for an input $\mathbf{x} \in \mathcal{X}_k$, that is, an input $\mathbf{x}$ of class $k$, TCAV defines the *concept sensitivity score* of $\mathbf{x}$ as

$$S_{C,k,\ell}(\mathbf{x}) := \nabla g_{\ell,k}\big(h_\ell(\mathbf{x})\big) \cdot v_C^\ell. \tag{1}$$

The score $S_{C,k,\ell}(\mathbf{x})$ measures a concept's influence on the classification of $\mathbf{x}$. A positive score indicates that the concept positively influences the prediction toward class $k$, while a negative score indicates that the concept opposes this prediction. To aggregate these local sensitivities, the TCAV *score* is defined as the proportion of samples $\mathbf{x}$ from the class $k$ whose concept sensitivity $S_{C,k,\ell}(\mathbf{x})$ is positive, that is,

$$\text{TCAV}_{C,k,\ell} := \frac{\big|\{\mathbf{x} \in \mathcal{X}_k \mid S_{C,k,\ell}(\mathbf{x}) > 0\}\big|}{|\mathcal{X}_k|}. \tag{2}$$

A score close to 1 means concept $C$ shifts the classification toward class $k$, a low score indicates a strong negative effect, while a score near 0.5 implies no consistent effect.

Thus, Kim et al. (2018) suggest to run the TCAV algorithm multiple times with $s$ different random sets and then compute the mean value of the obtained TCAV scores in their official implementation. Precisely, the multi-run approach partitions a set of $R$ random samples into $s$ disjoint subsets, each of size $N = R/s$. For each subset, a separate logistic regression is trained to find a CAV $v_C^{(j)}$, with $j \in [1, \ldots, s]$. This yields $s$ individual TCAV scores

$$T_j := \text{TCAV}\big(v_C^{(j)}\big) = \frac{\big|\{\mathbf{x} \in \mathcal{X}_k \mid S(\mathbf{x}, v_C^{(j)}) > 0\}\big|}{|\mathcal{X}_k|}.$$

The final **multi-run TCAV score**, $T_{\text{multi}}$, is the average of these individual scores $T_{\text{multi}} := \frac{1}{s} \sum_{j=1}^{s} T_j$. Finally, a

two-tailed $t$-test (see, *e.g.*, (Hogg et al., 2019), Chapter 4.5) is performed to ensure that the effect captured by the TCAV score is not due to random variation.

## 4. Theoretical Analysis

We now present our main theoretical results regarding the variability of CAVs obtained by logistic regression when the number of random examples goes to infinity. This reflects a common practical scenario: annotated concept examples are costly to obtain, a typical concept class in the widely used Broden dataset has on average 50 examples (Bau et al., 2017), whereas random samples can be drawn freely from large unlabelled datasets such as ImageNet (Deng et al., 2009). Understanding the asymptotic behaviour in this regime allows practitioners to answer the key question: *how many random samples are sufficient* to achieve stable CAVs or TCAV scores?

### 4.1. Theoretical Setting

To formalize the algorithmic computation of CAVs described just above, we consider two sets of samples:

- $n$ **fixed points**, $\{x_i\}_{i=1}^n$, representing our "concept" samples and are assigned the class label $Y = 1$.

- $N$ **random points**, $\{z_j\}_{j=1}^N$, drawn from the distribution $F_0$ of training samples and assigned $Y = 0$.

We denote the *intercept* by $\alpha \in \mathbb{R}$ and the *coefficients* by $\beta \in \mathbb{R}^d$. Following Owen (2007), for any input $w \in \mathbb{R}^d$, we write the logistic regression model used in TCAV as

$$\mathbb{P}(Y = 1 | X = w) = \sigma(\alpha + \beta^\top w), \tag{3}$$

where $\sigma(u) := (1 + e^{-u})^{-1}$, with $u \in \mathbb{R}$. For any given $N$, the default TCAV implementation fits the logistic regression model Eq. (3) by maximizing the $L^2$-penalized log-likelihood $\mathcal{L}_N^{(\lambda)}(\alpha, \beta)$, where $\alpha \in \mathbb{R}$, $\beta \in \mathbb{R}^d$ and $\lambda$ is a positive regularization parameter. We denote by $(\alpha_N, \beta_N)$ the unique maximizer of this function. The coefficient vector $\beta_N$ is our mathematical modelisation of the empirical CAV $v_C^\ell$ computed with $N$ random samples. We adopt this notation to distinguish the theoretical quantity under analysis from the algorithmic CAV introduced in Section 3.

## 4.2. Variability of Concept Activation Vectors

In this section, we prove asymptotic normality of CAVs in the special case of infinite imbalance. This specific setting, where the number of samples in one class grows infinitely large while the other is held fixed, was previously analyzed for logistic regression by Owen (2007) and Goldman & Zhang (2022). Our result, similar to that of Owen (2007), is conditional upon the "Surrounded Mean" Assumption. This is a weak condition, postulating that infinite random samples surround the concept's mean $\bar{x}$ in the latent space, which generally holds in practice (see Appendix B.1).

**Assumption 1 (Surrounded Mean).** The distribution $F_0$ on $\mathbb{R}^d$ has the point $\bar{x}$ *surrounded*, that is

$$\int_{(z-\bar{x})^\top \omega > \varepsilon} \mathrm{d}F_0(z) > \delta \qquad (4)$$

holds for some $\varepsilon > 0$, some $\delta > 0$ and all $\omega \in \Omega$ where $\Omega = \{\omega \in \mathbb{R}^d | \omega^\top \omega = 1\}$ is the unit sphere in $\mathbb{R}^d$ and $\bar{x}$ is the mean of the concept embeddings $\{x_i\}_{i=1}^n$.

Intuitively, this condition requires that infinite random samples surround the concept's mean $\bar{x}$ in the latent space: in every direction, the random references must populate a non-trivial cap around $\bar{x}$. This mild geometric assumption generally holds in practice (see Appendix B.1). The second assumption concerns the behavior of the intercept as the class imbalance grows.

**Assumption 2 (Intercept Scaling Limit).** The limit $A_\infty = \lim_{N \to \infty} N e^{\alpha_N}$ exists.

As $N \to \infty$ with $n$ fixed, the intercept $\alpha_N \to -\infty$ to compensate for the growing class imbalance. This assumption requires that $\alpha_N$ diverges at a controlled rate, specifically $\alpha_N \approx -\log N + \log A_\infty$. This mild technical condition holds whenever the regression problem remains well-posed in the limit, and is illustrated in Appendix B.1. The accompanying notebooks provide empirical checks.

The third assumption ensures that the CAV itself stabilizes.

**Assumption 3 (Consistency of $\beta_N$).** There exists a deterministic vector $\beta_\infty \in \mathbb{R}^d$ such that $\beta_N \xrightarrow{p} \beta_\infty$ as $N \to \infty$.

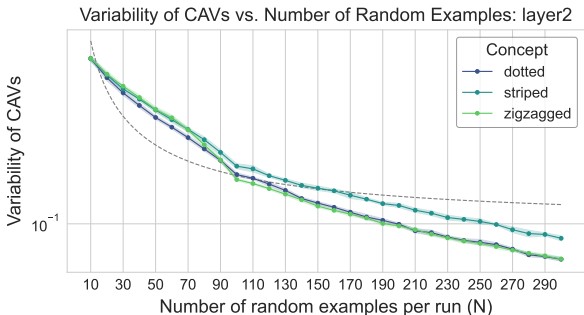

*Figure 3.* Mean variability of CAVs for `ResNet-50` at `'layer2'` as a function of the number of random examples per run ($N$), shown for "striped"-, "zigzagged"-, and "dotted" concepts. Error bars indicate $\pm 1$ standard deviation; the $y$-axis is log-scaled. Variance of CAVs is estimated by taking the sum of per-feature variances across ten independent runs. The dashed line shows a fit of the form $y = a/N + b$ estimated on the average of all concepts.

This is the central assumption of our analysis: as we increase the number of random samples, the estimated CAV converges to a well-defined limiting direction $\beta_\infty \in \mathbb{R}^d$. Note, that this assumption can be relaxed to almost sure convergence under stronger moment conditions on $F_0$. Our last assumption is a technical moment condition on $F_0$.

**Assumption 4 (Exponential moments under $F_0$).** For every $\beta \in \mathbb{R}^d$, $\mathbb{E}_{F_0}\left[\|Z\|^2 \exp(2\beta^\top Z)\right] < \infty$.

This assumption ensures that the random samples $Z \sim F_0$ have sufficiently light tails. The condition is satisfied by any distribution with bounded support, as well as by Gaussian. In practice, neural network activations are typically bounded, so this assumption holds for most architectures encountered in deep learning.

We provide empirical verification of these assumptions in the accompanying notebooks under "Checking Assumptions." We can now state our main result:

**Theorem 4.1 (Asymptotic Normality of CAV s).** *Assume that Assumption 1, 2, 3, and 4 hold. Then*

$$\sqrt{N}\left(\beta_N - \beta_\infty\right) \xrightarrow{D} \mathcal{N}(0, \Sigma), \qquad (5)$$

*where $\xrightarrow{D}$ denotes convergence in distribution. A precise expression for $\Sigma$ is given in Eq. (20).*

Informally, Theorem 4.1 states that, for large $N$, the random fluctuations of $\sqrt{N}(\beta_N - \beta_\infty)$ are well-approximated by a centered multivariate normal distribution with covariance matrix $\Sigma$. This matches the usual $\mathcal{O}(1/N)$ variance scaling from classical asymptotic theory: the CAV estimator $\beta_N$ is closely related to a logistic regression estimator (Goldman & Zhang, 2022), and such estimators typically have covariance that decreases proportionally to $1/N$. We show that this also holds in the infinitely imbalanced setting.

Let us now return to our main topic: the variability of the CAVs. For this we choose the trace of the covariance matrix of the CAVs, *i.e.*, the sum of the per-feature variance, as a variability measure. This measure equals zero, *if and only if* the CAVs have no variability at all.

The following definition is standard in multivariate statistics; we restate it here for completeness and to fix notation.

**Definition 1 (Variability of CAVs).** We define the variability of $\beta_N$ as the trace of the covariance matrix

$$\text{Var}(\beta_N) := \sum_j \text{Cov}(\beta_N)_{jj}. \quad (6)$$

We are now able to examine the variability of CAVs .

**Corollary 1 (Variability decay of CAVs).** *Under the assumptions of Theorem 4.1, the estimator $\beta_N$ satisfies*

$$\text{Cov}(\beta_N) = \mathcal{O}(1/N) \qquad \text{as } N \to \infty.$$

*In particular, each component has $\text{Var}((\beta_N)_j) = \mathcal{O}(1/N)$.*

See Appendix B.1, Corollary 3 for a precise statement and proof. In words: as the random dataset size ($N$) grows, the variability in the direction of the Concept Activation Vector ($\beta_N$) shrinks at a rate proportional to $1/N$. For instance, multiplying the number of random examples $N$ by ten reduces the variance by a factor of ten. Figure 3 confirms this scaling empirically; additional results across datasets and models appear in Section 5.

### 4.3. Variance of Concept Sensitivity Scores

We now analyze whether the asymptotic normality of CAVs transfers to the concept sensitivity scores. In the notation of this section, they are written

$$S(\mathbf{x}, \beta_N) := \nabla g_{\ell,k}(h_\ell(\mathbf{x})) \cdot \beta_N. \quad (7)$$

The following corollary shows that we can transfer the asymptotic convergence of the CAVs. The variance of concept sensitivity decreases at the rate $\mathcal{O}(1/N)$, like the variance of $\beta_N$, as demonstrated in Figure 4.

**Corollary 2 (Asymptotic Normality of Concept Sensitivity).** *Under the conditions specified in Theorem 4.1, the concept sensitivity score $S(\mathbf{x}, \beta_N)$ satisfies*

$$Z_N := \sqrt{N}(S(\mathbf{x}, \beta_N) - S(\mathbf{x}, \beta_\infty)) \xrightarrow{D} \mathcal{N}(0, V(\mathbf{x})), \quad (8)$$

*where the asymptotic variance $V(\mathbf{x})$ is given by*

$$V(\mathbf{x}) := \nabla g_{\ell,k}(h_\ell(\mathbf{x}))^\top \cdot \Sigma \cdot \nabla g_{\ell,k}(h_\ell(\mathbf{x})). \quad (9)$$

This corollary follows directly from Theorem 4.1: since $S(\mathbf{x}, \beta_N)$ is a linear function of $\beta_N$, the asymptotic normality transfers immediately. The $\mathcal{O}(1/N)$ variance decay is confirmed empirically in Figure 4 (see also Appendix B.1.4 for the full proof).

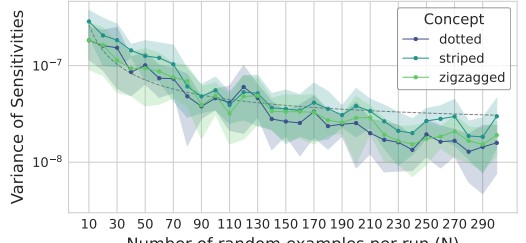

*Figure 4.* Mean variance of concept sensitivity scores *vs.* number $N$ of random samples used for the class "zebra" in the Imagenet classification setting for one input. The $y$-value shows the geometric mean variance of these concept sensitivity scores $\pm 1$ standard deviation, averaged over all $r = 10$ runs for a fixed positive input of a zebra. Log scale cut at $\geq 3 \times 10^{-9}$ so near-zero bands do not dominate the picture.

### 4.4. Variance of TCAV Scores

Extending our variance analysis to the TCAV scores in Eq. (2), we make a surprising observation. In practice, the variance does not always decrease with more random samples, as one might assume. Experimentally, we observe that the variance of the TCAV scores initially decreases and then plateaus in many settings for medium to high values of $N$ of random embedding vectors used (see Figure 5).

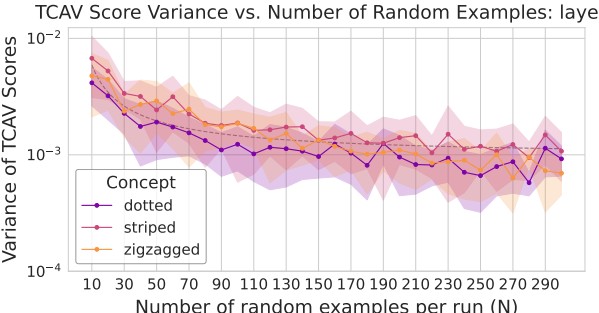

*Figure 5.* Mean variance of TCAV scores at layer `layer2` vs. the number of random examples per concept set $N$ for "striped"-, "zigzagged"-, and "dotted" concepts on the Imagenet dataset; error bars denote $\pm 1$ standard deviation. $Y$-axis clipped at $\geq 10^{-4}$ to reduce distortion from near-zero standard deviations.

Our intuition for this quite surprising fact is the following: across the embeddings of samples from a specific class over which we compute the TCAV score, a subset of them may lie **on** or **near** the decision boundary. This makes their classification highly sensitive to small changes in the model. Those samples, which we call "borderline points," therefore still contribute to the variance of the TCAV score. This holds even when the variance of the CAVs vanishes asymptotically. For any other "non-borderline" point, the classification is asymptotically stable, and its contribution to the variance is negligible. The covariance between any two borderline points, on the other hand, is a constant value,

$\mathcal{O}(1)$, that does not decrease as the CAV estimate improves. Therefore, the total variance is dominated by the sum over the pairs of "borderline points" and $\mathrm{Var}(\mathrm{TCAV}) = \mathcal{O}(1)$. More examples of this behavior are provided in Appendix A. Based on our empirical and theoretical observations we can now give concrete recommendations for action.

### 4.5. Variability of the Multi-Run Method

Based on our observations, averaging TCAV scores over multiple runs, as done in the original implementation, is indeed the most favourable method. Our analysis shows that as $s$ increases, the variance decreases as $\mathcal{O}(1/s)$ (see Figure 6). This aligns with theoretical expectations: since the individual scores $T_1, \ldots, T_s$ are independent, averaging $s$ i.i.d. estimates with common variance $\sigma^2$ yields variance $\sigma^2/s$ (Hogg et al., 2019). Hence, assuming $\mathrm{Var}(\mathrm{TCAV}) = \mathcal{O}(1)$, the variance of the multi-run score satisfies $\mathrm{Var}(T_{\mathrm{multi}}) = \mathcal{O}(1/s)$.

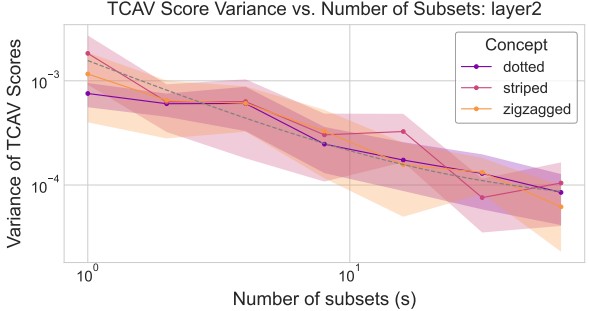

*Figure 6.* Variance of multi-run TCAV scores in the image setting. We use a fixed number $R = 2000$ of random samples and divide them over a varying number of subsets $s$, with each subset containing $N = R/s$ samples. To get the variance of the mean over all $s$ TCAV scores, we repeat this $r = 10$ times and calculate the variance for each $s$ over the $r$ runs. The plot shows that the variance of the mean TCAV score decreases as $s$ increases, confirming our theoretical analysis. Finally, we repeat this experiment $e = 10$ times and report the mean variance over those $e$ runs $\pm 1$ geometric standard deviation.

### 4.6. Recommendations for Practice

In Section 1 we ask: *"How can the variance of CAVs be minimized within a limited resource budget?"* Our theoretical and empirical analysis provides concrete guidance.

**Compute-variability trade-off.** Empirically, we find that compute cost scales linearly with $N$ for all CAV estimators examined (logistic regression, hinge loss SVM, and Difference of Means) and quadratically for CAR estimators (see Figure 17a in the Appendix).

By Corollary 1, the variance of CAVs decreases as $\mathcal{O}(1/N)$. However, the constant factors differ substantially across estimators. In the example of tabular data (see Section 5), we summarize the resulting trade-off between stability and com-

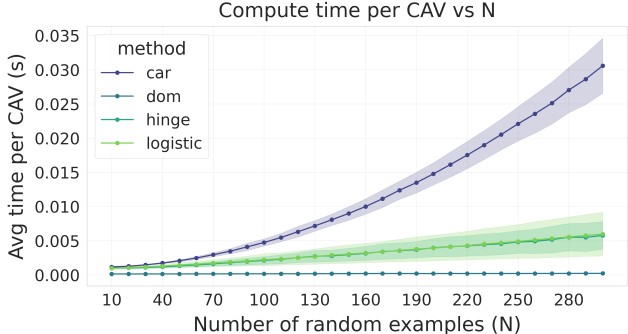

*Figure 7.* Average wall-clock time per concept representation as a function of the number of random reference examples per run ($N$). We compare logistic-regression CAVs (LOGISTIC), hinge-loss CAVs (HINGE), Difference-of-Means CAVs (DOM), and Concept Activation Regions (CAR). Shaded bands denote $\pm 1$ standard deviation across repeated runs.

putational effort in Figure 8. Each point represents a different sample size $N$, showing average compute time per CAV versus CAV variability. The `Difference-of-Means` estimator clearly achieves the lowest variability for any given compute budget. This aligns with Theorem 3, which shows `Difference-of-Means` exhibits the same $\mathcal{O}(1/N)$ convergence rate while requiring only a single mean computation rather than iterative optimization.

We can now present our practical recommendations given a fixed compute budget.

**Recommendation 1: For stable CAVs, use `Difference-of-Means` and maximize N.** According to Schmalwasser et al. (2025), this estimator is on average about $\sim 46 \times$ faster than logistic regression. We show that it also yields comparable or better stability for a given compute budget. Independent of the method, we recommend selecting $N$ based on your target stability level: by Equation (33), $\mathrm{Var}(\beta_N) = \mathrm{tr}(\Sigma_z)/N$, so **doubling N halves the variance**.

**Recommendation 2: For stable TCAV scores, use multiple runs with modest N.** Our surprising finding (Section 4.4, Figure 5) is that TCAV score variance does not always decrease with $N$. Instead, averaging over $s$ independent runs reduces variance as $\mathcal{O}(1/s)$. Given a fixed budget of $R$ total random samples, allocating them as $s$ runs of $N = R/s$ samples each is more effective for TCAV stability than a single run with all $R$ samples. Note that this increases compute time: doubling $s$ approximately doubles the number of classifier trainings (with a factor of $\approx 1.5 \times$ in our experiments due to overhead).

In conclusion, the optimal strategy depends on the use:

- **For CAV direction stability** (*e.g.*, bias mitigation, feature steering): maximize $N$ per run with respect to your

compute budget, use `Difference-of-Means`.
- **For TCAV score stability** (*e.g.*, concept importance rankings): use multiple modest-$N$ runs and average the scores.

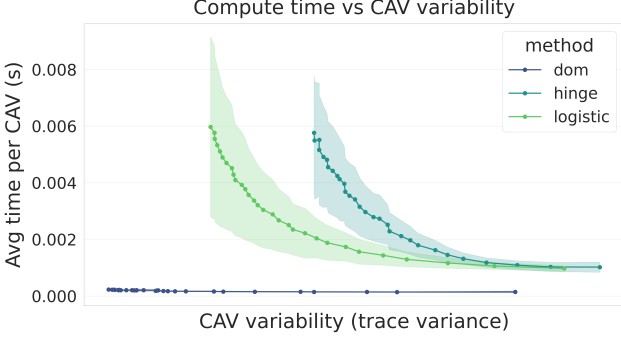

*Figure 8.* Compute-variability trade-off for CAV estimators on tabular data. We show average compute time per CAV vs. the CAV variability across estimators. Each point corresponds to a different random-sample size $N$ averaged over all layers.

## 5. Experiments

We validate our theoretical findings and assumptions for the four estimation methods using four data modalities: images, tables, text, and multimodal data. For each, we use appropriate datasets and concepts (some from existing benchmarks, others newly created). We verify all theorems and corollaries and empirically substantiate our practical recommendations. We summarize the main findings below; Appendix A contains the complete results for all estimators and modalities, concept-set-size variation in Appendix A.5, and implementation ablations in Appendix A.6.

**Experimental protocol.** For a given run, we vary the number of random examples, $N$ from 10 to 300. At each value of $N$, we compute $s = 50$ separate CAVs. Each of these CAVs is trained on a set of $N$ examples, which are sampled with replacement from a large pool of 1,000 (for images) to 50,000 (for text) samples. We then calculate the variability of the resulting 50 CAVs. We repeat this experiment $r = 10$ times for statistical significance and report the mean and standard deviation of the traces collected across all $r$ runs.

**Image Data.** Following the original TCAV setup, we use ImageNet (Deng et al., 2009) with concept definitions from Broden (Bau et al., 2017). Our main results use a pre-trained ResNet-50 (He et al., 2016) (`layer2` and `layer3`), but we also run experiments on GoogLeNet (Szegedy et al., 2015), MobileNetV3 (Howard et al., 2019), and ViT-B/16 (Dosovitskiy et al., 2021). All show comparable variance scaling. We focus on ResNet-50 for clarity; full results appear in Appendix A.1.

**Tabular Data.** While image classification is the typical use case for concept-based explanations, our theoretical results

apply to any differentiable model. To verify this generality, we adapt the framework to income prediction on the *UCI Adult* dataset (Becker & Kohavi, 1996). Unlike vision tasks where concepts require external datasets, tabular data often contains interpretable attributes that can serve directly as concept definitions. We leverage this by defining two concepts, "male" and "female," from the `sex` attribute. The model is a small network trained to predict whether income exceeds $50,000. Results are provided in Appendix A.2.

**Text Data.** We apply CAVs in the NLP setting using the IMDB sentiment dataset (Maas et al., 2011). We consider two text model families. First, we use the pre-trained BERT-based sentiment classifier from Captum (Kokhlikyan et al., 2020) and extract representations from layers `bert_encoder_2` and `bert_encoder_4`. Concepts are defined by hand-picked sets of positive, negative, and neutral adjectives, extending the Captum notebooks. Second, we evaluate `SmolLM2-1.7B-Instruct` (Allal et al., 2025) and extract CAVs from decoder layers `decoder_hidden_6` and `decoder_hidden_12`. For both text model families, we report results for binary cross-entropy, hinge loss, Concept Activation Regions, and Difference of Means in Appendix A.3.

**Multimodal Data.** To evaluate our framework on vision-language models, we use CLIP (Radford et al., 2021) and LLaVA-1.5 (Liu et al., 2024). For CLIP, we pair zebra images with the prompt "a photo of a zebra" and use image-text cosine similarity as the scalar output instead of a target class logit. Concepts are defined by image sets, as in the image experiments. We extract CAVs from CLIP vision layers `vision_hidden_4` and `vision_hidden_8`. For LLaVA-1.5, we use the same visual concept protocol and extract CAVs from vision layers `vision_hidden_7` and `vision_hidden_15`. For both vision-language model families, we report results for binary cross-entropy, hinge loss, Concept Activation Regions, and Difference of Means in Appendix A.4.

**Results.** Across all modalities, model families, and estimators, CAV variability decays approximately as $1/N$, consistent with our theoretical predictions. Implementation ablations further show that layer choice, feature normalization, regularization, initialization, and optimizer choice mainly affect finite-sample constants rather than the central stabilization trend. The TCAV score tells a different story. In most settings, TCAV score variance remains approximately constant even as the underlying CAV variability decreases with $N$. We attribute this apparent paradox largely to the borderline points discussed in Section 4.4: the TCAV score depends on the *sign* of concept sensitivities, and points near the decision boundary can flip classification regardless of how precisely the CAV direction is estimated.

## 6. Limitations

Our primary focus is on stability. Our analysis does not determine which method performs best on downstream tasks, since such comparisons are difficult without available ground-truth. Nevertheless, we find that the resulting CAVs are very similar across methods and converge toward comparable estimates. This is consistent with prior work demonstrating that different CAV computation methods yield similar results under common conditions (Soudry et al., 2018; Schmalwasser et al., 2025; Schnoor et al., 2026).

Moreover, the ablations show that the constants in the variance decay can depend on the representation layer and implementation choices. Thus, the qualitative recommendation to increase the number of random reference examples is robust, but numerical budgets should be recalibrated when moving to a new architecture, layer, or TCAV implementation.

## 7. Conclusion

In this paper, we analyze the stability of the TCAV method. To this end, we introduce a mathematical framework to theoretically analyze variability in the limit of infinitely imbalanced logistic regression, which applies beyond the scope of this work. Building on this analysis and extensive experiments, we provide practical recommendations on how many random samples to use and how to allocate them for optimal stability in TCAV. We further present a thorough exploration of the compute-consistency trade-off, quantifying how a fixed budget is best split between (i) more independent runs with fewer random samples per run and (ii) fewer runs with larger per-run sample sizes. *Crucially, this trade-off is method- and implementation-dependent*: it differs for TCAV scores versus CAVs and varies with the logistic-regression solver, regularization, feature normalization, and stopping criteria. We demonstrate that, for stable TCAV scores, a modest number of random samples distributed across multiple independent runs is sufficient and compute-efficient. By contrast, applications that require stable CAVs, such as bias mitigation or feature steering, benefit more from larger per-run sample sizes. We emphasize again that there is no single best setting: the optimal compute allocation should be re-evaluated for the specific method and implementation at hand.

Building on our findings, we identify the development of more stable TCAV variants as an important direction for future work. Rather than only quantifying variability or mitigating it through repeated runs and larger reference sets, future methods could incorporate stability directly into the construction of concept directions and scores. Moreover, further work should investigate implementation-specific compute-variability trade-offs, since our results show that stability depends not only on the number of random samples, but also on choices such as the estimator, solver, regularization, feature normalization, and stopping criteria.

## Impact Statement

Reliable concept-based explanations are only useful if they are reproducible. Yet, in practice TCAV can shift across runs because the concept direction is learned from random reference samples, leaving teams guessing whether a reported effect is real or just sampling noise. Our results turn "how many random examples do I need?" into concrete, budgetable rules. We hope this strengthens TCAV and related methods by improving reproducibility, enabling fairer comparisons across studies, and reducing wasted compute by investing resources where they actually buy stability.

## Acknowledgements

We thank Gabriele Ciravegna for his valuable feedback. This work has been supported by a student assistant position at the Group of Theory of Machine Learning, University of Würzburg. Prototyping and parts of the development were carried out on the Julia 2 cluster. Julia 2 was funded as DFG project as "Forschungsgroßgerät nach Art 91b GG" under INST 93/1145-1 FUGG.

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

# Appendix

**Table of Contents**

## A. Appendix A: Additional Results

In Appendix A, we provide the full experimental evidence behind Section 5. We report results across the four modalities studied in the paper and across the four CAV estimation methods: binary cross-entropy, hinge loss, Concept Activation Regions, and Difference of Means. For each setting, we report the relevant variability quantities: CAV/CAR variability, concept-sensitivity variance where available, and TCAV/TCAR score variance. We also include additional SmolLM2 and LLaVA experiments, concept-set-size variation, and implementation ablations over layer choice, feature normalization, regularization, initialization, and optimizer choice.

**Layer selection.** Unless otherwise stated, we report two representative intermediate layers per architecture, one earlier/middle layer and one later layer. This keeps the main comparison readable while still testing layer dependence; a dedicated layer ablation is provided in Appendix A.6.

**Note.** All experiments were conducted on a MacBook Pro running macOS Sonoma 14.6. The system was equipped with an Apple $M3$ Max chip (14-core CPU) and 36 GB of unified memory.

### A.1. TCAV for Images

Finally, we evaluate TCAV on images. We report results for ResNet50 (He et al., 2016), but the method is readily applicable to a wide range of architectures; we have also implemented it for GoogLeNet (Szegedy et al., 2015), EfficientNet (Tan & Le, 2019), MobileNetV3 (Howard et al., 2019), and ViT-B/16 (Dosovitskiy et al., 2021). Across models, the empirical behavior aligns with our propositions and theorems, though to varying degrees. Some settings converge more slowly, which is expected given the asymptotic nature of our results.

#### A.1.1. EMPIRICAL FINDINGS WITH BINARY CROSS-ENTROPY LOSS

Finally, we present the results for image data using the logistic regression classifier with binary cross-entropy loss. When analysing the variance with logistic regression classifiers, we observe a somewhat unexpected behaviour: initially, the variance increases until it asymptotically decreases after about 200 images. This is not a contradiction to our theoretical statements, as the latter only applies asymptotically. However, the exact reasons for this behaviour are worth investigating.

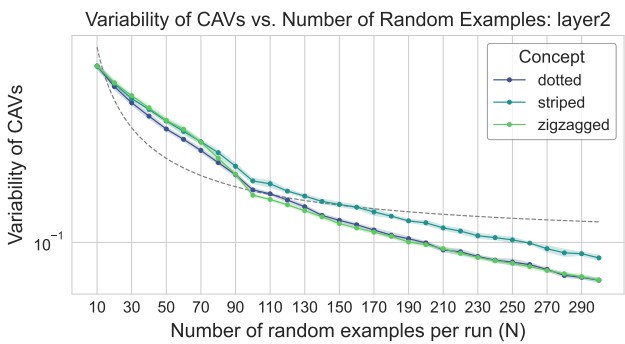
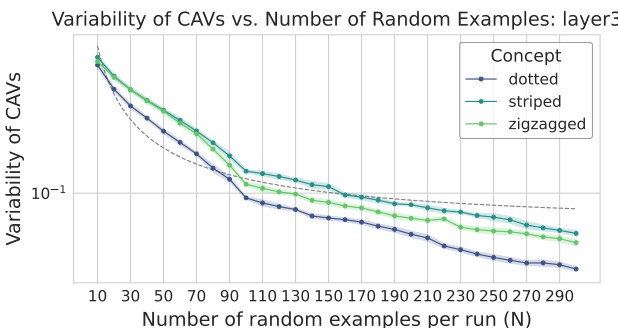

*Figure 9.* Mean variability of CAVs for "striped", "zigzagged," and "dotted" as a function of random examples ($N$) for 'layer2' and 'layer3' of the **Resnet50 model**. The classifiers were trained using **binary cross-entropy loss**. Error bars indicate $\pm 1$ SD; the $y$-axis is log-scaled. We fitted a curve of the form $f(N) = a/N + b$ to it. For layer2 the parameters were $a = 7.71, b = 0.0999$, for layer3 they were $a = 5.58, b = 0.0643$.

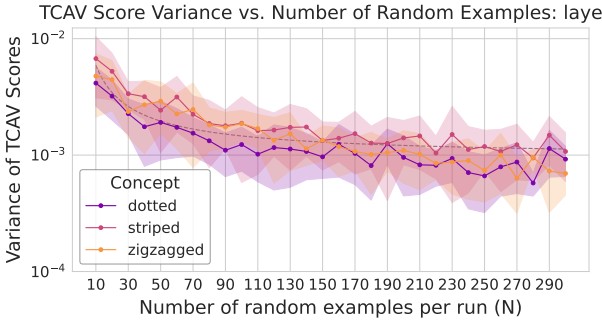
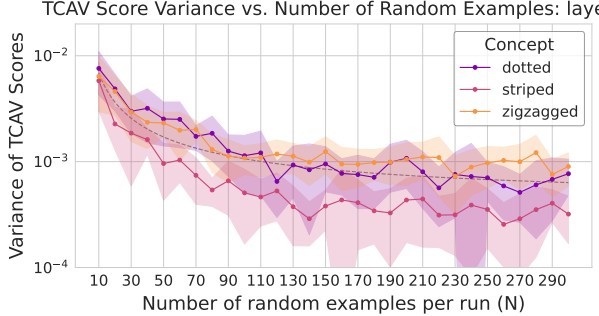

*Figure 10.* Variance of TCAV scores at 'layer2' and 'layer3' of the **Resnet50 model** vs. the number of examples per concept set ($N$) for the visual concepts "striped", "zigzagged", and "dotted". Error bars denote $\pm 1$ standard deviation. We fitted a curve of the form $f(N) = a/N + b$ to it. For layer2 the parameters were $a = 0.0396, b = 0.00134$, for layer3 they were $a = 0.0784, b = 5.83 \times 10^{-4}$. For layer2 the $y$-axis was clipped at $\geq 10^{-4}$.

### A.1.2. EMPIRICAL FINDINGS WITH HINGE LOSS

Next, we show the corresponding results with hinge loss.

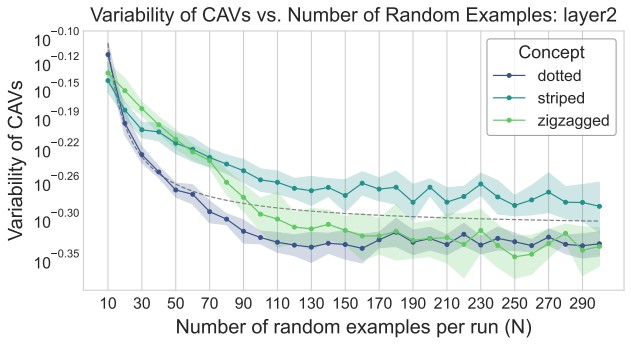
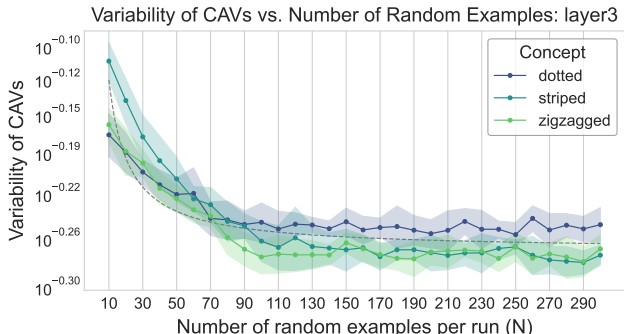

*Figure 11.* Mean variability of CAVs for the visual concepts "striped", "zigzagged", and "dotted" as a function of the number of random examples per run ($N$). Results are shown for two different layers of the **Resnet50 model** ('layer2' on the left, 'layer3' on the right). Error bars indicate $\pm 1$ SD; the $y$-axis is log-scaled. Variance is estimated by the sum of per-feature variances across runs. We fitted a curve of the form $f(N) = a/N + b$ to it. For layer2 the parameters were $a = 3.01, b = 0.488$, for layer3 $a = 2.12, b = 0.538$.

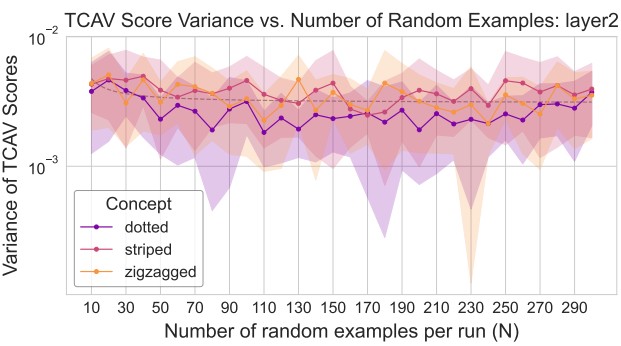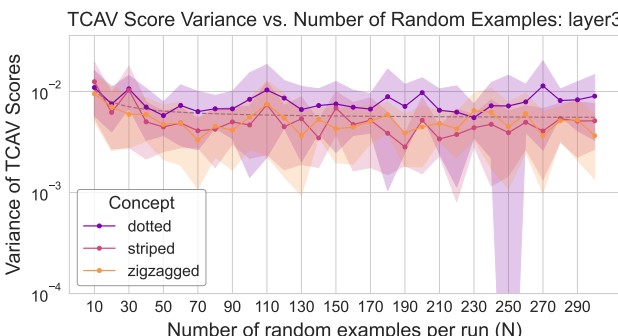

*Figure 12.* Variance of TCAV scores at 'layer2' and 'layer3' of the **ResNet50 model** vs. the number of examples per concept set ($N$) for the visual concepts "striped", "zigzagged", and "dotted". The underlying CAVs were trained using **hinge loss**. Error bars denote $\pm 1$ standard deviation. We fitted a curve of the form $f(N) = a/N + b$ to it. For layer2 the parameters were $a = 0.0168, b = 0.00307$, for layer3 they were $a = 0.0507, b = 0.00539$. For layer3 the scale is trimmed at $\geq 10^{-4}$ to prevent small-variance tails collapsing the plot. $Y$-axis clipped at $\geq 10^{-6}$ for 'layer2' and $\geq 10^{-4}$ for layer3 to reduce distortion from near-zero standard deviations.

### A.1.3. EMPIRICAL FINDINGS FOR CONCEPT ACTIVATION REGIONS

Second, we establish that the same theoretical guarantees apply to the Concept Activation Region framework (Crabbé & van der Schaar, 2022).

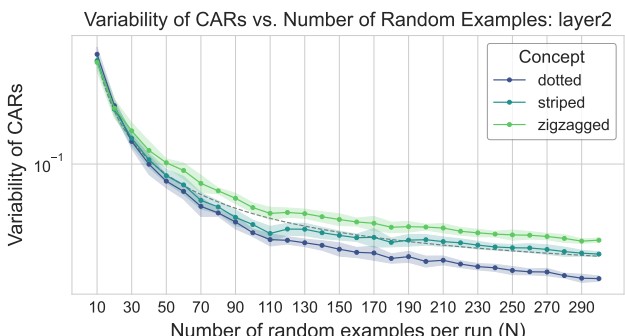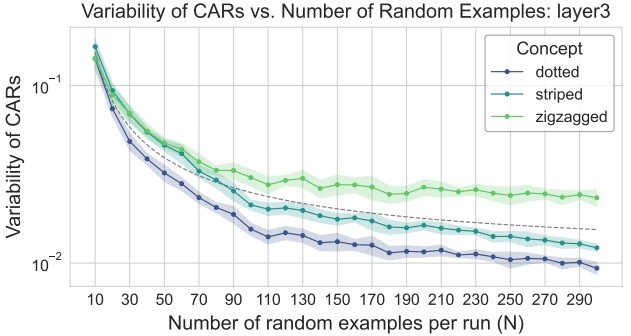

*Figure 13.* Mean variability of CARs for the visual concepts "striped", "zigzagged", and "dotted" as a function of the number of random examples per run ($N$). Results are shown for two different layers of the **Resnet50 model** ('layer2' on the left, 'layer3' on the right). Error bars indicate $\pm 1$ SD; the $y$-axis is log-scaled. Variance is estimated by the sum of per-feature variances across five independent runs. We fitted a curve of the form $f(N) = a/N + b$ to it. For layer2 the parameters were $a = 3.3, b = 0.0225$, for layer3 they were $a = 1.43, b = 0.0107$.

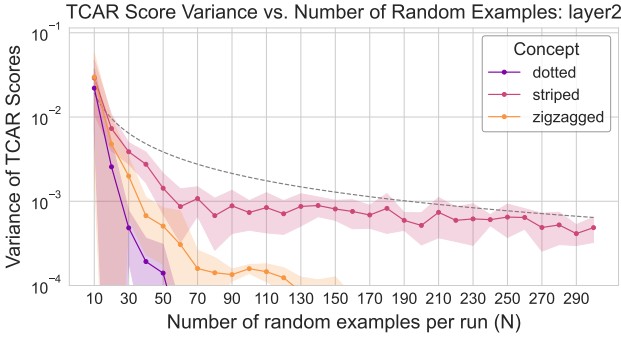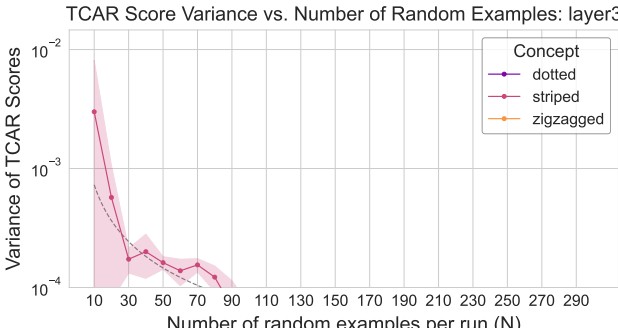

*Figure 14.* Variance of TCAR scores at 'layer2' and 'layer3' of the **ResNet50 model** vs. the number of examples per concept set ($N$) for the visual concepts "striped", "zigzagged", and "dotted". The underlying CARs were trained using non-linear SVM. Error bars denote $\pm 1$ standard deviation. We fitted a curve of the form $f(N) = a/N + b$ to it. For layer2 the parameters were $a = 0.193, b = 4.55 \times 10^{-11}$, for layer3 they were $a = 0.0507, b = 0.00539$.

### A.1.4. EMPIRICAL FINDINGS WITH DIFFERENCE OF MEANS

Again, we found that CAVs computed via the *Difference of Means*-method most closely follow a variance decline of $\mathcal{O}(1/N)$, compared to the other two methods.

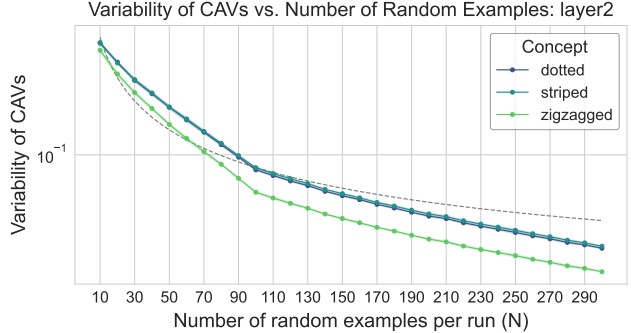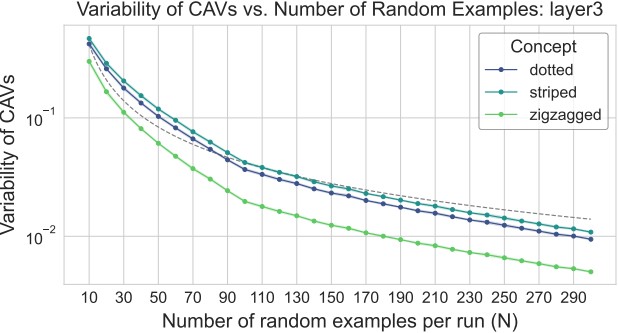

*Figure 15.* Mean variability of CAVs in dependance of the number of random examples per run $N$ for **ResNet50 model**. The CAVs were generated using **difference-of-means**. Error bars indicate $\pm 1$ SD; the $y$-axis is log-scaled. Variance is estimated by the sum of per-feature variances across five independent runs. We fitted a curve of the form $f(N) = a/N + b$ to it. For `layer2` the parameters were $a = 7.24, b = 0.0106$, for `layer3` they were $a = 4.13, b = 3.18 \times 10^{-8}$.

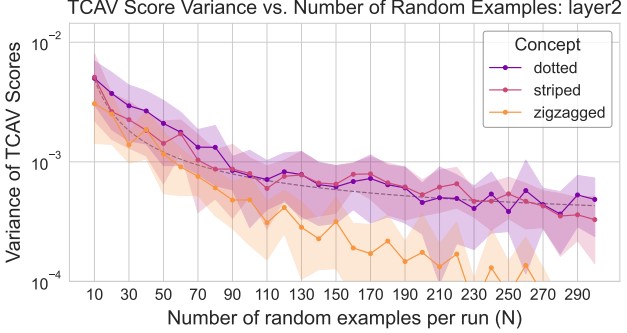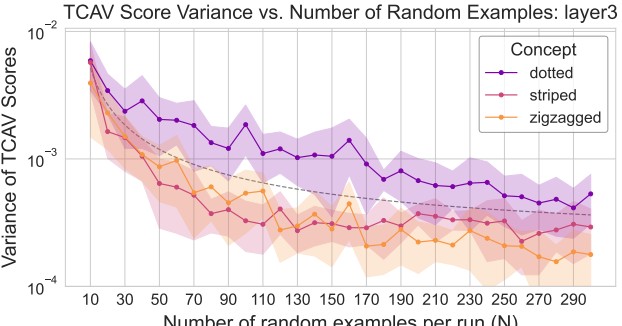

*Figure 16.* Variance of TCAV scores at `'layer2'` and `'layer3'` of the **ReseNet50 model** vs. the number $N$ of random examples used. The underlying CAVs were generated using **difference-of-means**. Error bars denote $\pm 1$ standard deviation. We fitted a curve of the form $f(N) = a/N + b$ to it. For `layer2` the parameters were $a = 0.0429, b = 3.29 \times 10^{-4}$, for `layer3` they were $a = 0.0548, b = 1.5 \times 10^{-4}$.

### A.1.5. COMPUTE TRADE-OFF

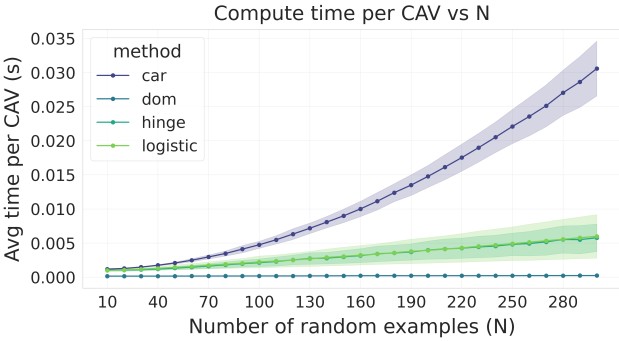 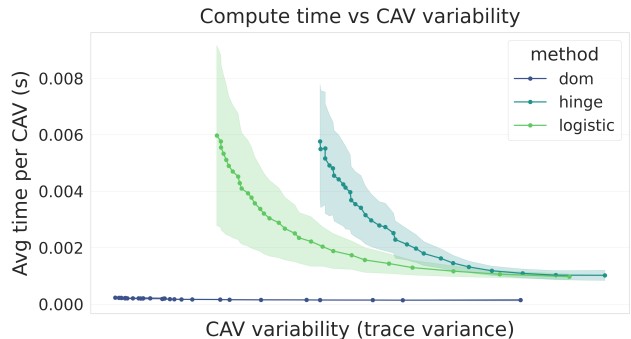

*(a)* **Compute vs.** $N$**.** Average wall-clock time per CAV as a function of the number of random examples per run ($N$). Shaded band denotes $\pm 1$ SD across runs.

*(b)* **Compute vs. variability.** Compute–stability trade-off: average time per CAV versus CAV variability (trace variance). Points correspond to different $N$ values (connected in increasing $N$). Shaded band denotes $\pm 1$ SD for compute time.

*Figure 17.* Compute cost and stability of CAV generation (see legend for the selected setting, e.g. layer/method). Left: compute time per CAV vs. $N$. Right: compute time vs. resulting CAV variability (trace variance), illustrating the compute–stability trade-off.

## A.2. TCAV for Tabular Data

Because this tabular task is straightforward, we train several simple models from scratch. Using the Adult dataset(Becker & Kohavi, 1996), we define two concepts from the gender field—male and female.

### A.2.1. EMPIRICAL FINDINGS WITH BINARY CROSS-ENTROPY LOSS

First, we analyze the binary cross-entropy loss from the logistic regression classifier.

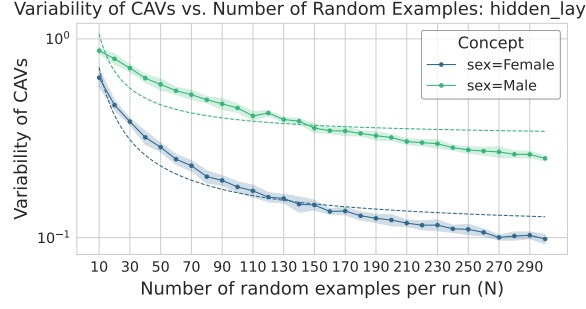 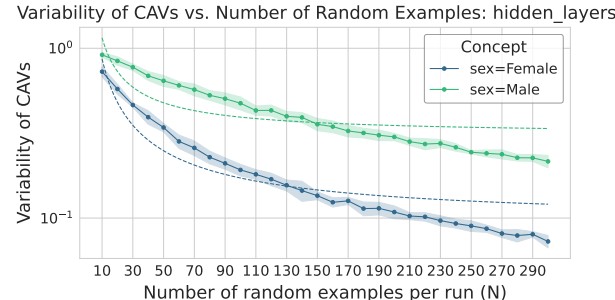

*Figure 18.* Mean variability of CAVs as a function of the number of random examples per run ($N$) on the **UCI Adult dataset**. Results are shown for two different hidden layers. Error bars indicate $\pm 1$ SD; the $y$-axis is log-scaled. Variance is estimated by the sum of per-feature variances across ten independent runs. We fitted a curve of the form $f(N) = a/N + b$ to it. For `hidden_layers_0` they were $4.75/N + 2.18 \times 10^{-13}$ for "sex=Female" and $6.74/N + 0.281$ for "sex=Male". For `hidden_layers_2` they were $6.82/N + 0.074$ *i.e.* $8.46/N + 0.228$.

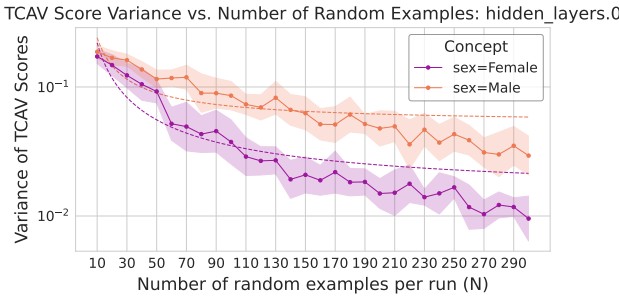 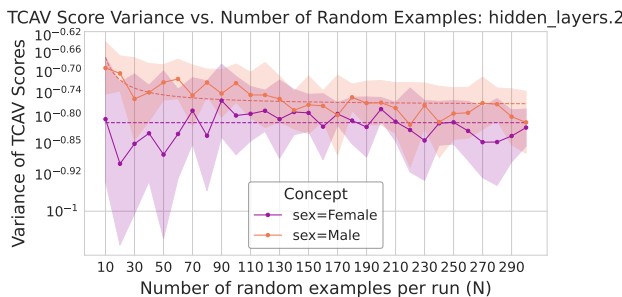

*Figure 19.* Variance of TCAV scores vs. the number of examples per concept set ($N$) for the concepts "male" and "female" on the **UCI Adult dataset**. Error bars denote $\pm 1$ standard deviation. We fitted a curve of the form $f(N) = a/N + b$ to it. For `hidden_layers_0` they were $2.2/N + 0.00348$ for "sex=Female" and $6.74/N + 0.281$ for "sex=Male". For `hidden_layers_2` they were $6.82/N + 0.074$ *i.e.* $8.46/N + 0.228$.

### A.2.2. EMPIRICAL FINDINGS WITH HINGE LOSS

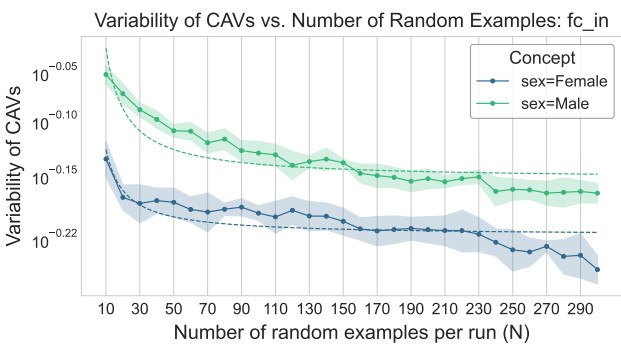 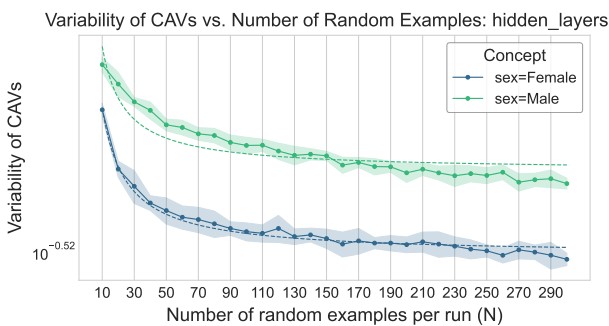

*Figure 20.* Mean variability of CAVs as a function of the number of random examples per run ($N$) on the **UCI Adult dataset**. Results are shown for two different hidden layers. The linear classifiers for CAV generation were trained using **hinge loss**. Variance is estimated by the sum of per-feature variances across ten independent runs. We fitted a curve of the form $f(N) = a/N + b$ to it. For `fc_in` the parameters were $1.41/N + 0.606$ for "sex=Female", and $2.61/N + 0.695$ for "sex=Male". For `hidden_layers_0` they were $3.7/N + 0.296$ *i.e.* $4.63/N + 0.47$.

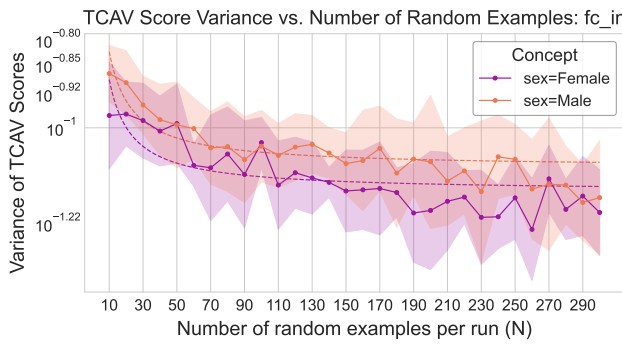 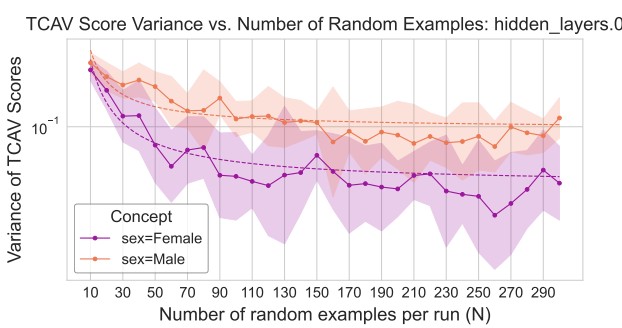

*Figure 21.* Variance of TCAV scores vs. the number of examples per concept set ($N$) for the concepts "male" and "female" on the **UCI Adult dataset**. The underlying CAVs were trained using **hinge loss**. Error bars denote $\pm 1$ standard deviation. We fitted a curve of the form $f(N) = a/N + b$ to it. For `fc_in` the parameters were $a = 0.581, b = 0.0712$ for "sex=Female" and $a = 0.693, b = 0.0808$ for "sex=Male". For `hidden_layers_0` they were $a = 1.53, b = 0.0533$ *i.e.* $a = 1.31, b = 0.098$.

### A.2.3. EMPIRICAL FINDINGS WITH CONCEPT ACTIVATION REGIONS

Third, we demonstrate that our theoretical results extend to the generalization of hinge loss: Concept Activation Regions.

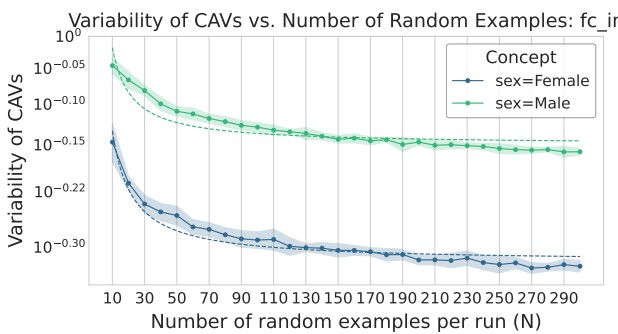 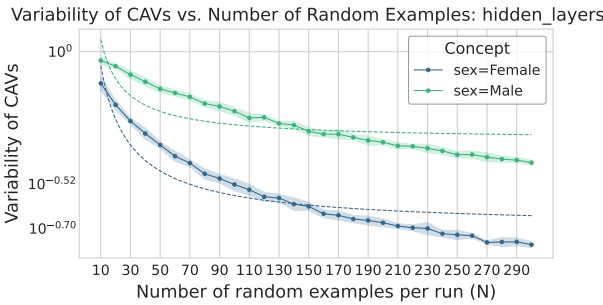

*Figure 22.* Mean variability of CARs as a function of the number of random examples per run ($N$) on the **UCI Adult dataset**. Results are shown for two different hidden layers. The CARs were obtained using the non-linear SVC as proposed by Crabbé & van der Schaar (2022). Variance is estimated by the sum of per-feature variances across ten independent runs. We fitted a curve of the form $f(N) = a/N + b$ to it. For `fc_in` the parameters were $2.78/N + 0.451$ for "sex=Female" and $2.76/N + 0.681$ for "sex=Male". For `hidden_layers_0` they were $6.03/N + 0.172$ *i.e.* $6.37/N + 0.412$.

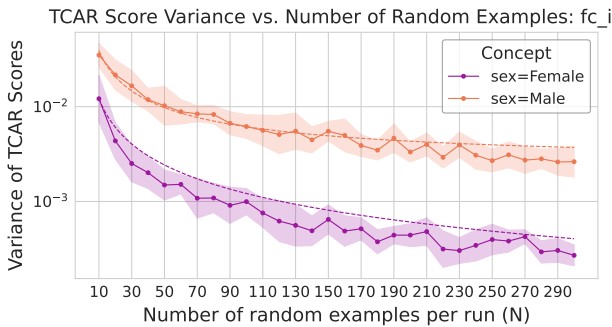 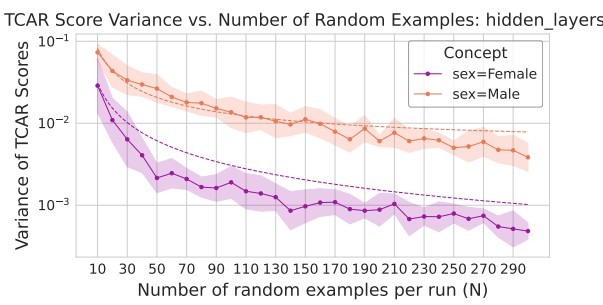

*Figure 23.* Variance of TCAR scores vs. the number of examples per concept set ($N$) for the concepts "male" and "female" on the **UCI Adult dataset**. The TCAR scores were obtained following the framework proposed by Crabbé & van der Schaar (2022). Error bars denote $\pm 1$ standard deviation. We fitted a curve of the form $f(N) = a/N + b$ to it. For `fc_in` the parameters were $a = 0.121, b = 2 \times 10^{-8}$ for concept "sex=Female" and $a = 0.364, b = 0.0025$ for concept "sex=Male". For `hidden_layers_0` they were $a = 0.306, b = 7.97 \times 10^{-10}$ *i.e.* $a = 0.769, b = 0.00526$.

### A.2.4. EMPIRICAL FINDINGS WITH DIFFERENCE OF MEANS

When CAVs are computed via the *Difference of Means*-method, the variance decline best resembles an $\mathcal{O}(1/N)$ rate, compared to the other two methods we evaluated.

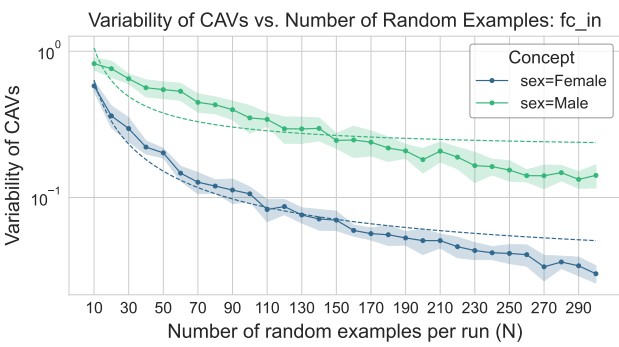 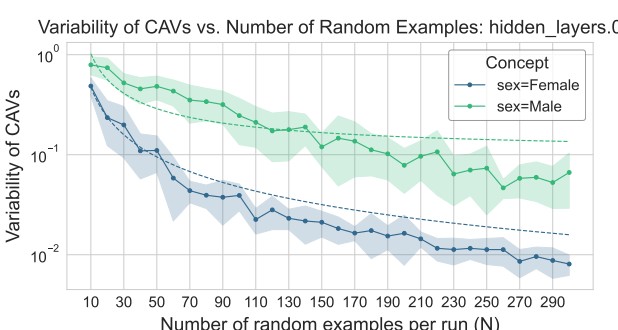

*Figure 24.* The linear classifiers for CAV generation were trained using **difference-of-means**. Error bars indicate $\pm 1$ SD; the $y$-axis is log-scaled. Variance is estimated by the sum of per-feature variances across ten independent runs. We fitted a curve of the form $f(N) = a/N + b$ to it. For `fc_in` the parameters were $6.05/N + 0.0306$ for "sex=Female", $8.44/N + 0.209$ for "sex=Male". For `hidden_layers_0` they were $4.75/N + 2.18 \times 10^{-13}$ *i.e.* $9.19/N + 0.105$. Moreover, for `hidden_layers_0` we clipped below $3 \times 10^{-3}$ to avoid visual distortion from extreme lows.

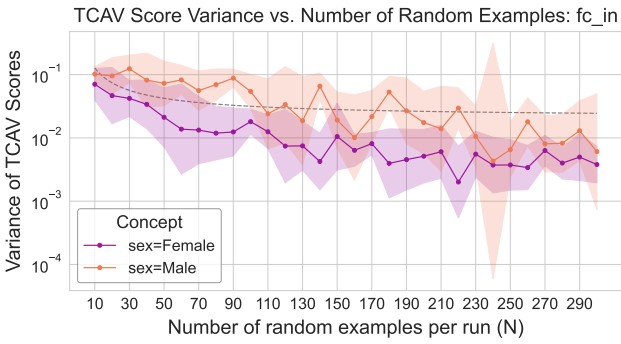 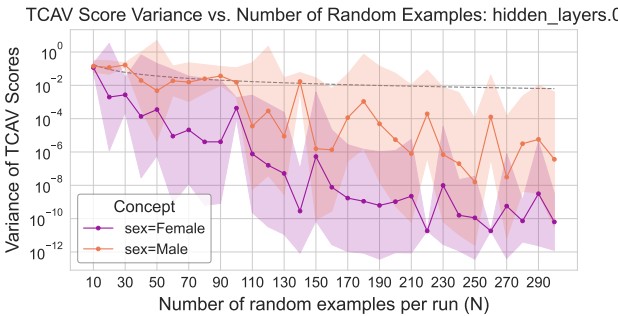

*Figure 25.* Mean geometrical variance of TCAV scores vs. the number of examples per concept set ($N$) for the concepts "male" and "female" on the **UCI Adult dataset**. The underlying CAVs were generated using **difference-of-means**. Error bars denote $\pm 1$ geometrical standard deviation. We fitted a curve of the form $f(N) = a/N + b$ to it. For `fc_in` the parameters were $a = 1.06, b = 0.0209$, for `hidden_layers_0` they were $a = 1.81, b = 3.05 \times 10^{-4}$.

### A.2.5. COMPUTE TRADE-OFF

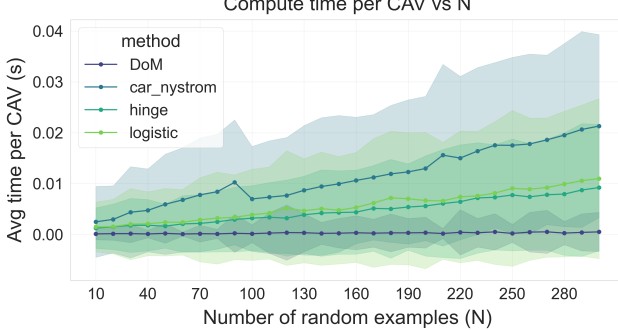 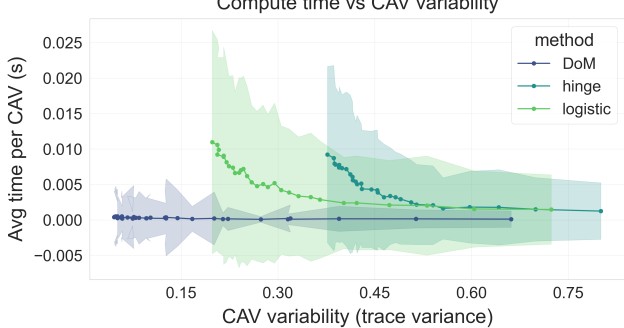

*(a)* **Compute vs.** $N$. Average wall-clock time per CAV as a function of the number of random examples per run ($N$). Shaded band denotes $\pm 1$ SD across runs.

*(b)* **Compute vs. variability.** Compute-stability trade-off: average time per CAV versus CAV variability (trace variance). Points correspond to different $N$ values (connected in increasing $N$). Shaded band denotes $\pm 1$ SD for compute time.

*Figure 26.* Compute cost and stability of CAV generation (see legend for the selected setting, e.g. layer/method). Left: compute time per CAV vs. $N$. Right: compute time vs. resulting CAV variability (trace variance), illustrating the compute-stability trade-off.

## A.3. TCAV for Text Data

### A.3.1. EMPIRICAL FINDINGS WITH BINARY CROSS-ENTROPY LOSS

For the text data (**IMDB**), we first report the results from the logistic regression classifier using binary cross-entropy loss. We also report the same two quantities for `SmolLM2-1.7B-Instruct` (Allal et al., 2025) at `decoder_hidden_6` and `decoder_hidden_12`.

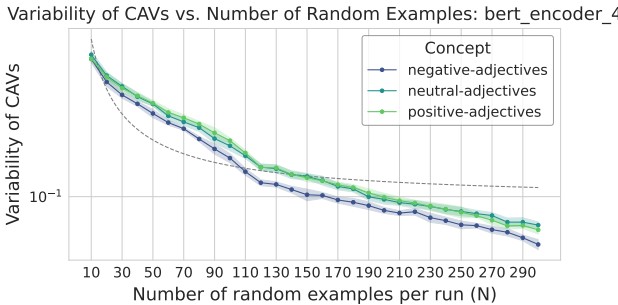

*Figure 27.* Mean variability of CAVs as a function of the number of random examples per run ($N$), shown for concepts at two different layers (`bert_encoder_2` on the left, `bert_encoder_4` on the right). Error bars indicate $\pm 1$ SD; the $y$-axis is log-scaled. Variance is estimated by the sum of per-feature variances across ten runs. We fitted a curve of the form $f(N) = a/N + b$ to it. For `bert_encoder_2` the parameters were $a = 4.02, b = 0.121$, for `bert_encoder_4` they were $a = 4.27, b = 0.0958$.

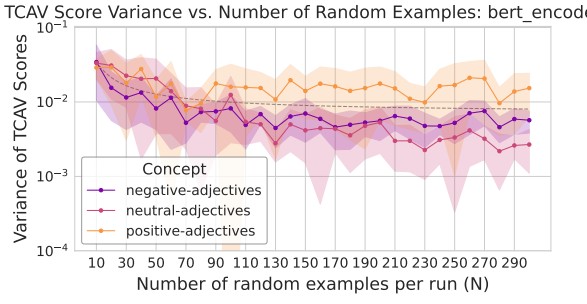
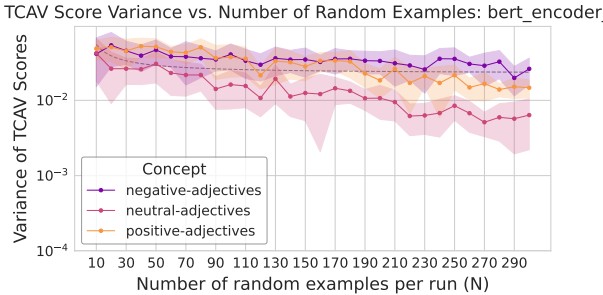

*Figure 28.* Variance of TCAV scores at layers `bert_encoder_2` and `bert_encoder_4` vs. the number of examples per concept set ($N$) for "positive"-, "negative"-, and "neutral"-adjective concepts on the IMDB dataset; error bars denote $\pm 1$ standard deviation. We fitted a curve of the form $f(N) = a/N + b$ to it. For `bert_encoder_2` the parameters were $a = 0.295, b = 0.00654$, for `bert_encoder_4` they were $a = 0.329, b = 0.0226$. For `bert_encoder_2` we cut the $log$-$y$-axis at $\geq 10^{-4}$ for readability to omit tiny standard deviation values.

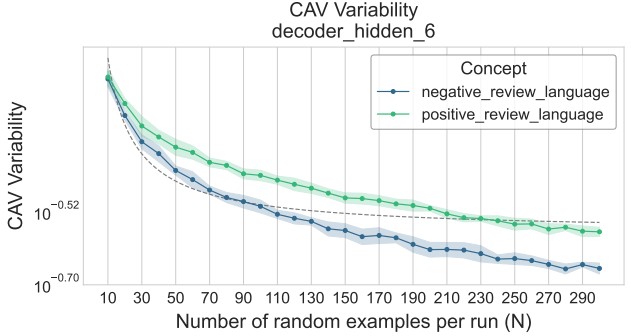
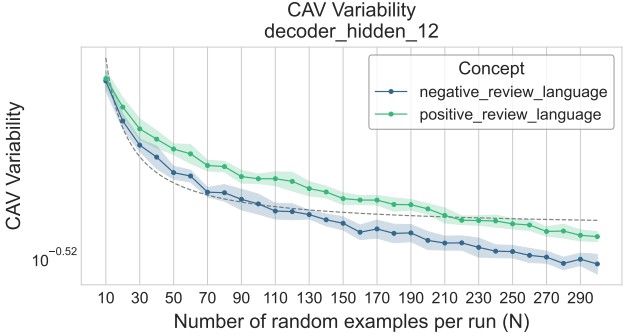

*Figure 29.* Mean variability of CAVs for `SmolLM2-1.7B-Instruct` as a function of the number of random examples per run ($N$) on the IMDB setting. Results are shown for decoder layers `decoder_hidden_6` and `decoder_hidden_12`. Error bars indicate $\pm 1$ SD; the $y$-axis is log-scaled.

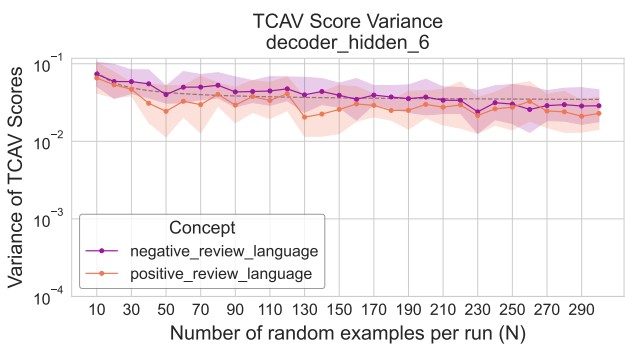
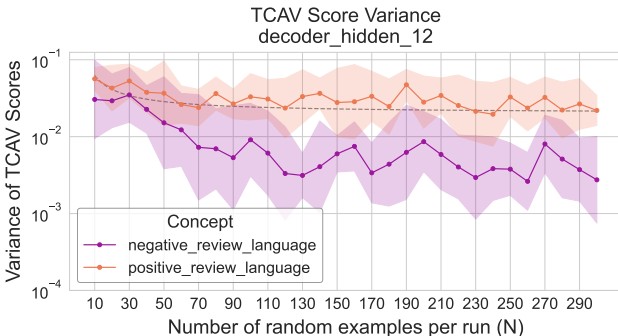

*Figure 30.* Variance of TCAV scores for `SmolLM2-1.7B-Instruct` at decoder layers `decoder_hidden_6` and `decoder_hidden_12` vs. the number of examples per concept set ($N$) on the IMDB setting. Error bars denote $\pm 1$ standard deviation.

### A.3.2. EMPIRICAL FINDINGS WITH HINGE LOSS

We now present the results for the `SGDLinearModel`, which uses hinge loss.

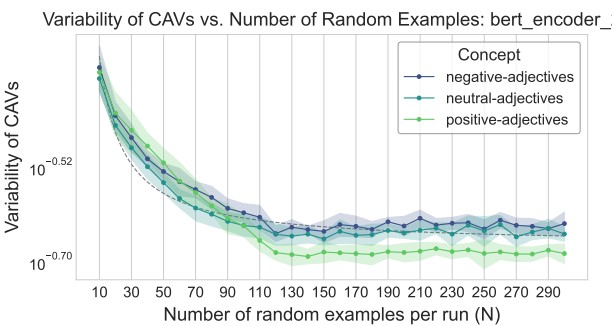
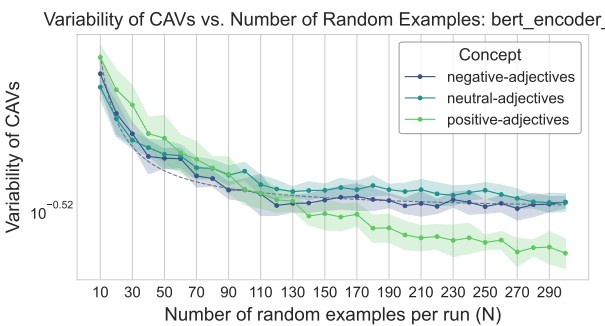

*Figure 31.* Mean variability of CAVs as a function of the number of random examples per run ($N$), shown for concepts at two different layers. Error bars indicate $\pm 1$ SD; the $y$-axis is log-scaled. Variance is estimated by the sum of per-feature variances across ten runs. We fitted a curve of the form $f(N) = a/N + b$ to it. For `bert_encoder_2` the parameters were $a = 2.69, b = 0.215$, for `bert_encoder_4` they were $a = 3.02, b = 0.298$.

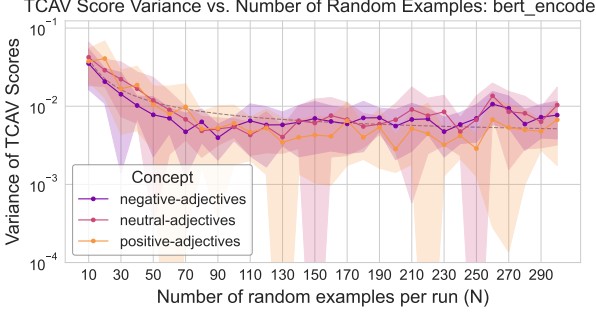
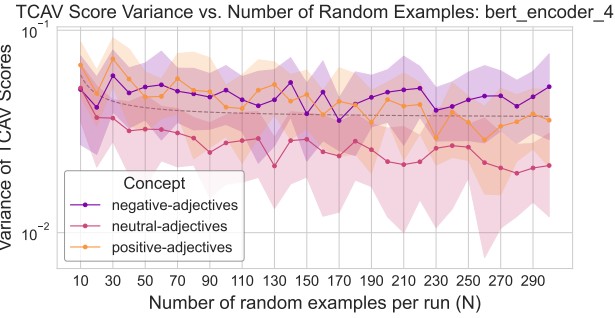

*Figure 32.* Variance of TCAV scores at layers `bert_encoder_2` and `bert_encoder_4` vs. the number of examples per concept set ($N$) for "positive"-, "negative"-, and "neutral"-adjective concepts on the IMDB dataset; error bars denote $\pm 1$ standard deviation. We fitted a curve of the form $f(N) = a/N + b$ to it. For `bert_encoder_2` the parameters were $a = 0.372, b = 0.00391$ and cut the log scale at $\geq 10^{-4}$ so near-zero bands do not dominate. For `bert_encoder_4` the fitted parameters were $a = 0.235$ and $b = 0.0368$.

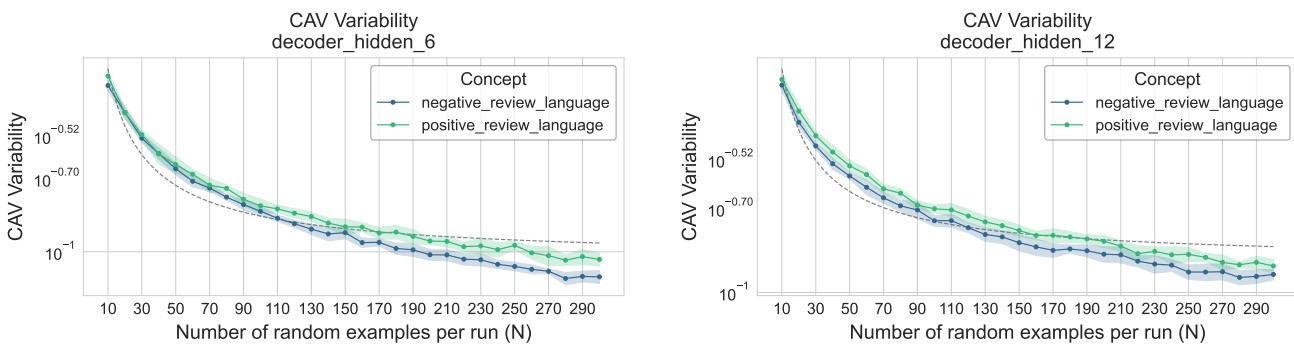

*Figure 33.* Mean variability of CAVs for `SmolLM2-1.7B-Instruct` as a function of the number of random examples per run ($N$) on the IMDB setting. Results are shown for decoder layers `decoder_hidden_6` and `decoder_hidden_12`. Error bars indicate $\pm 1$ SD; the $y$-axis is log-scaled.

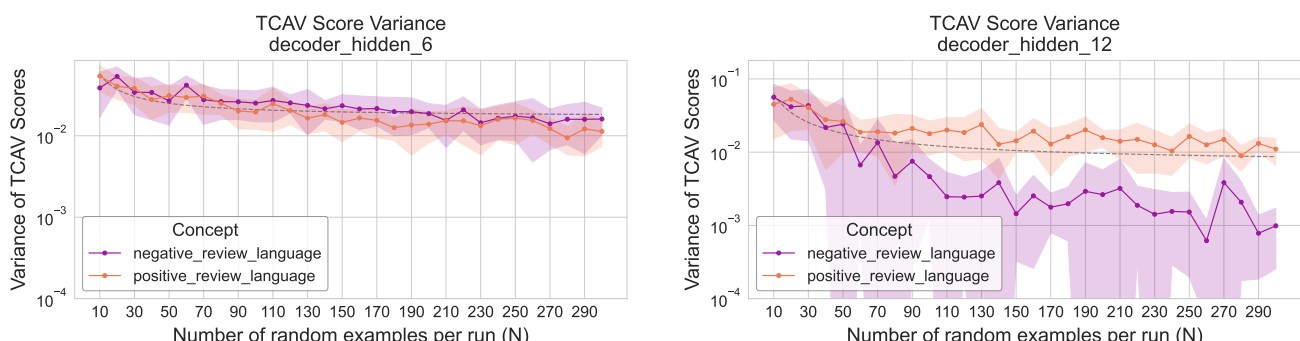

*Figure 34.* Variance of TCAV scores for `SmolLM2-1.7B-Instruct` at decoder layers `decoder_hidden_6` and `decoder_hidden_12` vs. the number of examples per concept set ($N$) on the IMDB setting. Error bars denote $\pm 1$ standard deviation.

### A.3.3. EMPIRICAL FINDINGS WITH CONCEPT ACTIVATION REGIONS

In the following, we report results for CARs and TCAR scores obtained with non-linear Support Vector Machines.

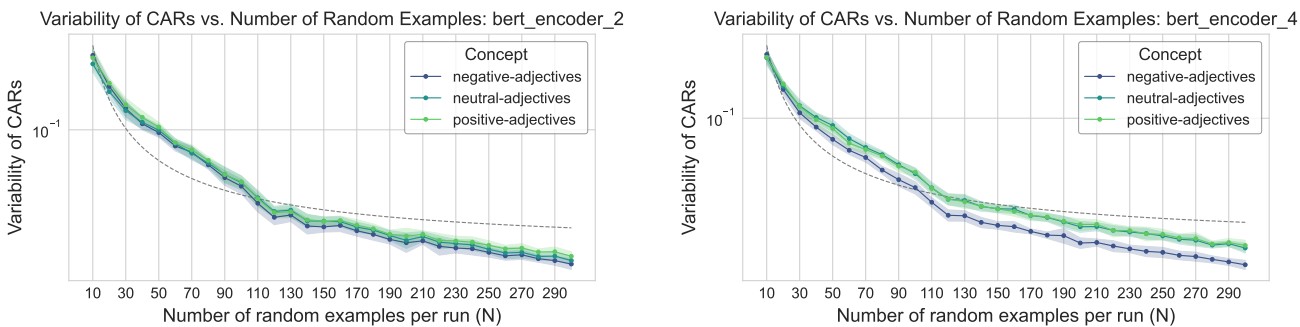

*Figure 35.* Mean variability of CARs as a function of the number of random examples per run ($N$), shown for concepts at two different layers. Error bars indicate $\pm 1$ SD; the $y$-axis is log-scaled. Variance is estimated by the sum of per-feature variances across ten runs. We fitted a curve of the form $f(N) = a/N + b$ to it. For `bert_encoder_2` the parameters were $a = 2.69, b = 0.215$, for `bert_encoder_4` they were $a = 3.02, b = 0.298$.

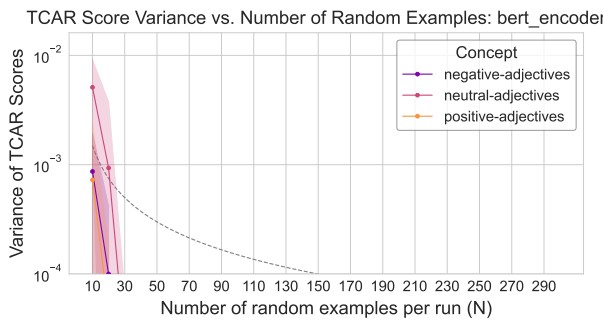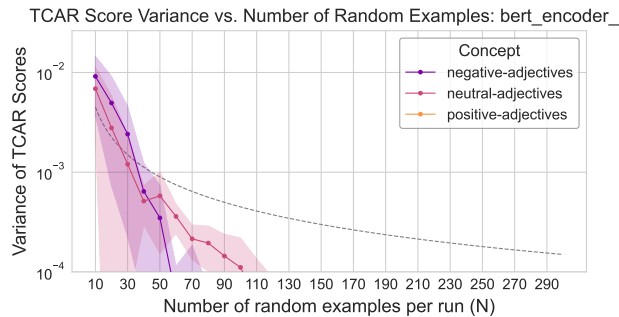

*Figure 36.* Variance of TCAR scores at layers `bert_encoder_2` and `bert_encoder_4` vs. the number of examples per concept set ($N$) for "positive"-, "negative"-, and "neutral"-adjective concepts on the IMDB dataset; error bars denote $\pm 1$ standard deviation. We fitted a curve of the form $f(N) = a/N + b$ to it. For `bert_encoder_2` the parameters were $a = 0.372, b = 0.00391$ and cut the log scale at $\geq 10^{-4}$ so near-zero bands do not dominate. For `bert_encoder_4` the fitted parameters were $a = 0.235$ and $b = 0.0368$.

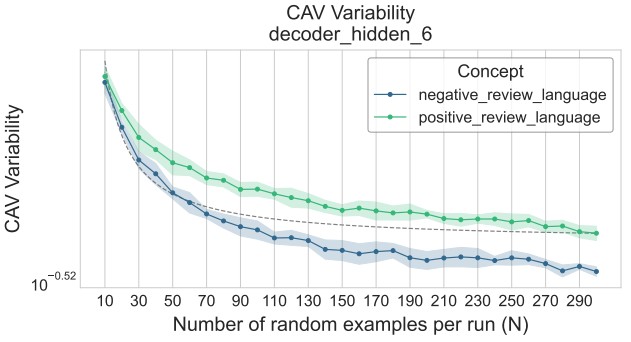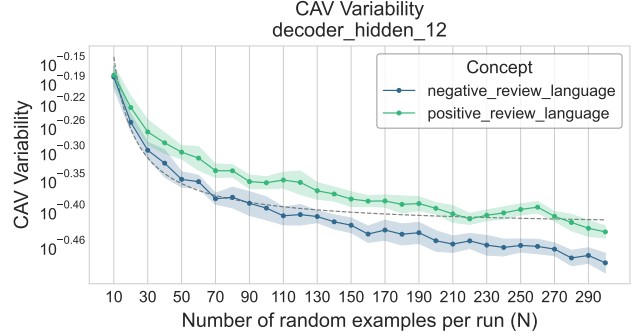

*Figure 37.* Mean variability of CARs for `SmolLM2-1.7B-Instruct` as a function of the number of random examples per run ($N$) on the IMDB setting. Results are shown for decoder layers `decoder_hidden_6` and `decoder_hidden_12`. Error bars indicate $\pm 1$ SD; the $y$-axis is log-scaled.

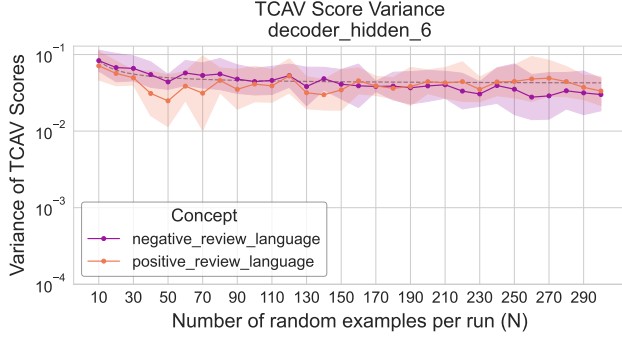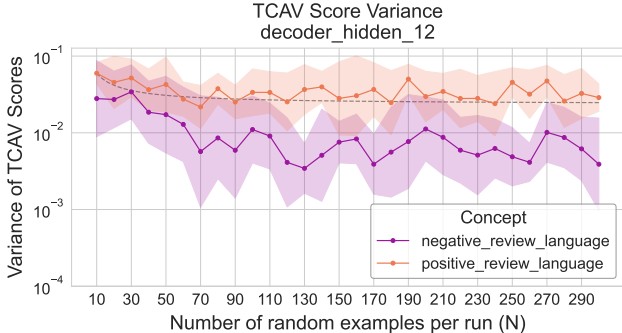

*Figure 38.* Variance of TCAR scores for `SmolLM2-1.7B-Instruct` at decoder layers `decoder_hidden_6` and `decoder_hidden_12` vs. the number of examples per concept set ($N$) on the IMDB setting. Error bars denote $\pm 1$ standard deviation.

### A.3.4. EMPIRICAL FINDINGS WITH DIFFERENCE OF MEANS

Again, we found that CAVs computed via the *Difference of Means*-method most closely follow a variance decline of $\mathcal{O}(1/N)$.

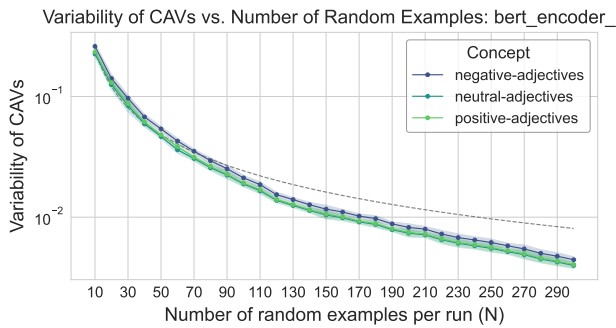 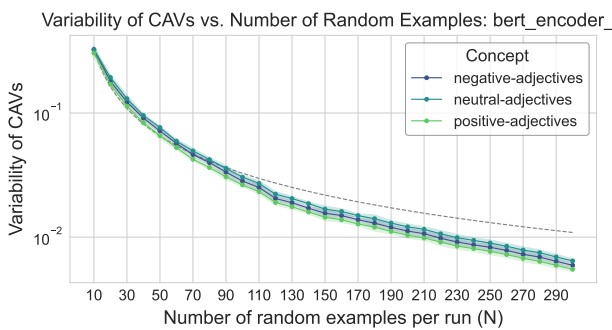

*Figure 39.* Mean variability of CAVs for the concepts "positive", "negative" and "neutral" adjectives as a function of the number of random examples per run ($N$). Results are shown for two different hidden layers. The CAV are generated using **difference-of-means**. Error bars indicate $\pm 1$ SD; the $y$-axis is log-scaled. Variance is estimated by the sum of per-feature variances across ten independent runs. We fitted a curve of the form $f(N) = a/N + b$ to it. For `bert_encoder_2` the parameters were $a = 2.42, b = 1.19 \times 10^{-7}$, for `bert_encoder_4` they were $a = 3.27, b = 3.24 \times 10^{-10}$.

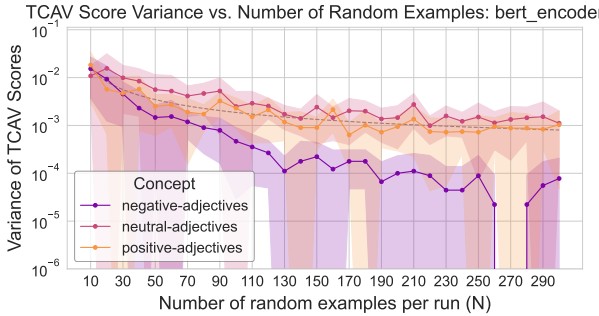 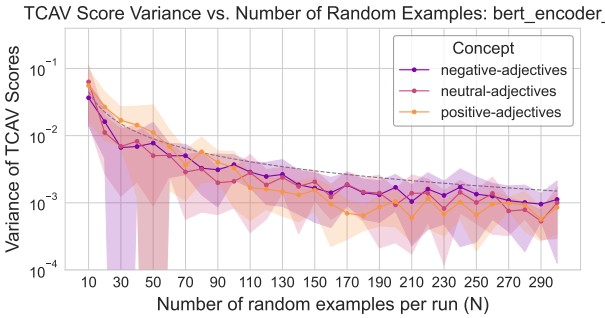

*Figure 40.* Variance of TCAV scores vs. the number of examples per concept set ($N$) for the concepts "positive", "negative" and "neutral" adjectives. The underlying CAVs were generated using **difference-of-means**. Error bars denote $\pm 1$ standard deviation. We fitted a curve of the form $f(N) = a/N + b$ to it. For `bert_encoder_2` the parameters were $a = 0.16, b = 2.7 \times 10^{-4}$, for `bert_encoder_4` they were $a = 0.449, b = 1.59 \times 10^{-8}$. We clipped below $10^{-6}$ respectively $10^{-4}$ to avoid visual distortion from extreme lows.

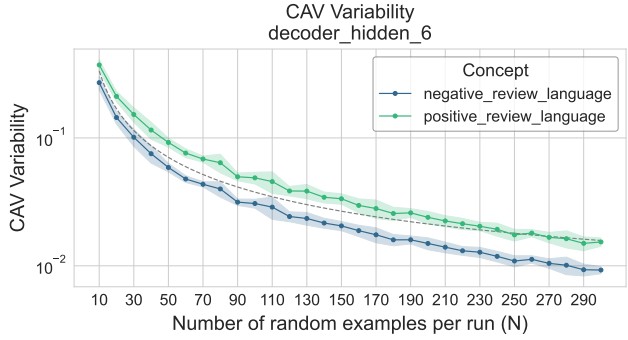 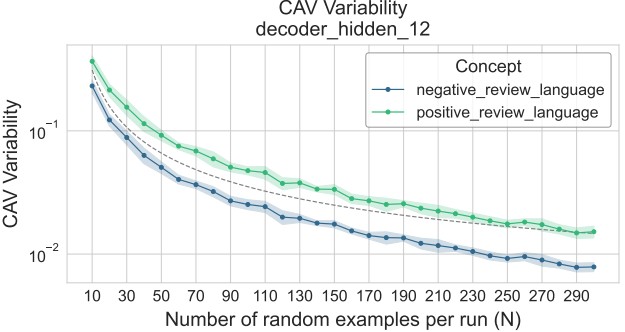

*Figure 41.* Mean variability of CAVs for `SmolLM2-1.7B-Instruct` as a function of the number of random examples per run ($N$) on the IMDB setting. Results are shown for decoder layers `decoder_hidden_6` and `decoder_hidden_12`. Error bars indicate $\pm 1$ SD; the $y$-axis is log-scaled.

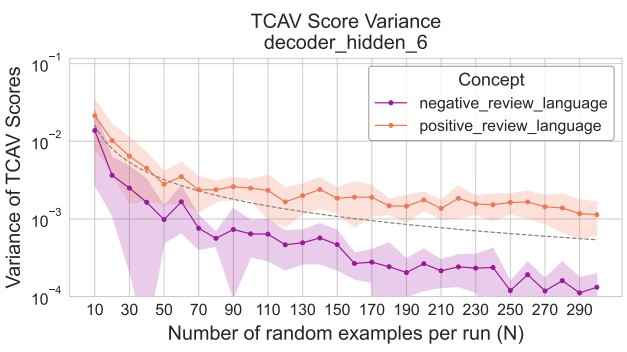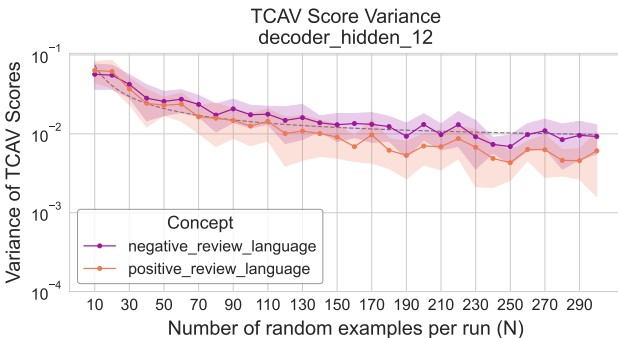

*Figure 42.* Variance of TCAV scores for `SmolLM2-1.7B-Instruct` at decoder layers `decoder_hidden_6` and `decoder_hidden_12` vs. the number of examples per concept set ($N$) on the IMDB setting. Error bars denote $\pm 1$ standard deviation.

## A.4. TCAV for Multimodal Data

We evaluate TCAV on **multimodal** (vision-language) data using **CLIP** (Radford et al., 2021) and LLaVA-1.5 (Liu et al., 2024). We pair zebra images with the prompt "a photo of a zebra" and use image-text similarity as the scalar output. Concepts are defined by **image sets**, as in the vision experiments (e.g., "striped", "zigzagged", "dotted"). We extract CAVs from CLIP at hidden layers $4$ and $8$ and from LLaVA-1.5 at vision layers $7$ and $15$. We present the results for this final dataset below.

### A.4.1. EMPIRICAL FINDINGS WITH BINARY CROSS-ENTROPY LOSS

We first report results for multimodal data using a logistic regression classifier trained with **binary cross-entropy loss**.

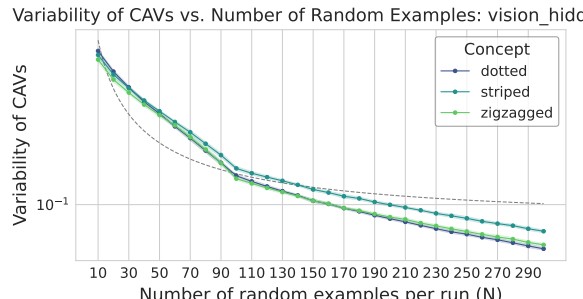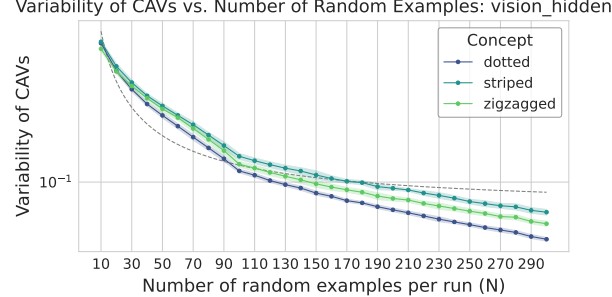

*Figure 43.* Mean variability of CAVs for the visual concepts "striped", "zigzagged", and "dotted" as a function of random examples ($N$) for CLIP hidden layers $4$ and $8$. Error bars indicate $\pm 1$ SD; the $y$-axis is log-scaled. We fitted a curve of the form $f(N) = a/N + b$ to it. For the layer `vision_hidden_4` we found $6.72/N + 0.0788$, for `vision_hidden_8` $4.81/N + 0.0731$.

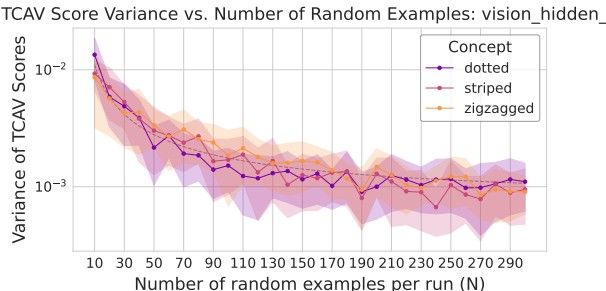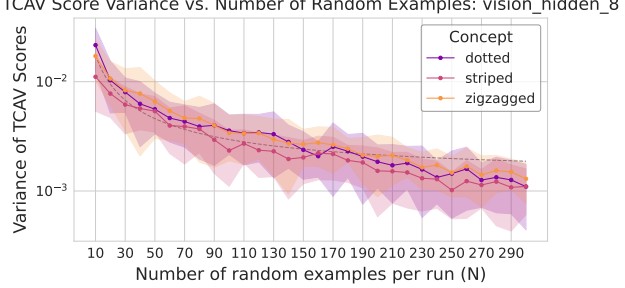

*Figure 44.* Variance of TCAV scores at CLIP hidden layers $4$ and $8$ vs. the number of examples per concept set ($N$) for the visual concepts "striped", "zigzagged", and "dotted". Error bars denote $\pm 1$ standard deviation. We fitted a curve of the form $f(N) = a/N + b$ to it. For the layer `vision_hidden_4` we found $0.103/N + 7.27 \times 10^{-4}$, for `vision_hidden_8` we found $0.163/N + 0.00133$.

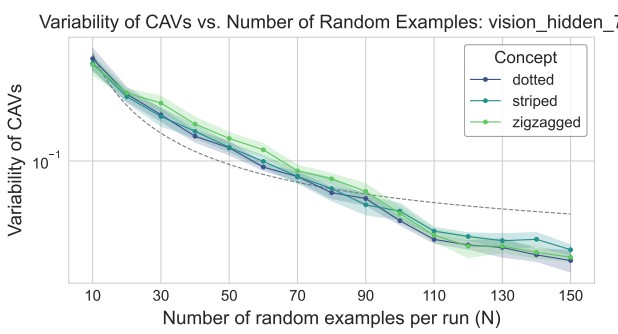 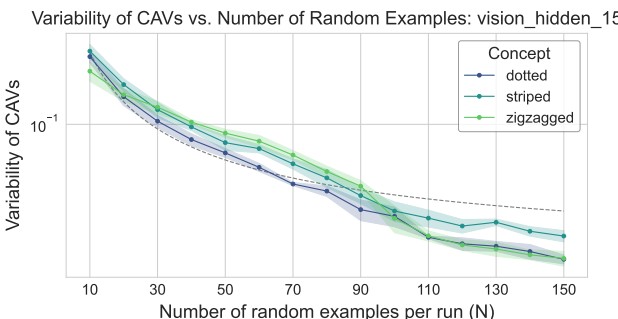

*Figure 45.* Mean variability of CAVs for LLaVA-1.5 as a function of the number of random examples per run ($N$) for Broden-style visual concepts. Results are shown for vision layers `vision_hidden_7` and `vision_hidden_15`. Error bars indicate $\pm 1$ SD; the $y$-axis is log-scaled.

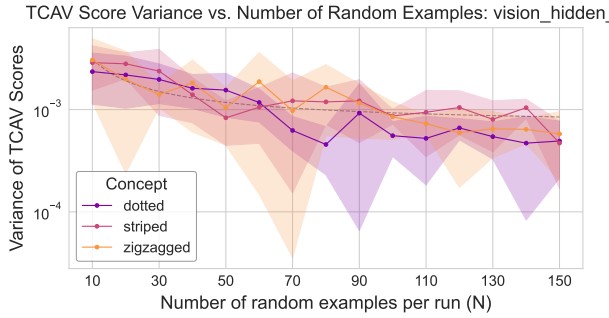 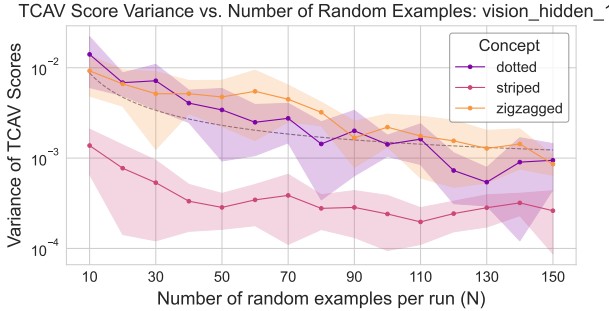

*Figure 46.* Variance of TCAV scores for LLaVA-1.5 at vision layers `vision_hidden_7` and `vision_hidden_15` vs. the number of examples per concept set ($N$) for Broden-style visual concepts. Error bars denote $\pm 1$ standard deviation.

### A.4.2. EMPIRICAL FINDINGS WITH HINGE LOSS

Next, we show the corresponding results with **hinge loss**.

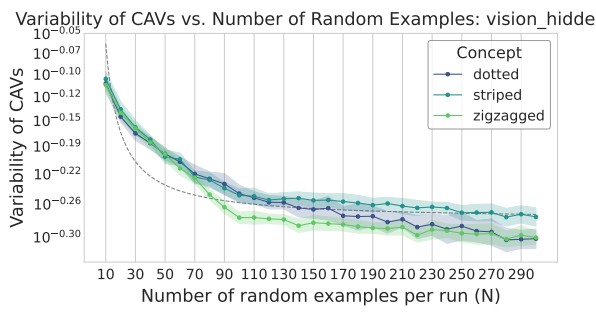 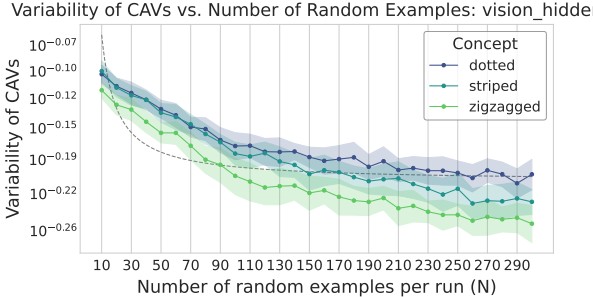

*Figure 47.* Mean variability of CAVs for the visual concepts "striped", "zigzagged", and "dotted" as a function of the number of random examples per run ($N$). Results are shown for CLIP hidden layers 4 and 8. Error bars indicate $\pm 1$ SD; the $y$-axis is log-scaled. Variance is estimated by the sum of per-feature variances across five independent runs. We fitted a curve of the form $f(N) = a/N + b$ to it.

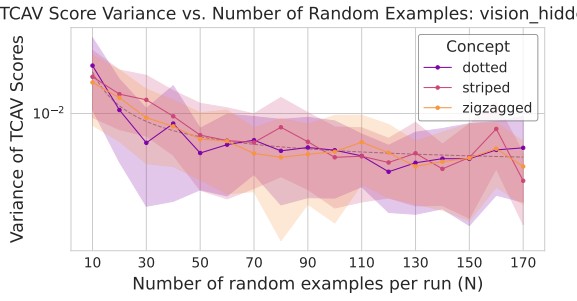 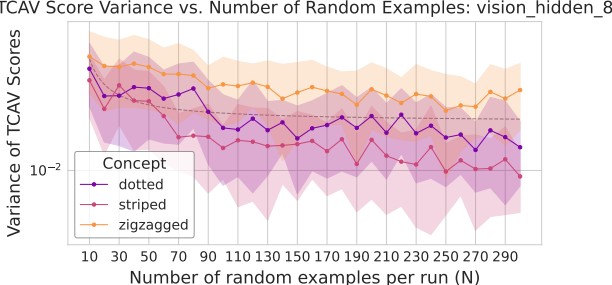

*Figure 48.* Variance of TCAV scores at CLIP hidden layers $4$ and $8$ vs. the number of examples per concept set ($N$) for the visual concepts "striped", "zigzagged", and "dotted". The underlying CAVs were trained using **hinge loss**. Error bars denote $\pm 1$ standard deviation. We fitted a curve of the form $f(N) = a/N + b$ to it. For `vision_hidden_4` $0.122/N + 0.00496$, for `vision_hidden_8` $0.223/N + 0.0183$.

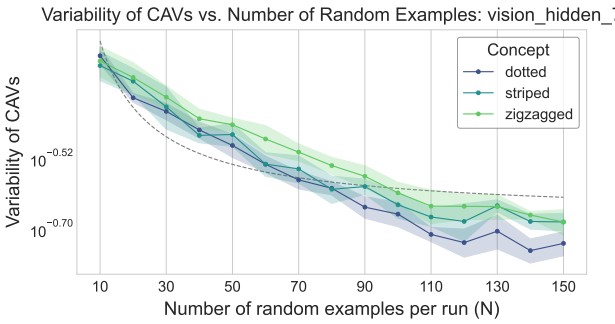 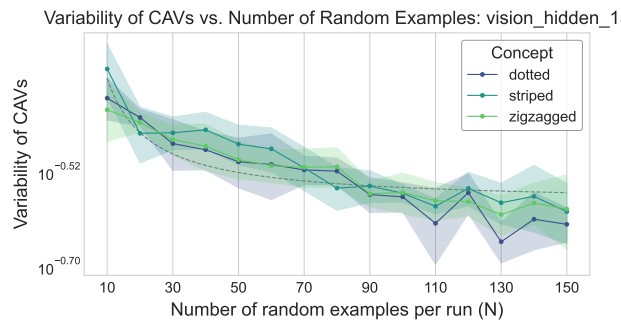

*Figure 49.* Mean variability of CAVs for LLaVA-1.5 as a function of the number of random examples per run ($N$) for Broden-style visual concepts. Results are shown for vision layers `vision_hidden_7` and `vision_hidden_15`. Error bars indicate $\pm 1$ SD; the $y$-axis is log-scaled.

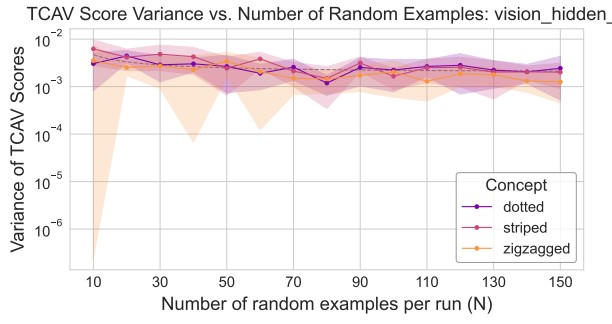 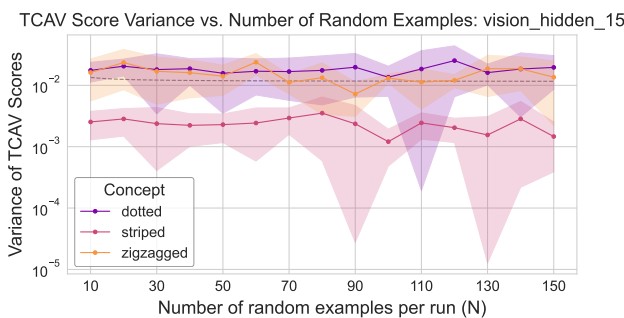

*Figure 50.* Variance of TCAV scores for LLaVA-1.5 at vision layers `vision_hidden_7` and `vision_hidden_15` vs. the number of examples per concept set ($N$) for Broden-style visual concepts. Error bars denote $\pm 1$ standard deviation.

### A.4.3. EMPIRICAL FINDINGS WITH CONCEPT ACTIVATION REGIONS

Second, we prove that our analysis generalizes to Concept Activation Regions.

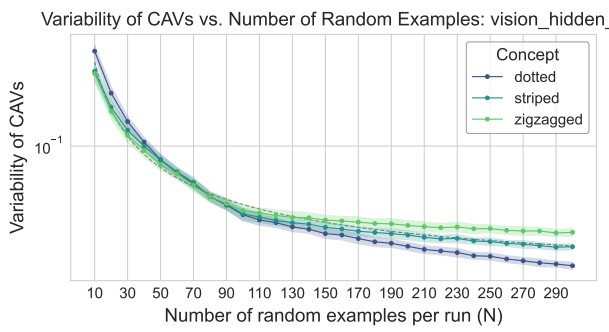
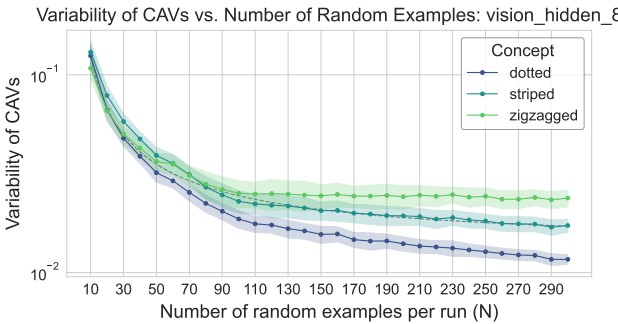

*Figure 51.* Mean variability of CARs for the visual concepts "striped", "zigzagged", and "dotted" as a function of the number of random examples per run ($N$). Results are shown for CLIP hidden layers 4 and 8. Error bars indicate $\pm 1$ SD; the $y$-axis is log-scaled. Variance is estimated by the sum of per-feature variances across five independent runs. We fitted a curve of the form $f(N) = a/N + b$ to it. For `vision_hidden_4` $2.91/N + 0.0168$, for `vision_hidden_8` $1.09/N + 0.0135$.

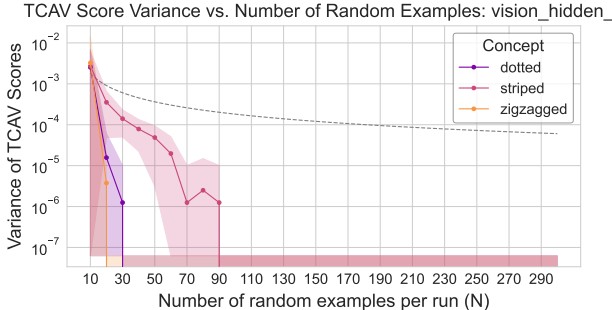
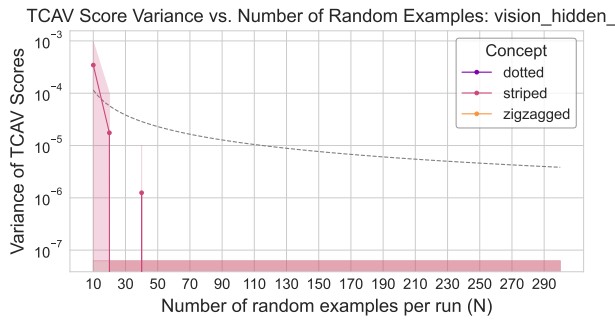

*Figure 52.* Variance of TCAR scores at CLIP hidden layers 4 and 8 vs. the number of examples per concept set ($N$) for the visual concepts "striped", "zigzagged", and "dotted". The underlying CARs were trained using a non-linear SVM. Error bars denote $\pm 1$ standard deviation. We fitted a curve of the form $f(N) = a/N + b$ to it.

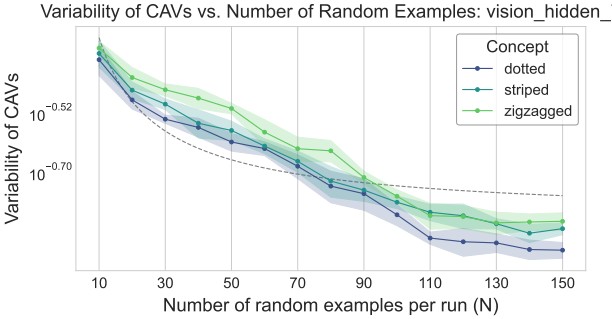
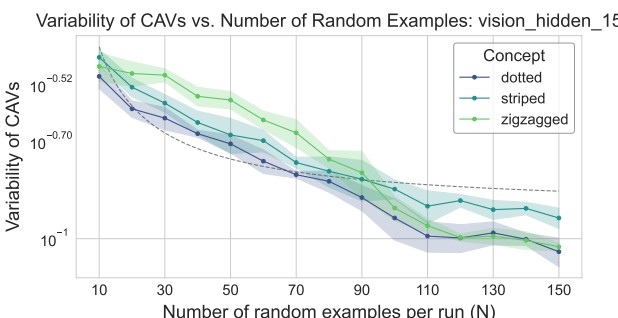

*Figure 53.* Mean variability of CARs for LLaVA-1.5 as a function of the number of random examples per run ($N$) for Broden-style visual concepts. Results are shown for vision layers `vision_hidden_7` and `vision_hidden_15`. Error bars indicate $\pm 1$ SD; the $y$-axis is log-scaled.

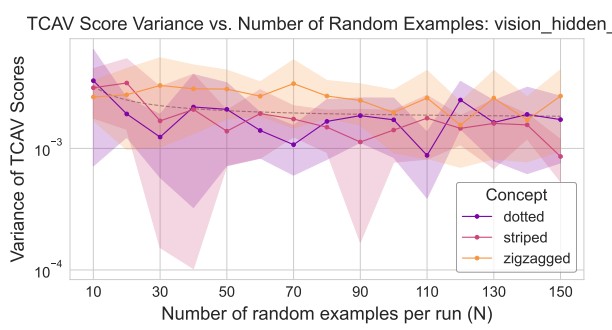
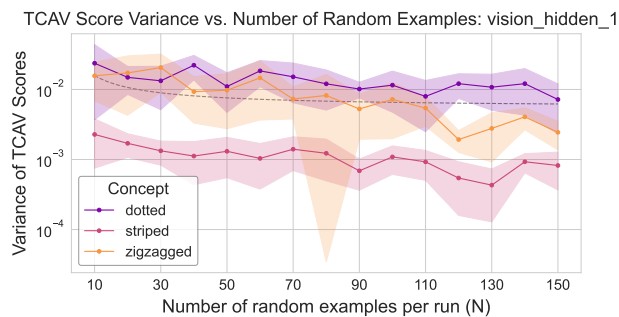

*Figure 54.* Variance of TCAR scores for LLaVA-1.5 at vision layers `vision_hidden_7` and `vision_hidden_15` vs. the number of examples per concept set ($N$) for Broden-style visual concepts. Error bars denote $\pm 1$ standard deviation.

### A.4.4. EMPIRICAL FINDINGS WITH DIFFERENCE OF MEANS

Again, we found that CAVs computed via the *Difference of Means* method most closely follow a variance decline of $\mathcal{O}(1/N)$.

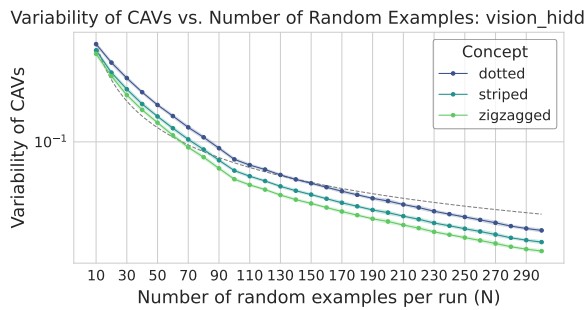
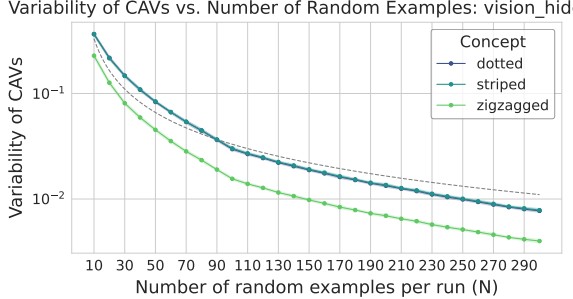

*Figure 55.* Mean variability of CAVs in dependence of the number of random examples per run $N$ for the CLIP model at hidden layers $4$ and $8$. The CAVs were generated using **difference-of-means**. Error bars indicate $\pm 1$ SD; the $y$-axis is log-scaled. Variance is estimated by the sum of per-feature variances across five independent runs. We fitted a curve of the form $f(N) = a/N + b$ to it.

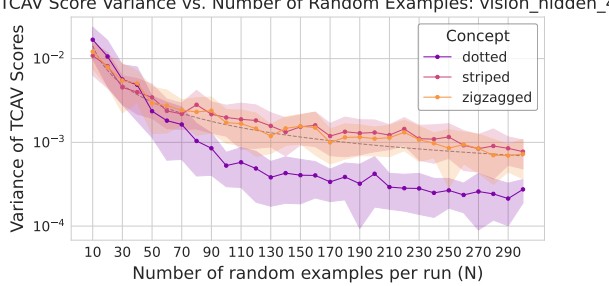
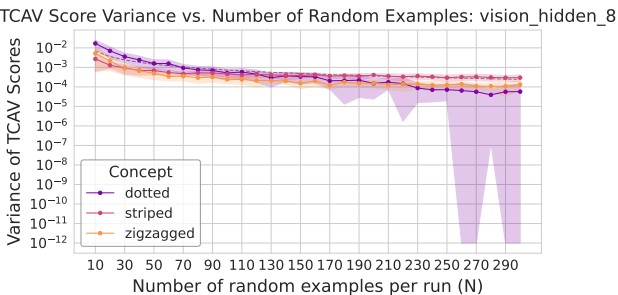

*Figure 56.* Variance of TCAV scores at CLIP hidden layers $4$ and $8$ vs. the number $N$ of random examples used for the visual concepts "striped", "zigzagged", and "dotted". The underlying CAVs were generated using **difference-of-means**. Error bars denote $\pm 1$ standard deviation. We fitted a curve of the form $f(N) = a/N + b$ to it.

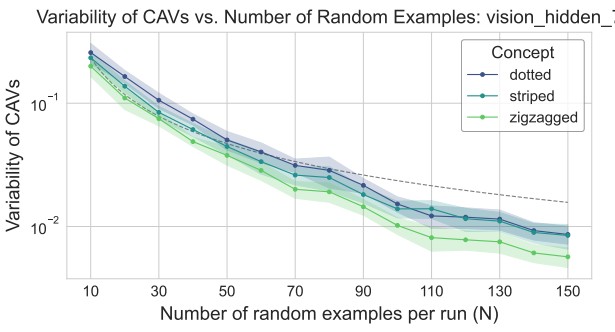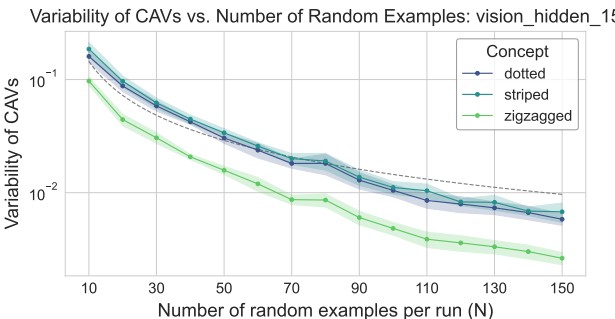

*Figure 57.* Mean variability of CᴀVs for LLaVA-1.5 as a function of the number of random examples per run ($N$) for Broden-style visual concepts. Results are shown for vision layers `vision_hidden_7` and `vision_hidden_15`. Error bars indicate $\pm 1$ SD; the $y$-axis is log-scaled.

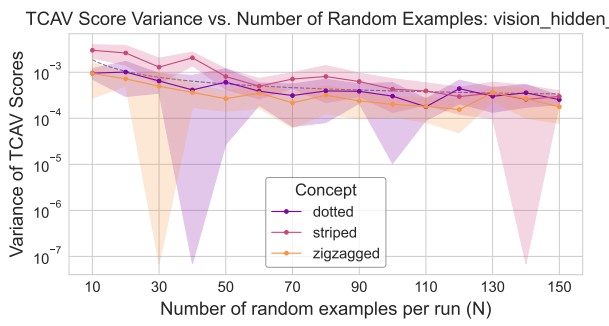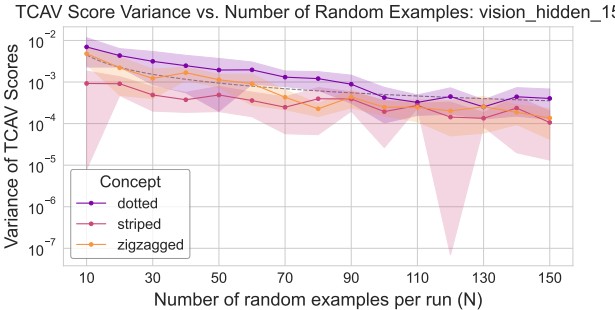

*Figure 58.* Variance of Tᴄᴀᴠ scores for LLaVA-1.5 at vision layers `vision_hidden_7` and `vision_hidden_15` vs. the number of examples per concept set ($N$) for Broden-style visual concepts. Error bars denote $\pm 1$ standard deviation.

### A.4.5. Cᴏᴍᴘᴜᴛᴇ Tʀᴀᴅᴇ-ᴏғғ

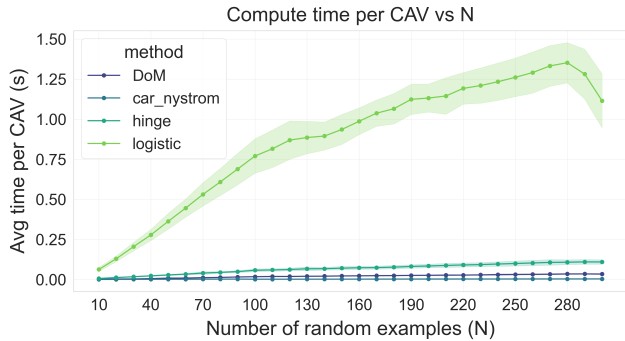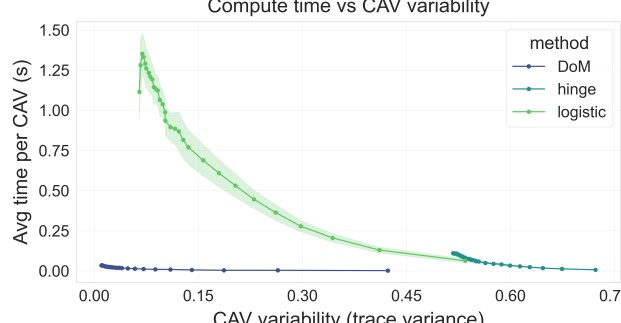

*(a)* **Compute vs. $N$.** Average wall-clock time per CᴀV as a function of the number of random examples per run ($N$). Shaded band denotes $\pm 1$ SD across runs.

*(b)* **Compute vs. variability.** Compute-stability trade-off: average time per CᴀV versus CᴀV variability (trace variance). Points correspond to different $N$ values (connected in increasing $N$). Shaded band denotes $\pm 1$ SD for compute time.

*Figure 59.* Compute cost and stability of CᴀV generation (see legend for the selected setting, e.g. layer/method). Left: compute time per CᴀV vs. $N$. Right: compute time vs. resulting CᴀV variability (trace variance), illustrating the compute-stability trade-off.

### A.5. Concept-Set-Size Variation

The main analysis in the paper studies the regime where the number of concept examples is fixed and the number of random reference examples $N$ increases. This reflects the common Tᴄᴀᴠ setting in which annotated concept examples are scarce, while random reference examples are comparatively easy to sample. In this appendix, we study the complementary regime: we fix the reference protocol and vary the number of concept examples.

We denote by $M$ the number of concept examples used in a run. For each value of $M$, we subsample concept examples from the same concept pool, recompute the CAV, and evaluate the same three quantities as in the main experiments: CAV variability, concept-sensitivity variance, and TCAV-score variance. This experiment isolates the finite-sample effect of the concept set size. It should not be confused with concept quality: increasing $M$ can reduce sampling variability, but a larger concept set can still be semantically poor if the examples do not faithfully represent the intended concept.

Across modalities, the results are consistent with the main message of the paper. Increasing the number of concept examples generally stabilizes the CAV direction and the continuous concept-sensitivity scores. TCAV scores are again more sensitive, because the hard sign threshold can amplify the effect of borderline evaluation points.

### A.5.1. IMAGE DATA

We first vary the number of concept examples in the image setting. As in the main image experiments, the concepts are visual concepts such as *striped*, *zigzagged*, and *dotted*. We report results for the same ResNet layers used throughout the image appendix, namely `layer2` and `layer3`.

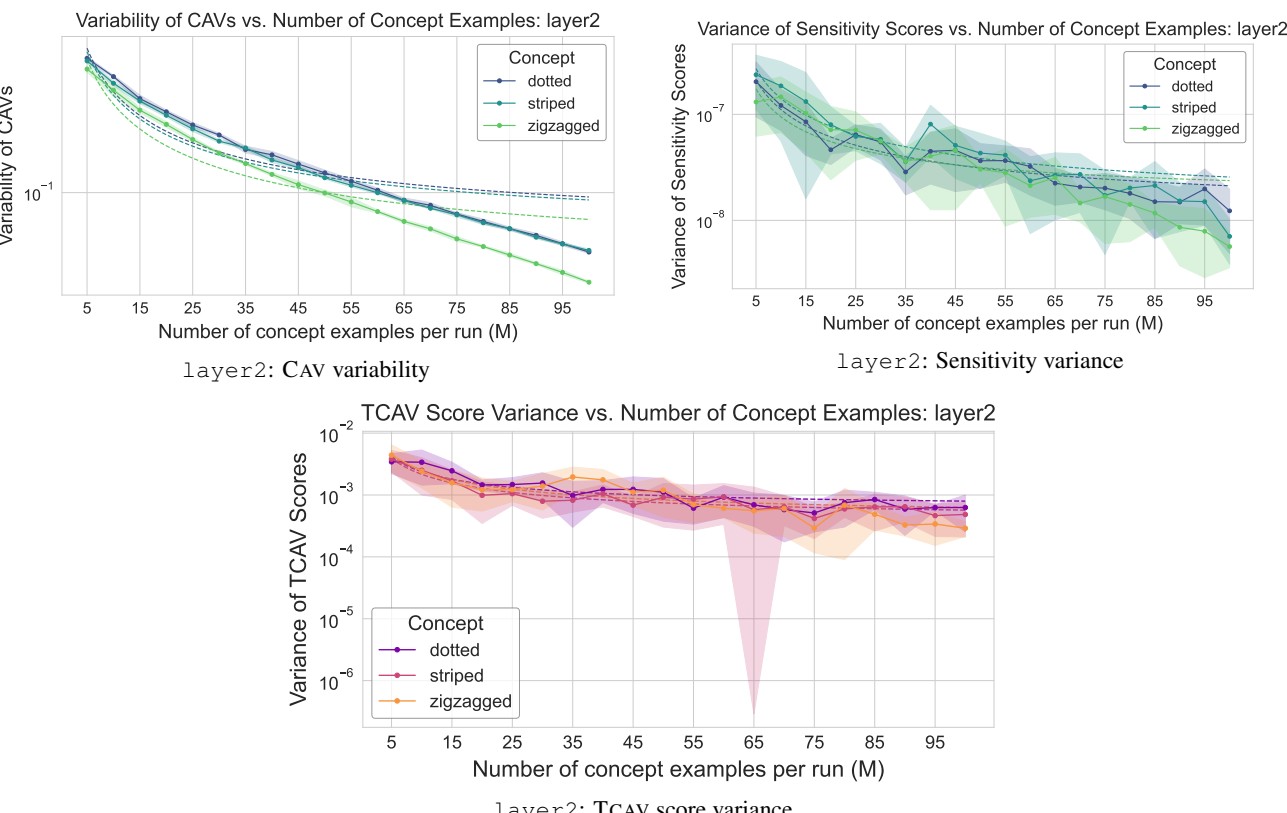

*Figure 60.* **Concept-set-size ablation for image data at ResNet `layer2`.** We vary the number $M$ of concept examples while keeping the reference protocol fixed. CAV variability and concept-sensitivity variance decrease as more concept examples are used. TCAV-score variance is less smooth because it depends on the sign of the sensitivity scores.

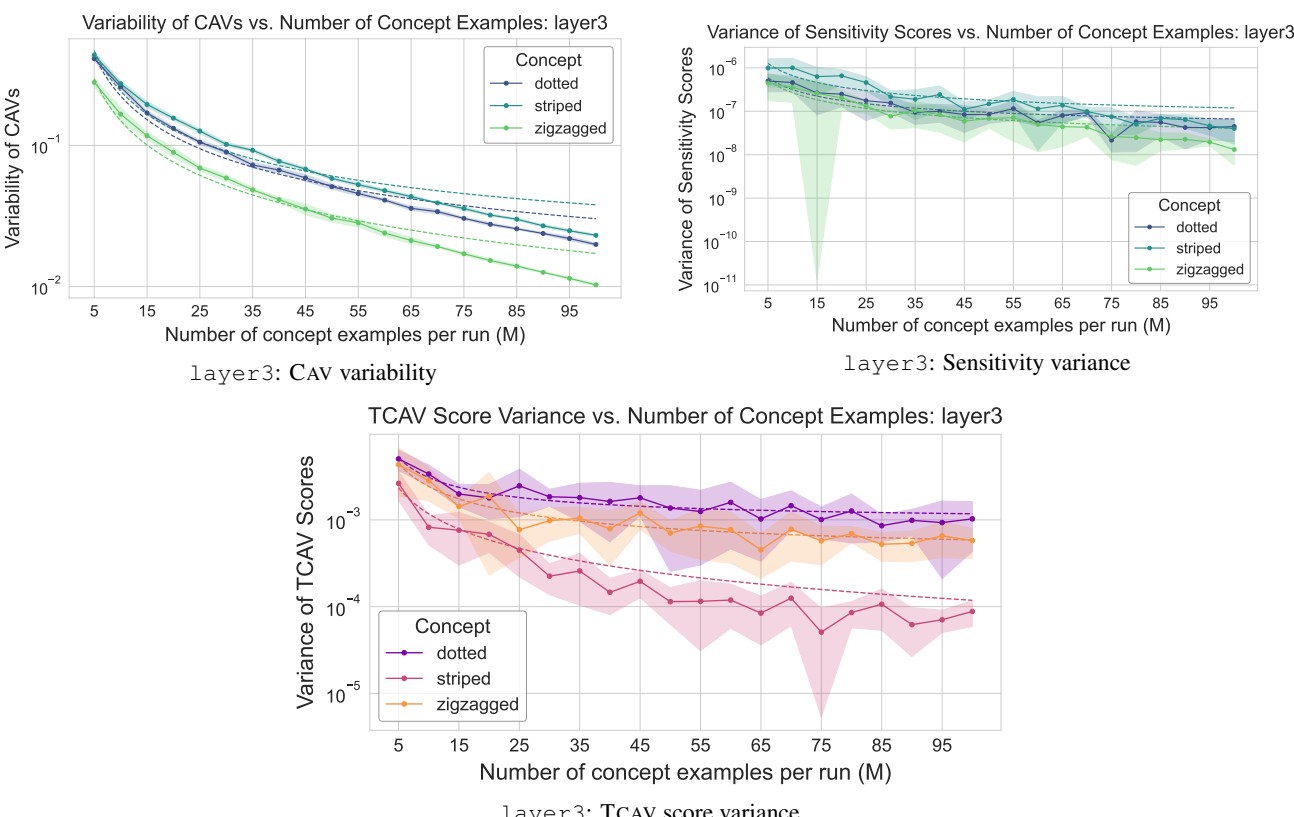

*Figure 61.* **Concept-set-size ablation for image data at ResNet `layer3`.** At `layer3`, increasing the number of concept examples again stabilizes the C$_{\text{AV}}$ direction.

### A.5.2. Tabular Data

We next consider the UCI Adult tabular setting. The concepts are derived from the gender attribute, namely `sex=Female` and `sex=Male`. We report results for the tabular layers that also appear in the other tabular experiments: the input projection layer `fc_in` and the hidden layer `hidden_layers.0`.

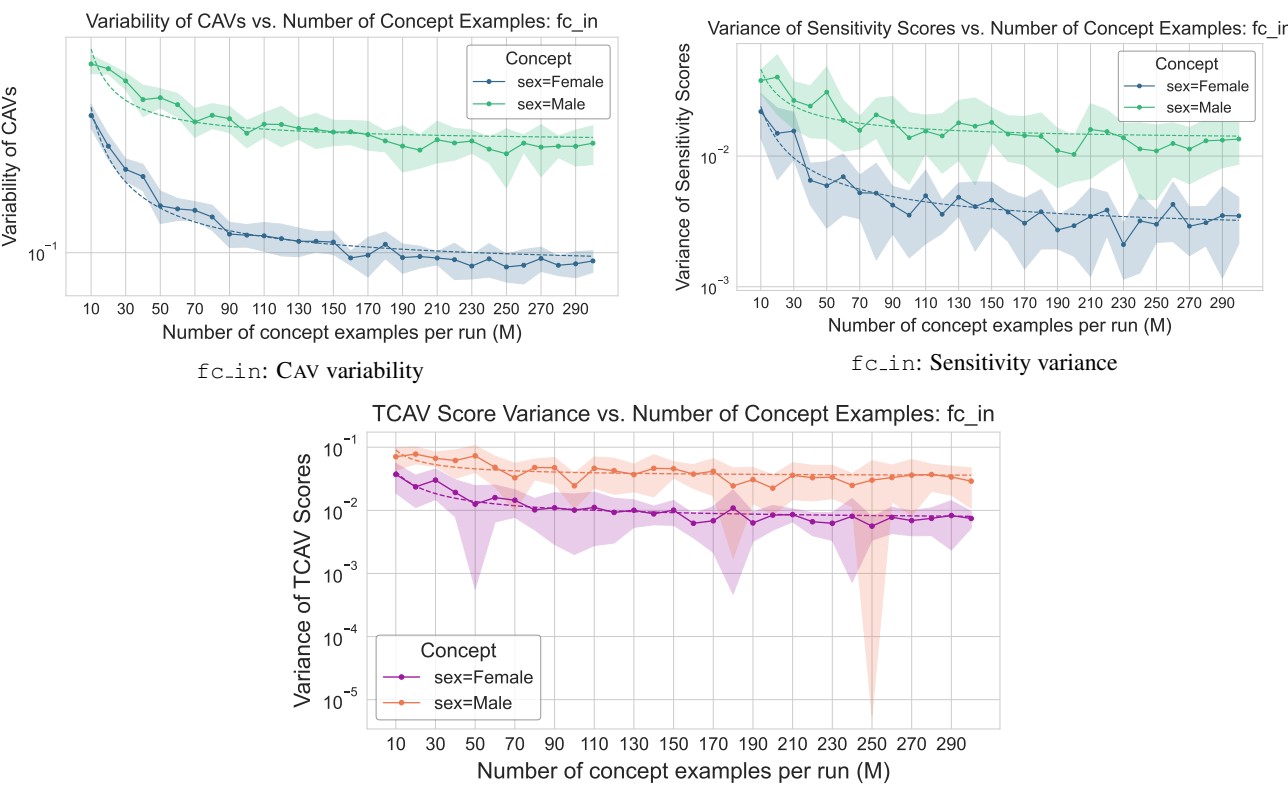

`fc_in`: CAV variability

`fc_in`: Sensitivity variance

`fc_in`: TCAV score variance

*Figure 62.* **Concept-set-size ablation for tabular data at `fc_in`.** For the input projection layer, increasing $M$ reduces the variability of the learned concept direction. Since the tabular concepts are defined directly from an interpretable attribute, this setting isolates the effect of concept-set size particularly clearly.

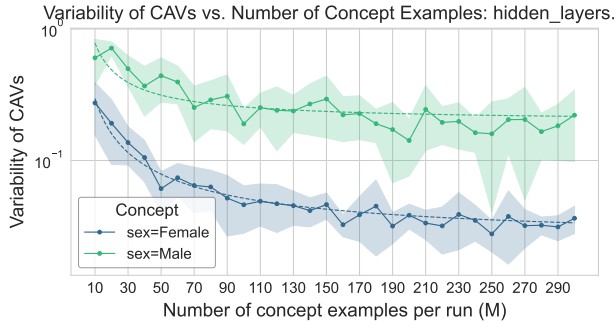

hidden_layers.0: CAV variability

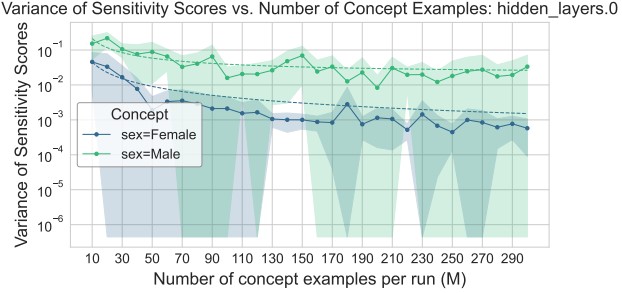

hidden_layers.0: Sensitivity variance

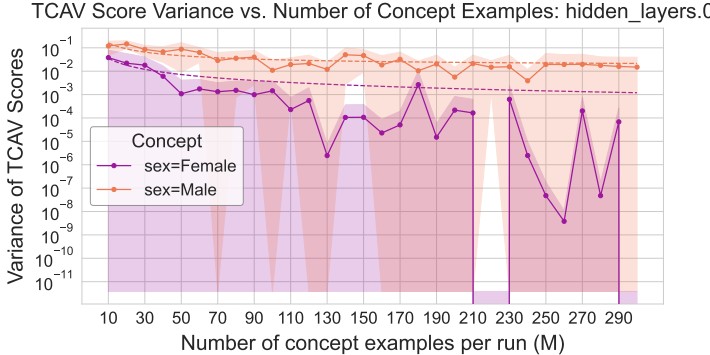

hidden_layers.0: TCAV score variance

*Figure 63.* **Concept-set-size ablation for tabular data at hidden_layers.0.** At the first hidden layer, the same qualitative trend appears. Larger concept sets stabilize the CAV and the sensitivity values, whereas the TCAV score can remain noisy because it aggregates signs rather than magnitudes.

### A.5.3. TEXT DATA

For text data, we use the IMDB sentiment setting. Concepts are the hand-curated adjective groups used throughout the paper: *positive-adjectives*, *negative-adjectives*, and *neutral-adjectives*. We report the same BERT encoder layers as in the text appendix, namely `bert_encoder_2` and `bert_encoder_4`.

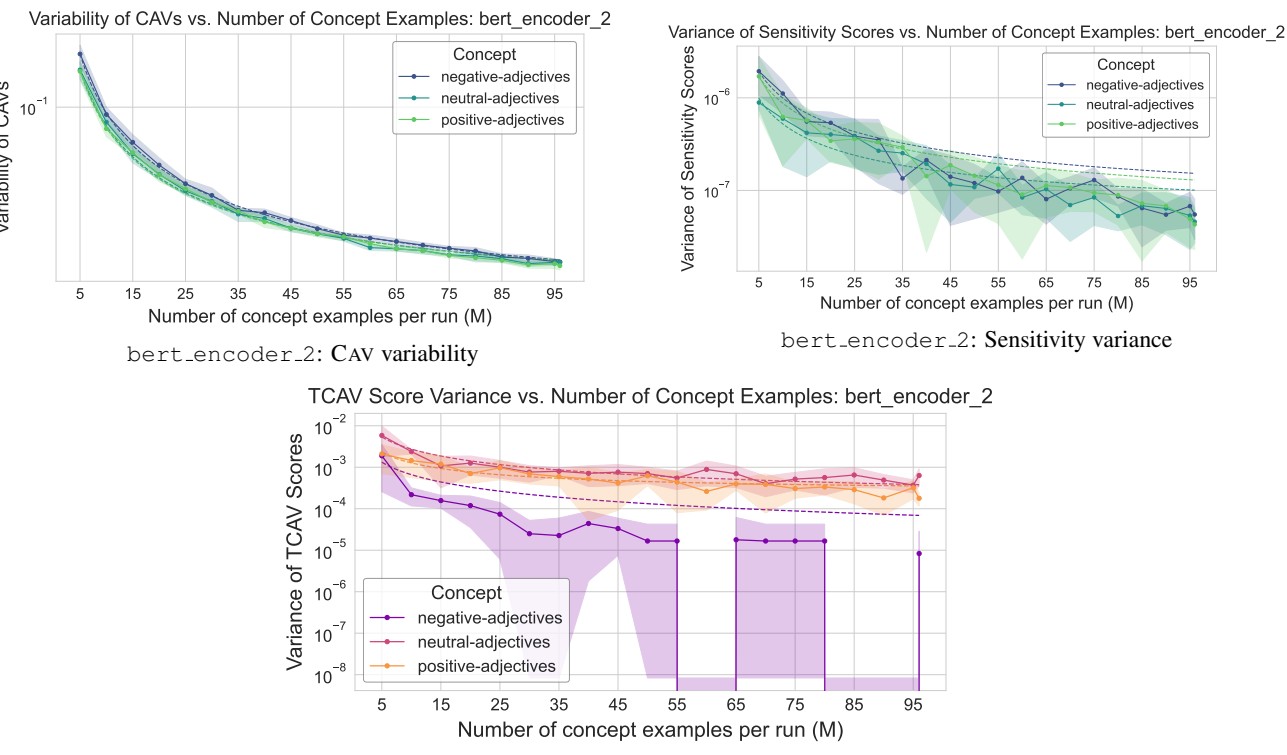

*Figure 64.* **Concept-set-size ablation for text data at `bert_encoder_2`.** At `bert_encoder_2`, increasing the number of adjective concept examples stabilizes the CAV direction. This supports the interpretation that concept-set sampling contributes to CAV variability in a way analogous to reference-set sampling.

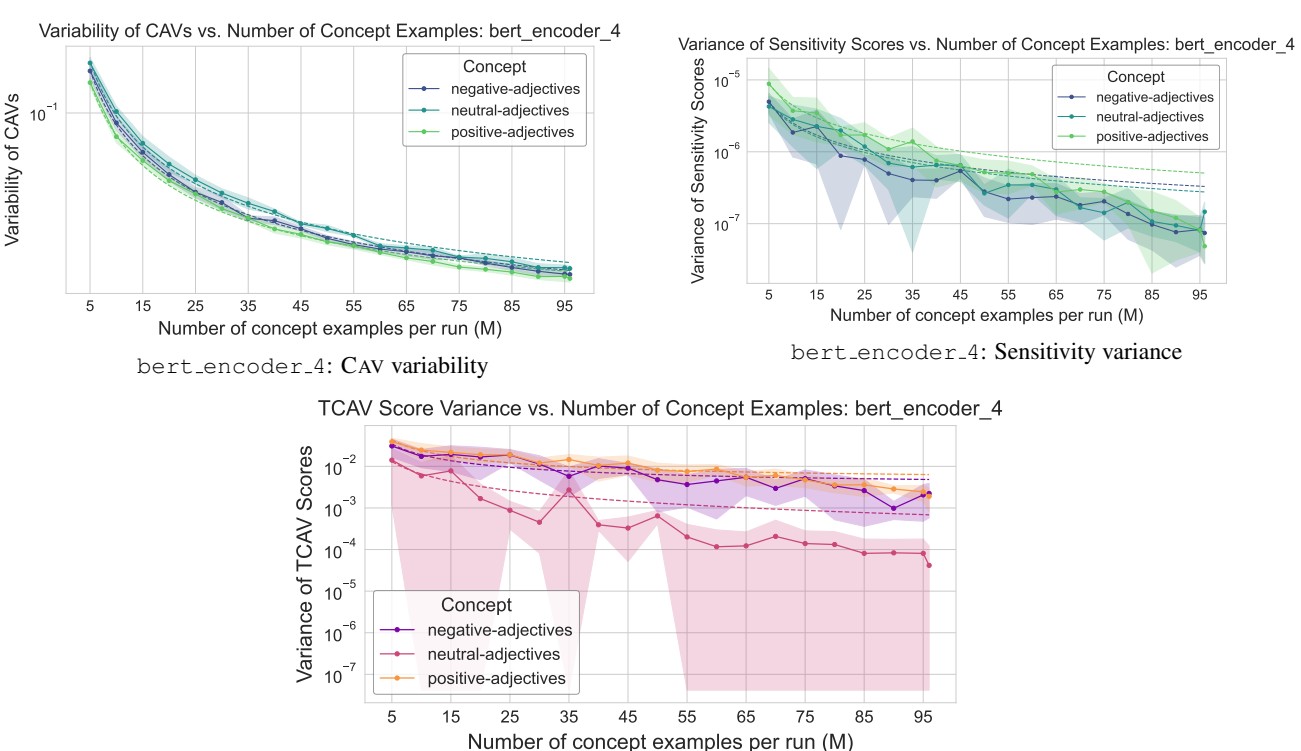

bert_encoder_4: CAV variability

bert_encoder_4: Sensitivity variance

bert_encoder_4: TCAV score variance

*Figure 65.* **Concept-set-size ablation for text data at bert_encoder_4.** At bert_encoder_4, the same qualitative pattern is visible. The layer-dependent constants differ, but increasing $M$ reduces the variability of the continuous quantities.

### A.5.4. MULTIMODAL DATA

Finally, we evaluate the effect of concept-set size in the CLIP-based multimodal setting. As in the main multimodal experiments, the concepts are image sets such as *striped*, *zigzagged*, and *dotted*, and the scalar output is the image–text similarity to the target prompt. We report the same CLIP vision hidden layers as in the multimodal appendix, namely `vision_hidden_4` and `vision_hidden_8`.

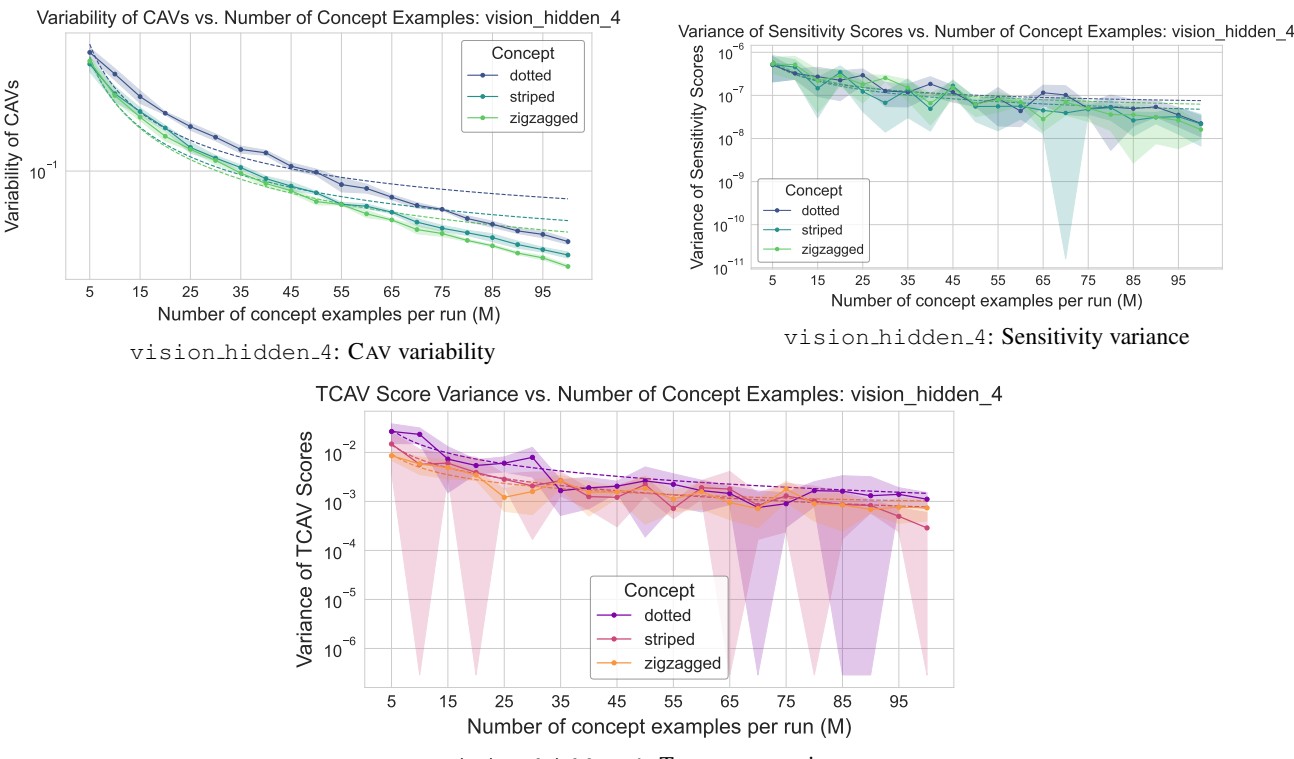

*Figure 66.* **Concept-set-size ablation for multimodal data at `vision_hidden_4`.** At CLIP `vision_hidden_4`, larger concept sets reduce CAV variability and sensitivity variance. This shows that the concept-set-size effect also appears in the vision–language setting.

**Summary.** These experiments complement the main fixed-concept-set analysis. Increasing the number $M$ of concept examples tends to reduce CAV variability and concept-sensitivity variance across image, tabular, text, and multimodal settings. The effect is layer- and modality-dependent through the constants, but the qualitative stabilization trend is robust. TCAV-score variance is again less predictable because it depends on the sign of the sensitivity, so near-borderline points can remain unstable even when the underlying CAV direction becomes more stable.

### A.6. Implementation Ablations

In this section, we investigate whether the empirical stability trends reported in the main paper are sensitive to common implementation choices. We focus on the tabular setting, where the model architecture and training choices can be varied in a controlled way. Unless stated otherwise, we follow the protocol of Section 5: for each number of random reference examples $N$, we fit multiple CAVs using independently sampled reference sets and measure the resulting CAV variability, concept-sensitivity variance, and TCAV-score variance.

Overall, the ablations confirm the qualitative conclusions of the main paper. Increasing $N$ stabilizes the CAV direction and the continuous concept-sensitivity scores across a range of implementation choices. These choices mainly affect the constants and the finite-sample regime, rather than changing the qualitative dependence on $N$. In contrast, the hard-thresholded TCAV score remains more sensitive to borderline points and can therefore decay more slowly or plateau.

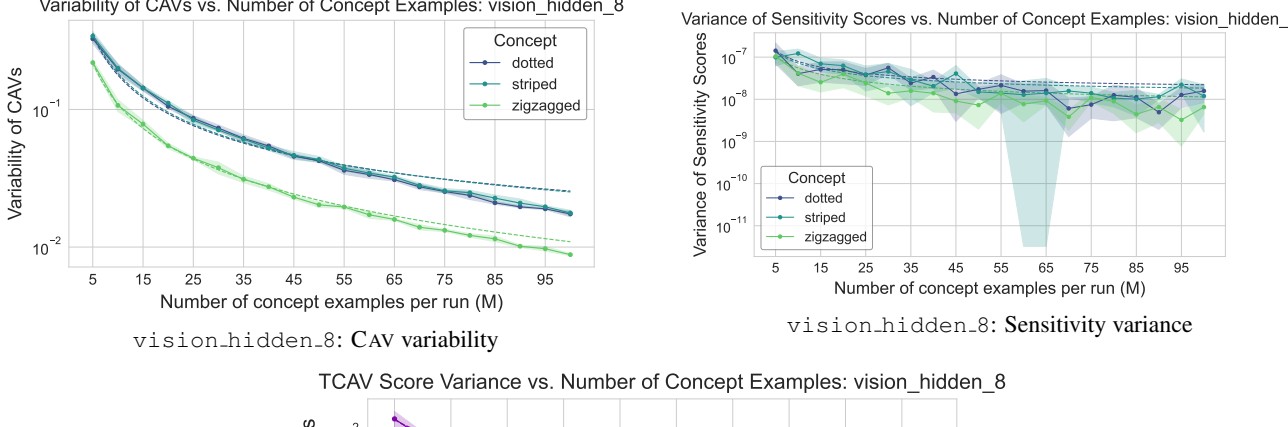

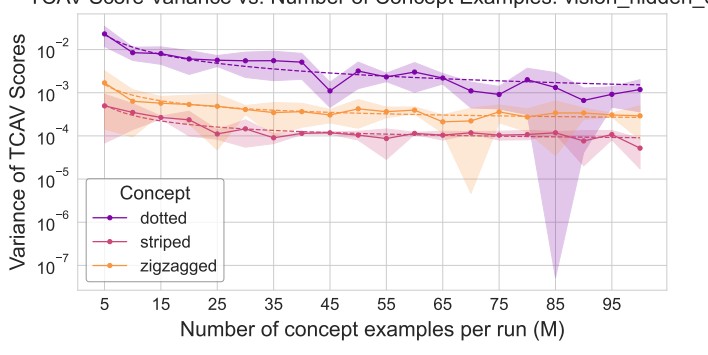

*Figure 67.* **Concept-set-size ablation for multimodal data at `vision_hidden_8`.** At CLIP `vision_hidden_8`, the qualitative stabilization trend remains visible. As in the other modalities, TCAV-score variance is less directly tied to CAV variance because of the hard threshold in the TCAV definition.

### A.6.1. LAYER ABLATION

We first vary the layer at which the CAV is computed. Figure 68 compares the input projection layer `fc_in` with two hidden layers, `hidden_layers.0` and `hidden_layers.2`. The CAV variability decreases with $N$ across all layers, but the constants differ substantially. This shows that the representation geometry of the chosen layer affects finite-sample stability.

The sensitivity variance also decreases with $N$, again with layer-dependent constants. The TCAV score variance is more layer-dependent than the CAV direction itself: even when the CAV direction stabilizes, hard-thresholded TCAV scores can remain noisy if the corresponding layer contains many borderline evaluation points.

### A.6.2. FEATURE NORMALIZATION

Next, we vary the feature normalization applied before fitting the CAV. Figure 69 compares raw activations, max-absolute normalization, and z-score normalization. All three choices preserve the qualitative trend that CAV variability decreases as $N$ grows. However, the magnitude of the variability changes across normalization schemes, which is expected because normalization rescales activation dimensions before the separating direction is fitted.

The sensitivity variance is especially affected by normalization. In this experiment, max-absolute normalization produces larger sensitivity variance than raw or z-score normalized activations. Thus, normalization should be treated as an implementation choice that affects constants and finite-sample stability, even though it does not invalidate the overall stabilization trend.

### A.6.3. REGULARIZATION STRENGTH

We also vary the regularization strength of the logistic-regression CAV estimator. In the scikit-learn convention, the parameter $C$ is the inverse regularization strength: smaller values of $C$ correspond to stronger regularization. Figure 70 compares $C = 0.25$, $C = 1$, and $C = 4$.

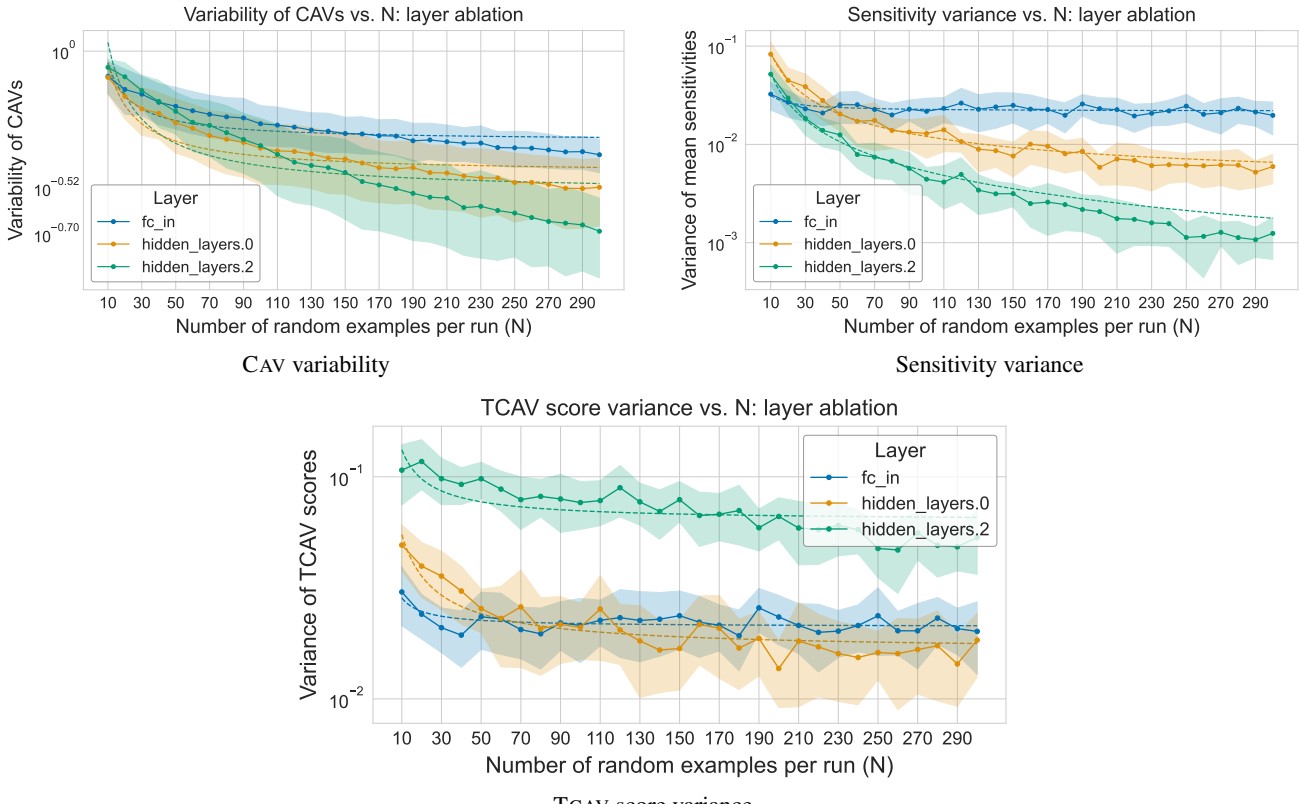

Figure 68. **Layer ablation.** We vary the layer used to compute CAVs in the tabular model. CAV variability decreases with $N$ across all layers, while the constants differ. TCAV score variance is more sensitive to the chosen layer because it depends on the signs of the concept sensitivities.

The results show that stronger regularization reduces CAV variability. This agrees with the role of the $\ell_2$ penalty in the theory: regularization restricts the effective parameter space and improves conditioning. The sensitivity variance follows the same qualitative stabilization trend. The TCAV score variance also benefits from stronger regularization in this experiment, although the hard sign threshold prevents the same clean behaviour observed for the CAV direction.

### A.6.4. CAV INITIALIZATION

Figure 71 studies the effect of the random initialization used when fitting the CAV estimator. Across the tested seeds, the curves nearly overlap for CAV variability, sensitivity variance, and TCAV score variance. This suggests that, in this setting, the dominant source of variability is the random reference set rather than the initialization of the optimizer.

This observation supports the interpretation of the main experiments: the variability analyzed in the paper is not merely an optimization artifact, but a genuine sampling effect caused by the random reference examples.

### A.6.5. OPTIMIZER

Finally, we compare different optimizers for fitting the CAV estimator: Adam, AdamW, and SGD. Figure 72 shows that all optimizers exhibit the same qualitative stabilization trend as $N$ increases. The constants differ, especially for sensitivity variance, but the dependence on $N$ remains qualitatively consistent.

**Summary.**  Across all ablations, the qualitative conclusions of the main paper remain stable. CAV variability and concept-sensitivity variance decrease as the number of random reference examples increases. Layer choice, feature normalization, regularization, and optimizer selection affect the constants, with normalization and regularization having especially visible effects in this tabular experiment. Initialization has little effect. These results support the view that the main source of CAV instability is the sampling of random reference examples, while implementation choices primarily modulate the magnitude

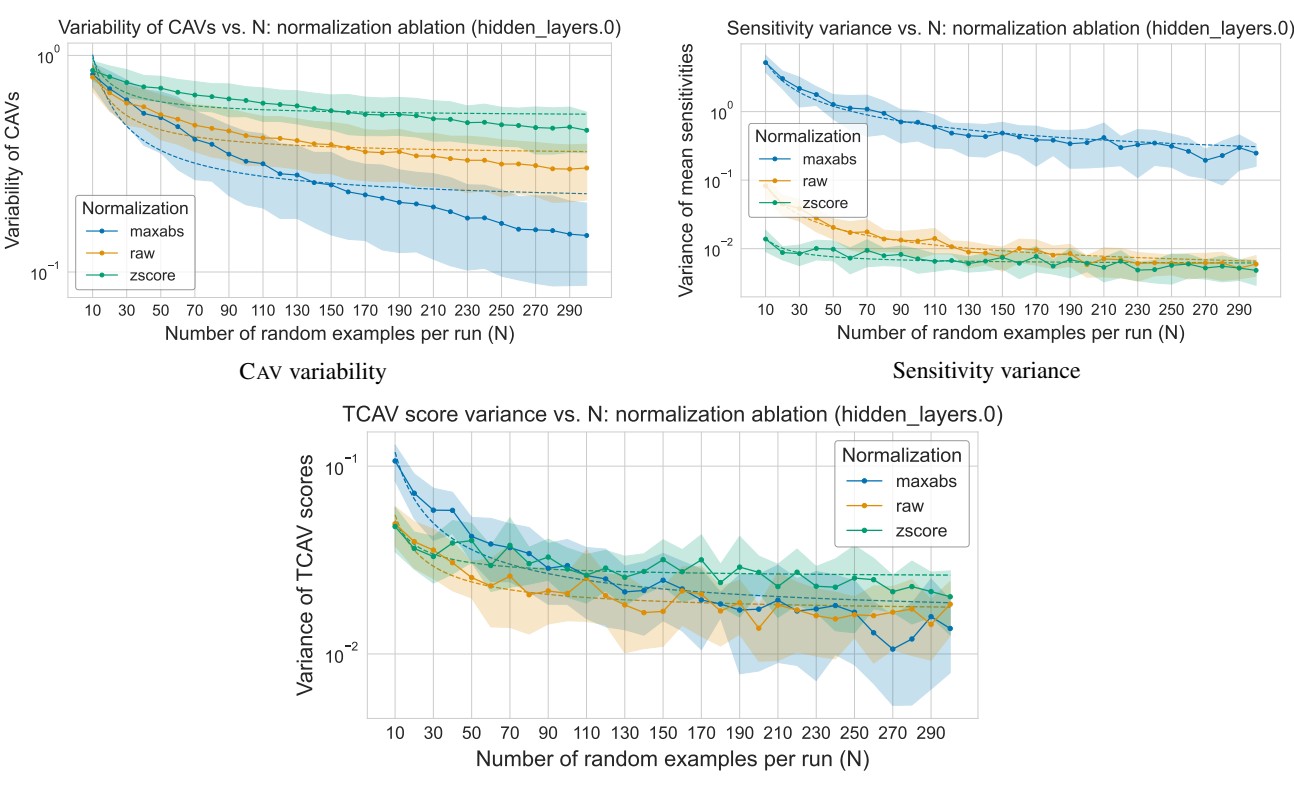

CAV variability

Sensitivity variance

TCAV score variance

*Figure 69.* **Normalization ablation.** We compare raw activations, max-absolute normalization, and z-score normalization at `hidden_layers.0`. The qualitative decrease of CAV variability with $N$ remains visible, but normalization changes the constants and can strongly affect sensitivity variance.

of the variance.

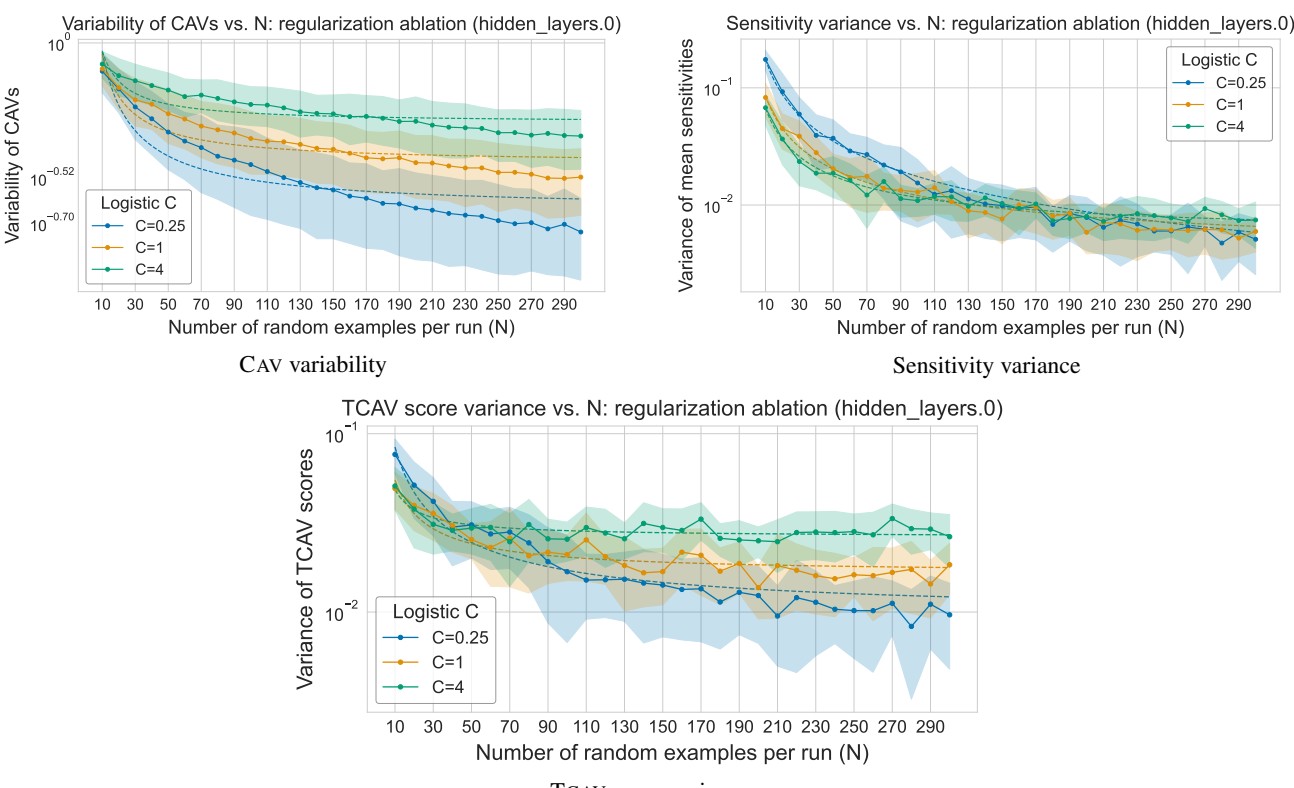

*Figure 70.* **Regularization ablation.** We vary the inverse regularization parameter $C$ of the logistic-regression CAV estimator. Smaller $C$ corresponds to stronger regularization. Stronger regularization lowers CAV variability and improves finite-sample stability, while the qualitative dependence on $N$ remains unchanged.

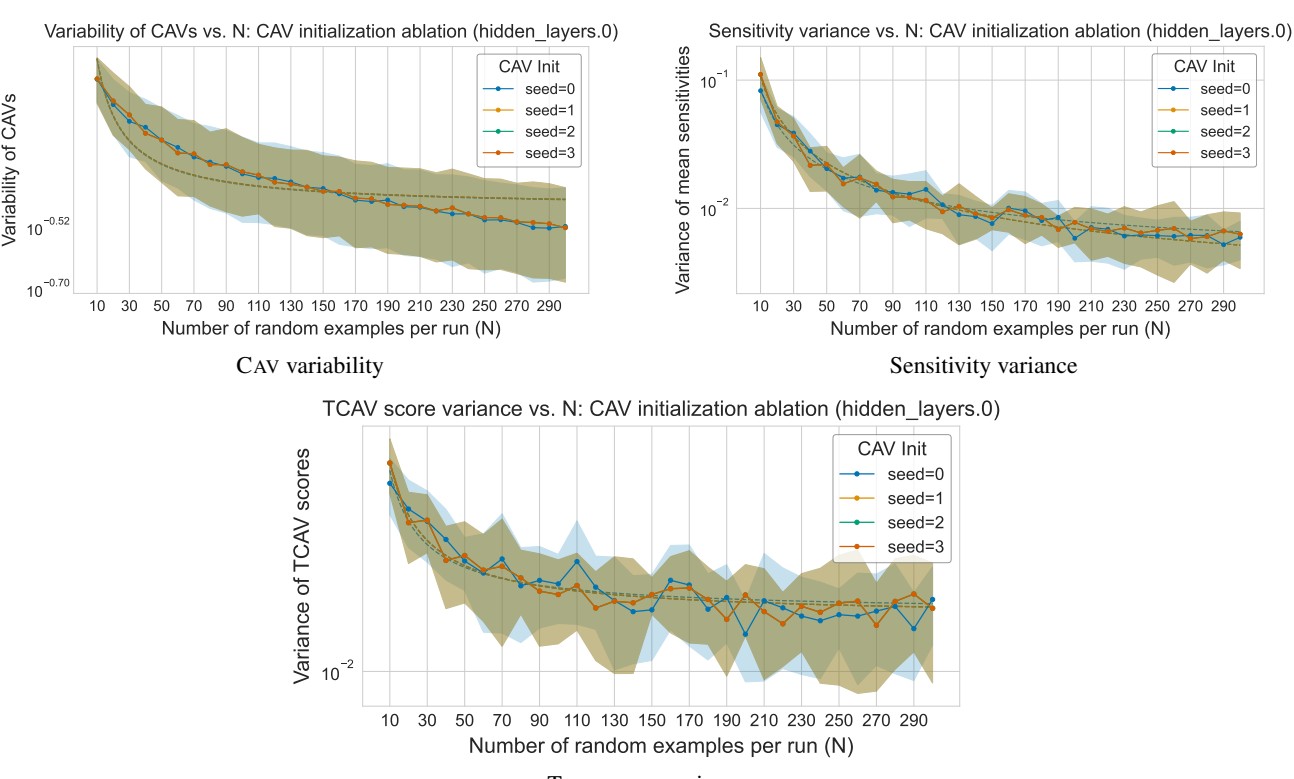

*Figure 71.* **CAV-initialization ablation.** We vary the random seed used for CAV initialization. The resulting curves nearly overlap, indicating that initialization has little effect compared to the variability induced by sampling random reference examples.

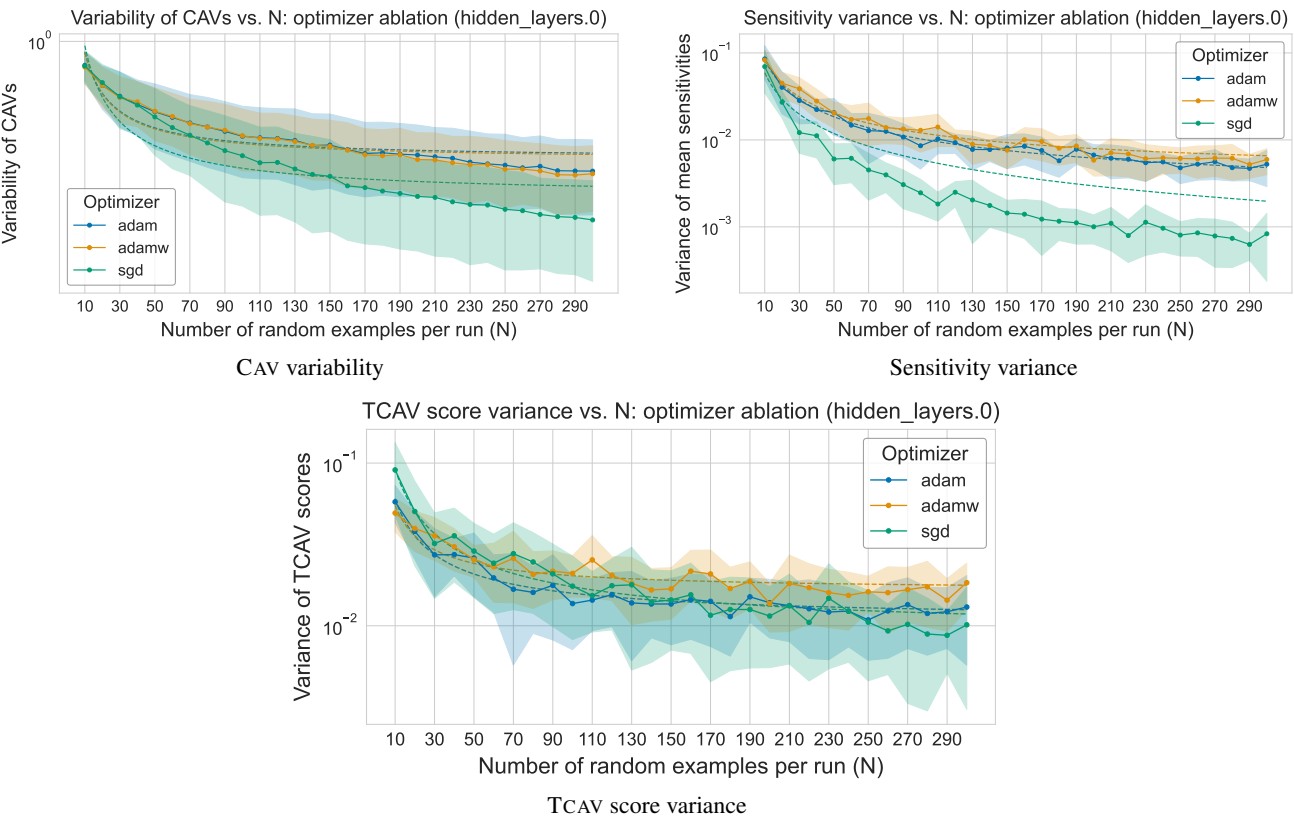

CAV variability

Sensitivity variance

TCAV score variance

*Figure 72.* **Optimizer ablation.** We compare Adam, AdamW, and SGD for fitting the CAV estimator. All optimizers show the same qualitative stabilization trend with increasing $N$. The optimizer mainly changes constants and finite-sample behaviour rather than the overall dependence on the number of random reference examples.

# B. Proofs of the Main Results

## B.1. Asymptotic Normality with Binary Cross-Entropy Loss

In Section 4.2, we established the asymptotic normality of CAV estimators trained with binary cross-entropy loss. This appendix provides the complete proofs.

**Main Idea and Proof Strategy.** Our goal is to show that CAV estimators are asymptotically normal. The key challenge is that we operate in an *infinitely imbalanced regime*: the number of negative (random) samples $N \to \infty$ while the number of concept samples $n$ remains fixed.

Our proof strategy, adapted from Owen (2007) and Goldman & Zhang (2022), proceeds as follows:

1. **Establish parameter bounds** (Section B.1.1): We first show that even though the intercept $\alpha_N$ diverges to $-\infty$, the quantity $A_N = Ne^{\alpha_N}$ remains bounded, and that the slope $\beta_N$ converges to a limit $\beta_\infty$.

2. **Prove asymptotic normality** (Section B.1.2): Using a Taylor expansion of the score around $\beta_\infty$, we show that:
   - The (rescaled) Hessian converges in probability to a positive definite limit $H_\infty$.
   - The (rescaled) score is asymptotically normal by the Central Limit Theorem.

   Slutsky's theorem then yields the asymptotic normality of $\sqrt{N}(\beta_N - \beta_\infty)$.

### B.1.1. SETUP AND PRELIMINARY RESULTS

**Notation.** For any given $N$, the default TCAV implementation fits the logistic regression model Eq. (3) by maximizing the $L^2$-penalized log-likelihood

$$
\begin{aligned}
\mathcal{L}_N^{(\lambda)}(\alpha, \beta) := &\sum_{i=1}^n \log \sigma(\alpha + \beta^\top x_i) \\
&+ \sum_{j=1}^N \log(1 - \sigma(\alpha + \beta^\top z_j)) - \frac{\lambda}{2} \|\beta\|^2 \,,
\end{aligned}
\tag{10}
$$

where $\alpha \in \mathbb{R}$, $\beta \in \mathbb{R}^d$ and $\lambda$ is a positive regularization parameter.

Recall that for any $N > 0$, we denote by $\alpha_N \in \mathbb{R}$ and $\beta_N \in \mathbb{R}^d$ the unique minimizers of the regularized loss (Eq. (10)). We also define $A_N := Ne^{\alpha_N}$.

We require four assumptions. The first ensures the concept samples are geometrically well-positioned relative to the random samples.

**Assumption 1 (Surrounded Mean).** The distribution $F_0$ on $\mathbb{R}^d$ has the point $\bar{x}$ "surrounded," that is

$$
\int_{(z-\bar{x})^\top \omega > \varepsilon} \mathrm{d}F_0(z) > \delta
\tag{11}
$$

holds for some $\varepsilon > 0$, some $\delta > 0$ and all $\omega \in \Omega$ where $\Omega = \{\omega \in \mathbb{R}^d | \omega^\top \omega = 1\}$ is the unit sphere in $\mathbb{R}^d$ and $\bar{x}$ is the mean of the concept embeddings $\{x_i\}_{i=1}^n$.

This assumption ensures that no hyperplane through $\bar{x}$ can separate the concept samples from the bulk of the random distribution. Intuitively, imagine you're standing at the mean point $\bar{x}$. The assumption guarantees that no matter which direction you look (any unit vector $\omega$), you'll find a meaningful amount of data "in front of you". Specifically, at least $\delta$ fraction of the data lies at distance greater than $\epsilon$ in that direction. Figure 73 provides a geometric visualization for the tabular setting.

The second and third assumptions concern parameter convergence.

**Assumption 2 (Intercept Scaling Limit).** The limit $A_\infty = \lim_{N \to \infty} Ne^{\alpha_N}$ exists.

**Assumption 3 (Consistency of $\beta_N$).** There exists a deterministic vector $\beta_\infty \in \mathbb{R}^d$ such that $\beta_N \xrightarrow{p} \beta_\infty$ as $N \to \infty$.

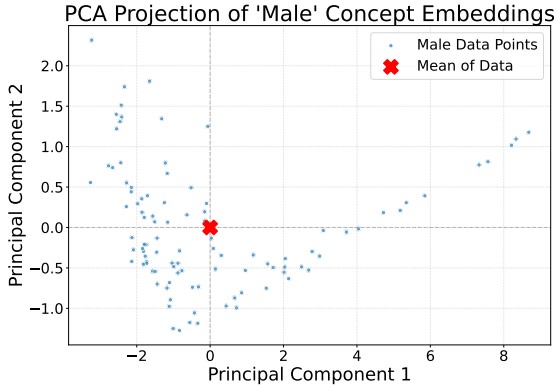 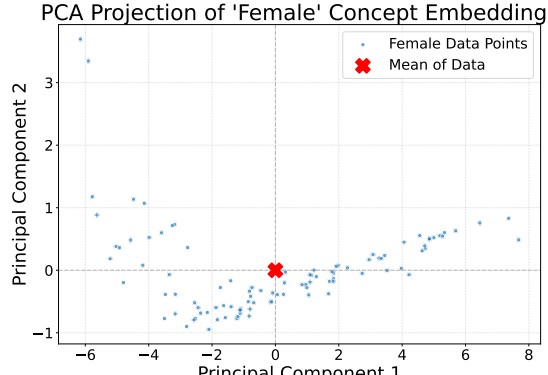

*Figure 73.* Geometric visualization of the surround assumption (Assumption 1) on the latent space. The 32-dimensional embeddings of the "Male" and "Female" concepts are projected onto their first two principal components. The data points are distributed on all sides of the sample mean (red 'X'), illustrating that they are not confined to a single half-space. The real empirical check is provided in the code.

Assumption 4 implies the integrability conditions needed in the Hessian and score analyses.

**Assumption 4 (Exponential moments under $F_0$).** For every $\beta \in \mathbb{R}^d$,

$$\mathbb{E}_{F_0}\left[\|Z\|^2 \exp(2\beta^\top Z)\right] < \infty. \tag{12}$$

We provide empirical verification of these assumptions in the accompanying notebooks under "*Checking Assumptions*."

**Lemma 1 (Consequences of Assumption 4).** *Assume Assumption 4. Then:*

1. *For every fixed $\beta \in \mathbb{R}^d$,*

$$\mathbb{E}\left[\|Z - \bar{x}\|^2 e^{2\beta^\top(Z - \bar{x})}\right] < \infty, \qquad \mathbb{E}\left[\|Z - \bar{x}\| e^{2\beta^\top(Z - \bar{x})}\right] < \infty.$$

2. *For every $B < \infty$,*

$$\mathbb{E}\left[\|Z - \bar{x}\|^2 e^{B\|Z - \bar{x}\|}\right] < \infty, \qquad \mathbb{E}\left[\|Z - \bar{x}\| e^{B\|Z - \bar{x}\|}\right] < \infty.$$

*Proof.* (1) Using $e^{2\beta^\top(Z - \bar{x})} = e^{-2\beta^\top \bar{x}} e^{2\beta^\top Z}$ and $\|Z - \bar{x}\|^2 \leq 2\|Z\|^2 + 2\|\bar{x}\|^2$ gives

$$\mathbb{E}\left[\|Z - \bar{x}\|^2 e^{2\beta^\top(Z - \bar{x})}\right] \leq C\left(\mathbb{E}\left[\|Z\|^2 e^{2\beta^\top Z}\right] + 1\right) < \infty$$

by Assumption 4. The first-moment bound follows since $\|u\| \leq 1 + \|u\|^2$ for all $u$.

(2) Let $u := Z - \bar{x}$. For any $u \in \mathbb{R}^d$,

$$e^{B\|u\|} \leq e^{B\|u\|_1} = \exp\left(B\sum_{k=1}^d |u_k|\right) = \prod_{k=1}^d e^{B|u_k|} \leq \prod_{k=1}^d \left(e^{Bu_k} + e^{-Bu_k}\right) = \sum_{s \in \{\pm 1\}^d} e^{Bs^\top u}.$$

Therefore,

$$\|u\|^2 e^{B\|u\|} \leq \sum_{s \in \{\pm 1\}^d} \|u\|^2 e^{Bs^\top u} = \sum_{s \in \{\pm 1\}^d} e^{-Bs^\top \bar{x}} \|Z - \bar{x}\|^2 e^{Bs^\top Z}.$$

Each summand has finite expectation by Assumption 4 applied with $\beta = \frac{B}{2}s$. The corresponding first-moment bound follows from $\|u\| \leq 1 + \|u\|^2$. $\square$

The following bound is an Owen-type control and implies $e^{\alpha_N} = O(N^{-1})$.

**Lemma 2 (Bounding $\alpha_N$; cf. Lemma 6 in (Owen, 2007)).** *Under Assumption 1, there exists $\eta > 0$ such that*

$$\inf_{\omega \in \Omega} \int \mathbf{1}\{(z - \bar{x})^\top \omega \geq 0\} \, \mathrm{d}F_0(z) \; \geq \; \eta. \tag{13}$$

*Then for any $N \geq 2n/\eta$,*

$$e^{\alpha_N} \leq \frac{2n}{N\eta}, \qquad equivalently \qquad A_N = Ne^{\alpha_N} \leq \frac{2n}{\eta}.$$

*Proof.* Assumption 1 implies (13) with $\eta := \delta$, since $\{(z - \bar{x})^\top \omega > \varepsilon\} \subseteq \{(z - \bar{x})^\top \omega \geq 0\}$. For fixed $\beta_N \in \mathbb{R}^d$, the derivative in $\alpha$ is

$$\nabla_\alpha \mathcal{L}_N^{(\lambda)}(\alpha, \beta_N) = n - \sum_{i=1}^n \frac{e^{\alpha + \beta_N^\top (x_i - \bar{x})}}{1 + e^{\alpha + \beta_N^\top (x_i - \bar{x})}} - \sum_{j=1}^N \frac{e^{\alpha + \beta_N^\top (z_j - \bar{x})}}{1 + e^{\alpha + \beta_N^\top (z_j - \bar{x})}}.$$

Writing $A := Ne^\alpha$ and applying the same bounding argument as in (Owen, 2007) yields

$$\nabla_\alpha \mathcal{L}_N^{(\lambda)}(\alpha_N, \beta_N) \leq n - \frac{A_N \eta}{1 + A_N/N}.$$

If $N \geq 2n/\eta$ but $A_N > 2n/\eta$, then

$$\frac{A_N \eta}{1 + A_N/N} > \frac{A_N \eta}{1 + (A_N \eta)/(2n)} > n,$$

so $\nabla_\alpha \mathcal{L}_N^{(\lambda)}(\alpha_N, \beta_N) < 0$. Concavity of $\alpha \mapsto \mathcal{L}_N^{(\lambda)}(\alpha, \beta_N)$ forces $A_N \leq 2n/\eta$, i.e., $e^{\alpha_N} \leq 2n/(N\eta)$. $\qquad\square$

This property is used to restrict the Hessian analysis to a compact parameter set for a uniform law of large numbers.

**Lemma 3 (Uniform bound on $\beta_N$; cf. Lemma 7 in (Owen, 2007)).** *Assume Assumption 1. Let $\eta > 0$ satisfy (13) and define*

$$\gamma := \inf_{\omega \in \Omega} \int \left[(z - \bar{x})^\top \omega\right]_+ \mathrm{d}F_0(z), \qquad [u]_+ := \max\{u, 0\}. \tag{14}$$

*Then $\gamma > 0$ and if $N \geq \max\{n, 2n/\eta\}$, then $\beta_N$ satisfies*

$$\|\beta_N\| \leq \frac{2}{\gamma}\left(1 + \frac{2}{\eta}\right) =: B. \tag{15}$$

*Consequently, for $\beta_t := \beta_\infty + t(\beta_N - \beta_\infty)$ with $t \in [0, 1]$,*

$$\sup_{t \in [0,1]} \|\beta_t\| \leq B.$$

*Proof.* First note that $\gamma > 0$ follows from Assumption 1: since $\{(z - \bar{x})^\top \omega > \varepsilon\} \subseteq \{(z - \bar{x})^\top \omega > 0\}$ and $(z - \bar{x})^\top \omega \geq \varepsilon$ on that set,

$$\int \left[(z - \bar{x})^\top \omega\right]_+ \mathrm{d}F_0(z) \; \geq \; \int \varepsilon \mathbf{1}\{(z - \bar{x})^\top \omega > \varepsilon\} \, \mathrm{d}F_0(z) \; \geq \; \varepsilon\delta,$$

uniformly over $\omega \in \Omega$, hence $\gamma \geq \varepsilon\delta > 0$. The presence of the $L^2$ penalty makes the result of Lemma 7 in (Owen, 2007) even more stringent ($-\frac{\lambda}{2}\|\beta\|^2$ can only decrease the objective away from $\beta = 0$), thus the same comparison argument applies to the penalized maximizer and we get

$$\|\beta_N\| \leq \frac{1}{\gamma}\left(1 + \frac{A_N}{N}\right)\left(1 + \frac{n}{N}\right). \tag{16}$$

Lemma 2 gives $A_N \leq 2n/\eta$. By assumption, $N \geq n$ and we obtain Eq. (16). The interpolation-path bound follows from $\|\beta_t\| \leq \max\{\|\beta_\infty\|, \|\beta_N\|\}$. $\qquad\square$

The next lemma gives the limiting mean of $Y = (Z - \bar{x})e^{\beta_\infty^\top (Z - \bar{x})}$ in terms of the penalty.

**Lemma 4 (Penalized mean identity;cf. Theorem 8 in (Owen, 2007)).** *Assume Assumptions 2, 3, and 4. Then*

$$\mathbb{E}_{F_0}\left[(Z-\bar{x})\,e^{\beta_\infty^\top(Z-\bar{x})}\right] = -\frac{\lambda}{A_\infty}\,\beta_\infty. \tag{17}$$

*Equivalently,*

$$\int (z-\bar{x})\,e^{z^\top\beta_\infty}\,\mathrm{d}F_0(z) = -\frac{\lambda\,e^{\bar{x}^\top\beta_\infty}}{A_\infty}\,\beta_\infty.$$

*Proof.* At the maximizer $(\alpha_N, \beta_N)$, the $\beta$-score vanishes:

$$\nabla_\beta \mathcal{L}_N^{(\lambda)}(\alpha_N, \beta_N) = 0.$$

Differentiating (10) yields

$$\sum_{i=1}^n (1-\sigma(t_i))(x_i - \bar{x}) - \sum_{j=1}^N \sigma(s_j)(z_j - \bar{x}) - \lambda\beta_N = 0, \tag{18}$$

where $t_i = \alpha_N + \beta_N^\top(x_i - \bar{x})$ and $s_j = \alpha_N + \beta_N^\top(z_j - \bar{x})$.

Using $\sum_{i=1}^n (x_i - \bar{x}) = 0$, the concept term is

$$\sum_{i=1}^n (1-\sigma(t_i))(x_i - \bar{x}) = -\sum_{i=1}^n \sigma(t_i)(x_i - \bar{x}),$$

and since $e^{\alpha_N} = A_N/N \to 0$ (Assumption 2) and $n$ is fixed, this term is $o_p(1)$.

For the random term, write

$$\sigma(s_j) = \frac{e^{\alpha_N + \beta_N^\top(z_j - \bar{x})}}{1 + e^{\alpha_N + \beta_N^\top(z_j - \bar{x})}} = \frac{(A_N/N)e^{\beta_N^\top(z_j - \bar{x})}}{1 + (A_N/N)e^{\beta_N^\top(z_j - \bar{x})}}.$$

Hence

$$\sum_{j=1}^N \sigma(s_j)(z_j - \bar{x}) = \frac{A_N}{N}\sum_{j=1}^N (z_j - \bar{x})e^{\beta_N^\top(z_j - \bar{x})} - \frac{A_N^2}{N^2}\sum_{j=1}^N (z_j - \bar{x})\frac{e^{2\beta_N^\top(z_j - \bar{x})}}{1 + (A_N/N)e^{\beta_N^\top(z_j - \bar{x})}}.$$

The second term is $o_p(1)$ because $A_N^2/N \to 0$ (Assumption 2) and, on the event $\{\|\beta_N\| \leq B\}$ (whose probability tends to 1 by Lemma 3),

$$\|(z-\bar{x})\|e^{2\beta_N^\top(z-\bar{x})} \leq \|z - \bar{x}\|e^{2B\|z-\bar{x}\|},$$

whose expectation is finite by Lemma 1(2). Thus the empirical average is $O_p(1)$ and the prefactor $A_N^2/N \to 0$ forces the product to be $o_p(1)$.

Therefore (18) becomes

$$\frac{A_N}{N}\sum_{j=1}^N (z_j - \bar{x})e^{\beta_N^\top(z_j - \bar{x})} + \lambda\beta_N = o_p(1).$$

Divide by $A_N$ and use $A_N \to A_\infty$ and $\beta_N \to \beta_\infty$ to obtain

$$\frac{1}{N}\sum_{j=1}^N (z_j - \bar{x})e^{\beta_N^\top(z_j - \bar{x})} \xrightarrow{p} -\frac{\lambda}{A_\infty}\beta_\infty.$$

By the law of large numbers and continuity in $\beta$ (justified by dominated convergence using Lemma 1(2)), the left-hand side converges in probability to $\mathbb{E}[(Z - \bar{x})e^{\beta_\infty^\top(Z-\bar{x})}]$, yielding (17). $\qquad\square$

### B.1.2. PROOF OF ASYMPTOTIC NORMALITY

Having established the preliminary bounds and limiting relationships, we now prove our main result on asymptotic normality. Our approach, adapted from Goldman & Zhang (2022), uses a Taylor expansion of the score equation to extract the limiting dynamics.

**Theorem 1 (Asymptotic Normality of CAV s).** *Assume Assumptions 1- 4. Then*

$$\sqrt{N}\,(\beta_N - \beta_\infty) \xrightarrow{D} \mathcal{N}(0, \Sigma), \tag{19}$$

*where*

$$\Sigma := H_\infty^{-1} \Sigma' (H_\infty^{-1})^\top, \qquad \Sigma' := \mathrm{Var}_{F_0}(Y_1), \qquad Y_1 := (Z - \bar{x})\, e^{\beta_\infty^\top (Z - \bar{x})}, \tag{20}$$

*and*

$$H_\infty := \mathbb{E}_{F_0}\left[ e^{\beta_\infty^\top (Z - \bar{x})} (Z - \bar{x})(Z - \bar{x})^\top \right] + \frac{\lambda}{A_\infty} I. \tag{21}$$

*Proof.* Define $\mathcal{F}(\beta) := \nabla_\beta \mathcal{L}_N^{(\lambda)}(\alpha_N, \beta)$. Since $(\alpha_N, \beta_N)$ maximizes $\mathcal{L}_N^{(\lambda)}$, we have $\mathcal{F}(\beta_N) = 0$. A first-order Taylor expansion around $\beta_\infty$ gives

$$0 = \mathcal{F}(\beta_N) = \nabla_\beta \mathcal{L}_N^{(\lambda)}(\alpha_N, \beta_\infty) + H_N(\beta_N - \beta_\infty), \tag{22}$$

where

$$H_N := \int_0^1 \nabla_\beta^2 \mathcal{L}_N^{(\lambda)}(\alpha_N, \beta_t)\, \mathrm{d}t, \qquad \beta_t := \beta_\infty + t(\beta_N - \beta_\infty).$$

Multiplying (22) by $e^{-\alpha_N}/\sqrt{N}$ and rearranging yields

$$\sqrt{N}(\beta_N - \beta_\infty) = \left( -\frac{e^{-\alpha_N}}{N} H_N \right)^{-1} \left( \frac{e^{-\alpha_N}}{\sqrt{N}} \nabla_\beta \mathcal{L}_N^{(\lambda)}(\alpha_N, \beta_\infty) \right). \tag{23}$$

We analyze the two factors on the right-hand side.

**Step 1: Hessian term.** The Hessian of $\mathcal{L}_N^{(\lambda)}$ with respect to $\beta$ is

$$\nabla_\beta^2 \mathcal{L}_N^{(\lambda)}(\alpha, \beta) = -\sum_{i=1}^n \sigma_i(1 - \sigma_i)(x_i - \bar{x})(x_i - \bar{x})^\top - \sum_{j=1}^N \sigma_j(1 - \sigma_j)(z_j - \bar{x})(z_j - \bar{x})^\top - \lambda I,$$

where $\sigma_i = \sigma(\alpha + \beta^\top(x_i - \bar{x}))$ and $\sigma_j = \sigma(\alpha + \beta^\top(z_j - \bar{x}))$.

Scale by $-e^{-\alpha_N}/N = -1/A_N$. The concept terms vanish since $n$ is fixed and $\sigma(\alpha_N + \beta_t^\top(x_i - \bar{x}))(1 - \sigma(\cdot)) = O(e^{\alpha_N})$, hence their contribution is $O(1/N)$. The penalty contributes

$$-\frac{e^{-\alpha_N}}{N}(-\lambda I) = \frac{\lambda}{A_N} I \rightarrow \frac{\lambda}{A_\infty} I.$$

For the random terms, use $\sigma(t)(1 - \sigma(t)) = \frac{e^t}{(1 + e^t)^2}$ to write

$$-\frac{e^{-\alpha_N}}{N} \int_0^1 \sum_{j=1}^N \sigma_{j,t}(1 - \sigma_{j,t})(z_j - \bar{x})(z_j - \bar{x})^\top \mathrm{d}t = \frac{1}{N} \sum_{j=1}^N \int_0^1 \frac{e^{\beta_t^\top(z_j - \bar{x})}}{(1 + e^{\alpha_N} e^{\beta_t^\top(z_j - \bar{x})})^2}(z_j - \bar{x})(z_j - \bar{x})^\top \mathrm{d}t.$$

By Lemma 3, for all $N \geq N_0$ we have $\sup_{t \in [0,1]} \|\beta_t\| \leq B$. Define $\Theta := [0,1] \times \{\beta : \|\beta\| \leq B\}$. For $(a, \beta) \in [0,1] \times \{\|\beta\| \leq B\}$, the integrand is dominated by

$$G_B(z) := \|z - \bar{x}\|^2 e^{B\|z - \bar{x}\|},$$

since $(1 + ae^u)^{-2} \leq 1$ and $e^{\beta^\top(z - \bar{x})} \leq e^{B\|z - \bar{x}\|}$. By Lemma 1(2), $\mathbb{E}[G_B(Z)] < \infty$. Therefore, a uniform law of large numbers for dominated parametric classes (Newey & McFadden, 1994) yields

$$-\frac{e^{-\alpha_N}}{N} H_N \xrightarrow{p} \mathbb{E}\left[ e^{\beta_\infty^\top(Z - \bar{x})}(Z - \bar{x})(Z - \bar{x})^\top \right] + \frac{\lambda}{A_\infty} I = H_\infty.$$

Moreover, $H_\infty$ is invertible automatically: for any nonzero $v \in \mathbb{R}^d$,

$$v^\top H_\infty v = \mathbb{E}\left[e^{\beta_\infty^\top(Z-\bar{x})}(v^\top(Z-\bar{x}))^2\right] + \frac{\lambda}{A_\infty}\|v\|^2 \geq \frac{\lambda}{A_\infty}\|v\|^2 > 0,$$

since $\lambda > 0$ and $A_\infty \in (0, \infty)$.

**Step 2: Score term.** Define the rescaled score

$$S_N := \frac{e^{-\alpha_N}}{\sqrt{N}}\nabla_\beta \mathcal{L}_N^{(\lambda)}(\alpha_N, \beta_\infty) = \frac{e^{-\alpha_N}}{\sqrt{N}}\nabla_\beta \mathcal{L}_N(\alpha_N, \beta_\infty) - \frac{\lambda e^{-\alpha_N}}{\sqrt{N}}\beta_\infty.$$

Using $\sum_{i=1}^n(x_i - \bar{x}) = 0$, the concept contribution to $\nabla_\beta \mathcal{L}_N(\alpha_N, \beta_\infty)$ equals $-\sum_{i=1}^n \sigma(\alpha_N + \beta_\infty^\top(x_i - \bar{x}))(x_i - \bar{x})$, which is $O(e^{\alpha_N})$ and thus becomes $O(N^{-1/2})$ after multiplying by $e^{-\alpha_N}/\sqrt{N} = N/(A_N\sqrt{N})$.

For the random term, use the identity

$$\sigma(t) = \frac{e^t}{1+e^t} = e^t - \frac{e^{2t}}{1+e^t},$$

with $t_j := \alpha_N + \beta_\infty^\top(z_j - \bar{x})$. Then

$$e^{-\alpha_N}\sigma(t_j) = e^{\beta_\infty^\top(z_j-\bar{x})} - e^{\alpha_N}\frac{e^{2\beta_\infty^\top(z_j-\bar{x})}}{1+e^{t_j}}.$$

Hence

$$\frac{e^{-\alpha_N}}{\sqrt{N}}\nabla_\beta \mathcal{L}_N(\alpha_N, \beta_\infty) = -\frac{1}{\sqrt{N}}\sum_{j=1}^N Y_j - R_N, \qquad Y_j := (z_j - \bar{x})e^{\beta_\infty^\top(z_j-\bar{x})},$$

where

$$\|R_N\| \leq \frac{e^{\alpha_N}}{\sqrt{N}}\sum_{j=1}^N \|z_j - \bar{x}\|e^{2\beta_\infty^\top(z_j-\bar{x})} = \frac{A_N}{\sqrt{N}} \cdot \frac{1}{N}\sum_{j=1}^N \|z_j - \bar{x}\|e^{2\beta_\infty^\top(z_j-\bar{x})}.$$

Since $A_N \to A_\infty$, we have $A_N/\sqrt{N} \to 0$. By Lemma 1(1), the empirical average is $O_p(1)$, hence $R_N = o_p(1)$.

Let $m := \mathbb{E}[Y_1]$. Then

$$-\frac{1}{\sqrt{N}}\sum_{j=1}^N Y_j = -\frac{1}{\sqrt{N}}\sum_{j=1}^N(Y_j - m) - \sqrt{N}\, m.$$

By Lemma 4, $m = -\frac{\lambda}{A_\infty}\beta_\infty$, so the deterministic part of $S_N$ is

$$-\sqrt{N}\, m - \frac{\lambda e^{-\alpha_N}}{\sqrt{N}}\beta_\infty = \frac{\lambda\sqrt{N}}{A_\infty}\beta_\infty - \frac{\lambda\sqrt{N}}{A_N}\beta_\infty \xrightarrow{p} 0,$$

because $A_N \to A_\infty$ and $e^{-\alpha_N} = N/A_N$. Consequently,

$$S_N = -\frac{1}{\sqrt{N}}\sum_{j=1}^N(Y_j - m) + o_p(1) \xrightarrow{D} \mathcal{N}(0, \Sigma'), \qquad \Sigma' = \text{Var}_{F_0}(Y_1),$$

by the multivariate CLT, using $\mathbb{E}\|Y_1\|^2 < \infty$ from Lemma 1(1).

**Conclusion.** Combining Step 1 and Step 2 in (23) and applying Slutsky's theorem (Slutsky, 1937) yields

$$\sqrt{N}(\beta_N - \beta_\infty) \xrightarrow{D} \mathcal{N}\left(0, H_\infty^{-1}\Sigma'(H_\infty^{-1})^\top\right),$$

which is (19) with $\Sigma$ given by (20). $\qquad\square$

B.1.3. CONVERGENCE RATE OF THE COVARIANCE TRACE

Finally, we establish the convergence rate of the covariance trace.

**Definition 2 (Uniform Integrability).** A sequence of random variables $\{X_N\}_{N \geq 1}$ is uniformly integrable if

$$\lim_{K \to \infty} \sup_{N \geq 1} \mathbb{E}\left[|X_N| \cdot \mathbf{1}_{\{|X_N| > K\}}\right] = 0.$$

**Corollary 3 (Asymptotic Behavior of the Covariance Trace).** *Under the assumptions of Theorem 1 and uniform integrability of $\{\|\sqrt{N}(\beta_N - \beta_\infty)\|^2\}_{N \geq 1}$, the covariance matrix satisfies $\mathrm{Var}(\beta_N) = \frac{1}{N}\Sigma + o(N^{-1})$. Consequently, $\mathrm{tr}(\mathrm{Var}(\beta_N)) = \mathcal{O}(N^{-1})$.*

*Proof.* From Theorem 1, $\sqrt{N}(\beta_N - \beta_\infty) \xrightarrow{D} \mathcal{N}(0, \Sigma)$. Uniform integrability ensures moment convergence (van der Vaart, 1998):

$$\lim_{N \to \infty} \mathrm{Cov}(\sqrt{N}(\beta_N - \beta_\infty)) = \Sigma.$$

Since $\mathrm{Cov}(\sqrt{N}(\beta_N - \beta_\infty)) = N \cdot \mathrm{Cov}(\beta_N)$, we have $\lim_{N \to \infty} N \cdot \mathrm{Cov}(\beta_N) = \Sigma$. By continuity of the trace operator:

$$\mathrm{tr}(\Sigma) = \lim_{N \to \infty} N \cdot \mathrm{tr}(\mathrm{Var}(\beta_N)),$$

which implies $\mathrm{tr}(\mathrm{Var}(\beta_N)) = \mathcal{O}(N^{-1})$. $\qquad\square$

B.1.4. ASYMPTOTIC NORMALITY OF CONCEPT SENSITIVITY SCORES

We proof the asymptotic normality of concept sensitivity scores applying a special case of the multivariate delta method (with $\varphi(\beta) = a(\mathbf{x})^\top \beta$). Note, that none of our proofs in this section depend on *how* $\beta_N$ is constructed: logistic regression, hinge loss, difference of means, kernel support vector machines, etc.. The only input is (24). Therefore, this holds for all methods analyzed in the remaining sections of the Appendix.

**Lemma 5 (Linear maps preserve asymptotic normality).** *Let $(\beta_N)_{N \geq 1}$ be a sequence of $\mathbb{R}^d$-valued random vectors and let $\beta_\infty \in \mathbb{R}^d$, $\Sigma \in \mathbb{R}^{d \times d}$ be deterministic with $\Sigma$ symmetric positive semidefinite. Assume*

$$\sqrt{N}\left(\beta_N - \beta_\infty\right) \xrightarrow{D} \mathcal{N}(0, \Sigma). \tag{24}$$

*Then for any fixed matrix $A \in \mathbb{R}^{m \times d}$,*

$$\sqrt{N}\left(A\beta_N - A\beta_\infty\right) \xrightarrow{D} \mathcal{N}\left(0, A\Sigma A^\top\right). \tag{25}$$

*Proof.* Define $\Delta_N := \sqrt{N}(\beta_N - \beta_\infty)$. By (24), $\Delta_N \xrightarrow{D} Z$ with $Z \sim \mathcal{N}(0, \Sigma)$.

Consider the map $T : \mathbb{R}^d \to \mathbb{R}^m$ given by $T(u) = Au$. This map is linear and hence continuous. By the continuous mapping theorem,

$$A\Delta_N = T(\Delta_N) \xrightarrow{D} T(Z) = AZ.$$

Finally, since $Z$ is multivariate normal, $AZ$ is multivariate normal with mean $A\,\mathbb{E}[Z] = 0$ and covariance

$$\mathrm{Cov}(AZ) = A\,\mathrm{Cov}(Z)\,A^\top = A\Sigma A^\top.$$

This proves (25). $\qquad\square$

**Corollary 4 (Asymptotic normality of concept sensitivity).** *Assume (24). Fix an input $\mathbf{x}$ and define*

$$a(\mathbf{x}) := \nabla g_{\ell,k}\left(h_\ell(\mathbf{x})\right) \in \mathbb{R}^d, \qquad S(\mathbf{x}, \beta) := a(\mathbf{x})^\top \beta.$$

*Then*

$$\sqrt{N}\left(S(\mathbf{x}, \beta_N) - S(\mathbf{x}, \beta_\infty)\right) \xrightarrow{D} \mathcal{N}\left(0, V(\mathbf{x})\right), \tag{26}$$

*where the asymptotic variance is*

$$V(\mathbf{x}) := a(\mathbf{x})^\top \Sigma\, a(\mathbf{x}). \tag{27}$$

*Proof.* We have the identity

$$\sqrt{N}\Big(S(\mathbf{x}, \beta_N) - S(\mathbf{x}, \beta_\infty)\Big) = a(\mathbf{x})^\top \sqrt{N}(\beta_N - \beta_\infty).$$

Apply Lemma 5 with $m = 1$ and $A = a(\mathbf{x})^\top \in \mathbb{R}^{1 \times d}$. Then (26) holds with covariance $A\Sigma A^\top = a(\mathbf{x})^\top \Sigma a(\mathbf{x})$, which is exactly (27). □

## B.2. Asymptotic Normality with Hinge Loss

By default, the TCAV implementations in both TensorFlow and PyTorch use `sklearn`'s `SGDClassifier` (Pedregosa et al., 2011), which operates by minimizing the hinge loss function. It achieves this by penalizing predictions that are not only incorrect but also those that are correct but fall within a specified "margin" around the decision boundary.

For a dataset of $N + n$ samples $\{(u_i, y_i)\}_{i=1}^{N+n}$ with true labels $y_i \in \{-1, 1\}$, the final loss $J_N^{(\lambda)}(\beta, \alpha)$ minimized is defined as the sum of the average hinge loss and an $L^2$ regularization term:

$$J_N^{(\lambda)}(\beta, \alpha) = \left(\frac{1}{N+n} \sum_{i=1}^{N+n} \max[0, 1 - y_i(\beta^\top u_i + \alpha)]\right) + \frac{\lambda}{2}\|\beta\|^2.$$

Here, $\lambda$ represents the regularization strength hyperparameter, $\beta$ is the weight vector, and $\alpha$ is the intercept term. Again we consider our known setting. The only difference is that we use a different labeling system. The first class consists of $n$ fixed points, $\{x_i\}_{i=1}^n \subset \mathbb{R}^d$, referred to as "concept" samples with label $y = 1$. The second class consists of $N$ random points, $\{z_j\}_{j=1}^N \subset \mathbb{R}^d$, drawn independently and identically from a distribution $F_0$ with label $y = -1$.

For our main theorem of asymptotic normality to hold, we require the following assumptions:

1. The distribution $F_0$ of the controls must be continuous. Specifically, the projection $\beta_\infty^\top Z$ must have a continuous probability density function, $f_{\beta_\infty^\top Z}(\cdot)$, in a neighbourhood of $-1$.

2. The distribution $F_0$ must have finite second moments, *i.e.*, $\mathbb{E}_{Z \sim F_0}[\|Z\|^2] < \infty$.

Given these assumptions we can now state the theorem.

**Theorem 2 (Asymptotic Normality with Hinge Loss).** *Let $\beta_N$ be the minimizer of the objective function $J_N(\beta)$. Let $\beta_\infty$ be the unique minimizer of the limiting objective function*

$$\lim_{N\to\infty} J_N^{(\lambda)}(\beta, \alpha) = \mathbb{E}_{Z \sim F_0}[\max(0, 1 + \beta^T Z + \alpha)] + \frac{\lambda}{2}\|\beta\|^2. \tag{28}$$

*Under the assumptions listed above in Section B.2, as $N \to \infty$ with $n$ fixed, the estimator is asymptotically normal:*

$$\sqrt{N}(\beta_N - \beta_\infty) \xrightarrow{D} \mathcal{N}(0, M^{-1}\Sigma_Z M^{-1}), \tag{29}$$

*where $M = \lambda I + \mathbb{E}[ZZ^\top | \beta_\infty^\top Z = -1] f_{\beta_\infty^\top Z}(-1)$, and $\Sigma_Z = Var\left(Z \cdot I(\beta_\infty^\top Z > -1)\right)$.*

*Proof.* As $N \to \infty$, the objective function $J_N(\beta)$ converges pointwise in probability to $\lim_{N\to\infty} J_N^{(\lambda)}(\beta)$. Under standard M-estimation arguments, the minimizer $\beta_N$ of $J_N(\beta)$ converges in probability to the minimizer $\beta_\infty$ of $\lim_{N\to\infty} J_N^{(\lambda)}(\beta)$.

We first perform a Taylor expansion of the gradient of the objective function around $\beta_\infty$, and set $\nabla J_N(\beta_N) = 0$:

$$0 = \nabla_\beta J_N(\beta_\infty) + \nabla_\beta^2 J_N(\beta_\infty)(\beta_N - \beta_\infty) + o_p(1). \tag{30}$$

Rearranging gives

$$\sqrt{N}(\beta_N - \beta_\infty) = -[\nabla_\beta^2 J_N(\beta_\infty)]^{-1}\sqrt{N}\nabla_\beta J_N(\beta_\infty) + o_p(1).$$

We analyze the two terms on the right. The optimality of $\beta_\infty$ for the limiting problem implies $\lim_{N\to\infty} \nabla_\beta J^{(\lambda)}(\beta) = 0$. This gives the condition $\mathbb{E}[Z \cdot I(\beta_\infty^T Z > -1)] + \lambda\beta_\infty = 0$. Let now $\mu_0 = \mathbb{E}[Z \cdot I(\beta_\infty^\top Z > -1)]$. The gradient of the

finite-sample objective at $\beta_\infty$ is

$$\nabla_\beta J_N(\beta_\infty) = \frac{N}{n+N}\left(\frac{1}{N}\sum_{j=1}^{N} z_j I(\beta_\infty^\top z_j > -1) - \mu_0\right)$$
$$- \frac{n}{n+N}\left(\frac{1}{n}\sum_{i=1}^{n} x_i I(\beta_\infty^\top x_i < 1) + \mu_0\right).$$

When scaled by $\sqrt{N}$, the second term vanishes as $N \to \infty$. By the Central Limit Theorem and our assumption on $\mathbb{E}_{Z \sim F_0}[\|Z\|^2] < \infty$, the first term converges in distribution. Thus, the scaled score has a normal limit

$$\sqrt{N}\nabla_\beta J_N(\beta_\infty) \xrightarrow{D} \mathcal{N}(0, \Sigma_Z) \tag{31}$$

where $\Sigma_Z = \text{Var}\left(Z \cdot I(\beta_\infty^\top Z > -1)\right)$. The Hessian matrix of the finite-sample objective, $H_N(\beta) = \nabla_\beta^2 J_N(\beta)$, converges in probability to the Hessian of the limiting objective

$$H := \lim_{N \to \infty} \nabla_\beta^2 J_N(\beta_\infty) = \lambda I + \mathbb{E}[ZZ^\top | \beta_\infty^\top Z = -1] f_{\beta_\infty^\top Z}(-1). \tag{32}$$

Slutsky's Theorem (Slutzky, 1937) and the assumed probability density function, $f_{\beta_\infty^\top Z}(\cdot)$ give us the asymptotic distribution of the estimator

$$\sqrt{N}(\beta_N - \beta_\infty) \xrightarrow{D} \mathcal{N}(0, H^{-1}\Sigma_Z H^{-1}).$$

This completes the proof. □

## B.3. Asymptotic Normality of Difference of Means

The Difference of Means (DoM) (Martin, 2019) method identifies a concept's direction within a model's activation space by simply taking the vector difference between the average activation for concept examples and the average activation for random examples. As this method was also adapted for calculating CAVs (De Santis et al., 2024), we analyze the stability of the direction vector $\beta_N = \bar{x} - \bar{z}$, where $\bar{x}$ is the mean of the $n$ fixed concept samples and $\bar{z}$ is the mean of $N$ random samples (from an independent distribution $F_0$ with covariance $\Sigma_z$).

**Theorem 3 (Asymptotic Variance of the Difference of Means Vector).** *Let $\{x_i\}_{i=1}^n$ be $n$ fixed points, and let $\{z_j\}_{j=1}^N$ be $N$ random samples drawn i.i.d. from an independent distribution $F_0$ with finite covariance $\Sigma_z$. Define the direction vector $\beta_N = \bar{x} - \bar{z}$ as the difference of the respective means.*

*Then, the total variance of $\beta_N$, is given by:*

$$\text{tr}(\text{Cov}(\beta_N)) = \frac{1}{N}\text{tr}(\Sigma_z) \tag{33}$$

*Proof.* The variance of $\beta_N$, measured by $\text{tr}(\text{Cov}(\beta_N))$, is derived as follows. Since the set $\{x_i\}_{i=1}^n$ is fixed, their mean $\bar{x}$ is a deterministic constant vector, and thus $\text{Cov}(\bar{x}) = 0$. The variance of $\beta_N$ is therefore determined exclusively by the random component $\bar{z}$.

$$\text{tr}(\text{Cov}(\beta_N)) = \text{tr}(\text{Cov}(\bar{x} - \bar{z})) \tag{34}$$

$$= \text{tr}\left(\frac{1}{N^2}\sum_{j=1}^{N}\text{Cov}(z_j)\right) \quad \text{(by i.i.d. assumption)} \tag{35}$$

$$= \text{tr}\left(\frac{1}{N^2}(N \cdot \Sigma_z)\right) \tag{36}$$

$$= \frac{1}{N}\text{tr}(\Sigma_z) \tag{37}$$

Since $\mathrm{tr}(\Sigma_z)$ is a fixed constant, the variance of $\beta_N$ declines at a rate of $\mathcal{O}(1/N)$ as the number of random samples $N$ increases.

$\square$

As the number of random samples $N \to \infty$ (while $n$ remains fixed), the variance of $\beta_N$ converges to zero at a rate of $\mathcal{O}(1/N)$. The *Difference of Means*-method thus exhibits the same convergence behaviour as the other two classifiers discussed.

### B.4. Asymptotic Normality of Concept Activation Regions

Concept Activation Regions extend the linear SVM classifier described in Appendix B.2 to nonlinear settings through the use of positive-definite kernels. In this section, we first introduce the necessary background on reproducing kernel Hilbert spaces, then describe the kernel SVM formulation used by CARs, and finally establish asymptotic concentration of the CAR estimator.

Let $\kappa : H \times H \to \mathbb{R}$ be a positive-definite kernel. By the Moore-Aronszajn theorem, there exists a unique reproducing kernel Hilbert space (RKHS) $\mathcal{H}_\kappa$ of functions $f : H \to \mathbb{R}$ satisfying the reproducing property $f(h) = \langle f, \kappa(\cdot, h) \rangle_{\mathcal{H}_\kappa}$. Equivalently, there exists a feature map $\Phi : H \to \mathcal{H}_\kappa$ such that

$$\kappa(h, h') = \langle \Phi(h), \Phi(h') \rangle_{\mathcal{H}_\kappa}.$$

We assume throughout that the kernel is bounded: $\sup_{h \in H} \kappa(h, h) \leq K^2$. For background on RKHS theory, see Steinwart & Christmann (2008) or Schölkopf & Smola (2002).

As before, let $\{h_\ell(x_i)\}_{i=1}^n$ denote the positive-class embeddings and $\{h_\ell(z_j)\}_{j=1}^N$ the negative-class embeddings. We define a labeled training set of size $m = n + N$ by

$$(s_r, y_r) = \begin{cases} \big(h_\ell(x_r), +1\big), & r = 1, \ldots, n, \\ \big(h_\ell(z_{r-n}), -1\big), & r = n+1, \ldots, n+N. \end{cases}$$

Crabbé & van der Schaar (2022) train a soft-margin SVM on this labeled set. The standard constrained formulation seeks $\mathbf{w} \in \mathcal{H}_\kappa$, bias $b \in \mathbb{R}$, and slack variables $\xi \in \mathbb{R}^m$ solving

$$\min_{\mathbf{w}, b, \xi} \quad \frac{1}{2} \|\mathbf{w}\|_{\mathcal{H}_\kappa}^2 + C \sum_{r=1}^m \xi_r \qquad \text{subject to} \quad y_r \big( \langle \mathbf{w}, \Phi(s_r) \rangle_{\mathcal{H}_\kappa} + b \big) \geq 1 - \xi_r, \quad \xi_r \geq 0.$$

Equivalently (up to rescaling of the regularization parameter), one may write the unconstrained regularized hinge-loss form:

$$\min_{\mathbf{w} \in \mathcal{H}_\kappa, b \in \mathbb{R}} \quad \frac{\lambda}{2} \|\mathbf{w}\|_{\mathcal{H}_\kappa}^2 + \frac{1}{m} \sum_{r=1}^m \max\big(0, \, 1 - y_r(\langle \mathbf{w}, \Phi(s_r) \rangle_{\mathcal{H}_\kappa} + b)\big).$$

**Dual formulation.** This optimization is solved via its dual. Introducing Lagrange multipliers $\{\alpha_r\}_{r=1}^m$ and applying the KKT conditions yields the representer expansion

$$\mathbf{w} = \sum_{r=1}^m \alpha_r y_r \Phi(s_r) \ \in \ \mathcal{H}_\kappa.$$

The dual problem includes box constraints $0 \leq \alpha_r \leq C$ and, when the bias $b$ is included, the equality constraint $\sum_{r=1}^m \alpha_r y_r = 0$. Substituting the expansion into the decision function gives the kernel classifier:

$$f(h) = \langle \mathbf{w}, \Phi(h) \rangle_{\mathcal{H}_\kappa} + b = \sum_{r=1}^m \alpha_r y_r \kappa(s_r, h) + b, \qquad \hat{y}(h) = \mathrm{sgn}\big(f(h)\big).$$

Only points with $\alpha_r \neq 0$ contribute; these are the *support vectors*. The same mechanism extends to non-linear decision boundaries by choosing a non-linear feature map $\Phi$ (implicitly) via a kernel $\kappa(h, h') = \langle \Phi(h), \Phi(h') \rangle$ as done by Crabbé & van der Schaar (2022).

For concept activation methods we seek a *direction* in representation space that captures the concept—analogous to the CAV vector $\beta_N \in \mathbb{R}^d$.

We therefore define the CAR vector as the bias-free normal vector of the separating hyperplane:

**Definition 3 (Concept Activation Region (CAR)).** Given the labeled training set $\{(s_r, y_r)\}_{r=1}^m$ constructed from positive and negative concept embeddings, the CAR *vector* is

$$\beta_N := \mathbf{w} = \sum_{r=1}^m \alpha_r y_r \Phi(s_r) \in \mathcal{H}_\kappa,$$

where $\{\alpha_r\}$ are the dual SVM coefficients. The associated *bias-free score* is

$$f_N(h) := \langle \beta_N, \Phi(h) \rangle_{\mathcal{H}_\kappa} = \sum_{r=1}^m \alpha_r y_r \kappa(s_r, h).$$

Setting the bias $b = 0$ yields a direction-like object directly comparable to a CAV, at the cost of discarding the optimal decision threshold.

Since $\beta_N \in \mathcal{H}_\kappa$, we extend Definition 4 to the CAR setting:

**Definition 4 (Variance of CARs).** The variance of the CAR vector $\beta_N$ is

$$\text{Var}(\beta_N) := \mathbb{E}\big\|\beta_N - \mathbb{E}[\beta_N]\big\|_{\mathcal{H}_\kappa}^2 = \text{tr}\big(\text{Cov}(\beta_N)\big).$$

This is the natural analogue of the total variance used for CAVs. When $\kappa$ is the linear kernel, we have $\mathcal{H}_\kappa \simeq \mathbb{R}^d$ and $\|\cdot\|_{\mathcal{H}_\kappa} = \|\cdot\|_2$, so Definition 4 reduces exactly to Definition 1.

Our goal is to show that the variance $\text{Var}(\beta_N)$ decreases as the number of negative samples $N$ grows. To obtain this bound, we analyze how $\beta_N$ depends on the random negative samples $\{z_j\}_{j=1}^N$. The key technical tool is the Efron-Stein inequality, which bounds the variance of a function of independent random variables in terms of the sensitivity to each input.

**Lemma 6 (Efron-Stein inequality for Hilbert spaces (Efron & Stein, 1981)).** *Let $f : \mathcal{X}^N \to \mathcal{H}$ be measurable, $W = f(X_1, \ldots, X_N) \in \mathcal{H}$, and assume $\mathbb{E}[\|W\|^2] < \infty$. For each $j$, let $X^{(j)}$ be the sequence where $X_j$ is replaced by an independent copy, and set $W^{(j)} = f(X^{(j)})$. Then*

$$\text{Var}(W) = \mathbb{E}\big\|W - \mathbb{E}W\big\|_{\mathcal{H}}^2 \leq \frac{1}{2}\sum_{j=1}^N \mathbb{E}\big\|W - W^{(j)}\big\|_{\mathcal{H}}^2.$$

Using this Lemma from Efron & Stein (1981) we can now state the following theorem.

**Theorem 4 (Variance of class-weighted kernel CARs).** *Let $\{h_\ell(x_i)\}_{i=1}^n \subset H$ be fixed positive embeddings and let $\{h_\ell(z_j)\}_{j=1}^N$ be i.i.d. from $F_0$. Assume $\sup_{h \in H} \kappa(h, h) \leq K^2$, and let $\beta_N \in \mathcal{H}_\kappa$ be trained by the class-weighted, $\lambda$-regularized hinge-loss objective. Then*

$$\text{Var}(\beta_N) \leq \frac{2K^2}{\lambda^2} \cdot \frac{1}{N} \qquad \textit{and hence} \qquad \text{Var}(\beta_N) = \mathcal{O}(1/N) \textit{ as } N \to \infty.$$

*Proof.* Let $Z = (z_1, \ldots, z_N)$ denote the negative sample and let $Z^{(j)}$ be the same sample except that $z_j$ is replaced by an independent copy $z_j' \sim F_0$. Write $\beta_N = \beta(Z)$ for the CAR vector trained on $Z$, and $\beta_N^{(j)} = \beta(Z^{(j)})$ for the CAR vector trained on the perturbed sample $Z^{(j)}$.

These are the minimizers of the respective empirical objectives

$$J_Z(\beta) = \frac{\lambda}{2}\|\beta\|_{\mathcal{H}_\kappa}^2 + \frac{1}{n}\sum_{i=1}^n \ell\big(\langle\beta, \Phi(h_\ell(x_i))\rangle_{\mathcal{H}_\kappa}\big) + \frac{1}{N}\sum_{j=1}^N \ell\big(-\langle\beta, \Phi(h_\ell(z_j))\rangle_{\mathcal{H}_\kappa}\big),$$

where $\ell(u) = \max(0, 1 - u)$ is the hinge loss.

Because of the quadratic regularizer, $J_Z(\beta)$ is $\lambda$-strongly convex in $\beta$. Recall that $\beta_N = \beta(Z)$ minimizes $J_Z$ and $\beta_N^{(j)} = \beta(Z^{(j)})$ minimizes $J_{Z^{(j)}}$. By the strong convexity inequality applied at each respective minimizer, we have

$$J_Z(\beta_N^{(j)}) - J_Z(\beta_N) \geq \frac{\lambda}{2}\|\beta_N^{(j)} - \beta_N\|_{\mathcal{H}_\kappa}^2, \qquad J_{Z^{(j)}}(\beta_N) - J_{Z^{(j)}}(\beta_N^{(j)}) \geq \frac{\lambda}{2}\|\beta_N^{(j)} - \beta_N\|_{\mathcal{H}_\kappa}^2.$$

Adding these two inequalities yields

$$\lambda\|\beta_N^{(j)} - \beta_N\|_{\mathcal{H}_\kappa}^2 \leq \left(J_Z(\beta_N^{(j)}) - J_{Z^{(j)}}(\beta_N^{(j)})\right) + \left(J_{Z^{(j)}}(\beta_N) - J_Z(\beta_N)\right).$$

The objectives $J_Z$ and $J_{Z^{(j)}}$ differ only in the $j$-th negative-class loss term: $J_Z$ includes $\ell\left(-\langle\beta, \Phi(h_\ell(z_j))\rangle\right)$ while $J_{Z^{(j)}}$ includes $\ell\left(-\langle\beta, \Phi(h_\ell(z_j'))\rangle\right)$, each scaled by $1/N$. Therefore

$$\lambda\|\beta_N^{(j)} - \beta_N\|_{\mathcal{H}_\kappa}^2 \leq \frac{1}{N}\left(\ell_{z_j}(\beta_N^{(j)}) - \ell_{z_j}(\beta_N)\right) + \frac{1}{N}\left(\ell_{z_j'}(\beta_N) - \ell_{z_j'}(\beta_N^{(j)})\right),$$

where we write $\ell_u(\beta) := \ell\left(-\langle\beta, \Phi(h_\ell(u))\rangle_{\mathcal{H}_\kappa}\right)$ for the hinge loss evaluated at embedding $h_\ell(u)$.

The hinge loss $\ell(t) = \max(0, 1 - t)$ is 1-Lipschitz in its scalar argument. Moreover, for any $u$,

$$\left|\langle\beta, \Phi(h_\ell(u))\rangle - \langle\beta', \Phi(h_\ell(u))\rangle\right| \leq \|\beta - \beta'\|_{\mathcal{H}_\kappa}\|\Phi(h_\ell(u))\|_{\mathcal{H}_\kappa} = \|\beta - \beta'\|_{\mathcal{H}_\kappa}\sqrt{\kappa(h_\ell(u), h_\ell(u))} \leq K\|\beta - \beta'\|_{\mathcal{H}_\kappa}.$$

Hence $\ell_u(\cdot)$ is $K$-Lipschitz in $\beta$. Applying this to both terms on the right-hand side gives

$$\lambda\|\beta_N^{(j)} - \beta_N\|_{\mathcal{H}_\kappa}^2 \leq \frac{2K}{N}\|\beta_N^{(j)} - \beta_N\|_{\mathcal{H}_\kappa}.$$

If $\|\beta_N^{(j)} - \beta_N\|_{\mathcal{H}_\kappa} > 0$, dividing both sides yields

$$\|\beta_N^{(j)} - \beta_N\|_{\mathcal{H}_\kappa} \leq \frac{2K}{\lambda N}.$$

The inequality holds trivially when $\|\beta_N^{(j)} - \beta_N\|_{\mathcal{H}_\kappa} = 0$.

Apply Lemma 6 with $W = \beta_N$:

$$\mathrm{Var}(\beta_N) = \mathbb{E}\|\beta - \mathbb{E}[\beta]\|_{\mathcal{H}_\kappa}^2 \leq \frac{1}{2}\sum_{j=1}^{N}\mathbb{E}\|\beta - \beta^{(j)}\|_{\mathcal{H}_\kappa}^2.$$

Using the Step 1 bound,

$$\mathrm{Var}(\beta_N) \leq \frac{1}{2}\sum_{j=1}^{N}\left(\frac{2K}{\lambda N}\right)^2 = \frac{1}{2} \cdot N \cdot \frac{4K^2}{\lambda^2 N^2} = \frac{2K^2}{\lambda^2} \cdot \frac{1}{N}.$$

$\square$

**Lemma 7 (Variance bound at a fixed point).** *For any fixed $h \in H$,*

$$\mathrm{Var}(f_N(h)) = \mathrm{Var}(\langle\beta_N, \Phi(h)\rangle) \leq \kappa(h, h)\,\mathrm{Var}(\beta_N). \tag{38}$$

*Proof.* Let $\bar\beta := \mathbb{E}[\beta_N]$. Then $f_N(h) - \mathbb{E}[f_N(h)] = \langle\beta_N - \bar\beta, \Phi(h)\rangle$. Hence $\mathrm{Var}(f_N(h)) = \mathbb{E}\left[\langle\beta_N - \bar\beta, \Phi(h)\rangle^2\right]$.

By Cauchy–Schwarz,

$$\langle\beta_N - \bar\beta, \Phi(h)\rangle^2 \leq \|\beta_N - \bar\beta\|_{\mathcal{H}_\kappa}^2\|\Phi(h)\|_{\mathcal{H}_\kappa}^2.$$

Taking expectations and using $\|\Phi(h)\|^2 = \kappa(h, h)$ yields

$$\mathrm{Var}(f_N(h)) \leq \mathbb{E}\|\beta_N - \bar\beta\|_{\mathcal{H}_\kappa}^2\,\kappa(h, h) = \mathrm{Var}(\beta_N)\,\kappa(h, h).$$

$\square$

For the RBF kernel, $\kappa(h, h) = 1$ for all $h$, so (38) simplifies to

$$\mathrm{Var}\big(f_N(h)\big) \leq \mathrm{Var}(\beta_N).$$

**Corollary 5 (Variance of kernel CAR decision values).** *For any $h \in H$, the decision-value variance decays as $\mathcal{O}(1/N)$:*

$$\mathrm{Var}\big(f_N(h)\big) \leq \kappa(h, h) \, \mathrm{Var}(\beta_N) \leq K^2 \cdot \frac{2K^2}{\lambda^2} \cdot \frac{1}{N} = \frac{2K^4}{\lambda^2} \cdot \frac{1}{N}.$$

**Finite-Dimensional Approximation via Nyström** The CAR vector While products between CAR vectors ($\beta$) can be computed via the dual form, but this requires storing all support vectors from each resampled CAR and $\mathcal{O}(R^2 \cdot n_{\mathrm{sv}}^2)$ kernel evaluations for variance estimation across $R$ runs.

The Nyström method (Williams & Seeger, 2000) approximates the kernel via a low-rank factorization using $q$ landmark points, which gives an explicit feature map $\tilde{\Phi} : H \to \mathbb{R}^q$ with $\tilde{\Phi}(h)^\top \tilde{\Phi}(h') \approx \kappa(h, h')$. Training a linear SVM on transformed features produces $\tilde{\beta}_N \in \mathbb{R}^q$, reducing variance computation to $\mathcal{O}(R \cdot q^2)$.

The guarantees of Theorem 4 transfer to this approximation: the objective remains $\lambda$-strongly convex on the restricted subspace, so the Efron-Stein argument yields identical $\mathcal{O}(1/N)$ variance bounds. Rudi et al. (2015) show that

$$q = \mathcal{O}(\sqrt{N}/\lambda) \tag{39}$$

landmarks suffice for approximation error to become negligible relative to statistical fluctuations. We use $q = 200$ landmarks with a shared basis across runs, which satisfies Eq. (39).

