# OpenReview forum: "On the Variability of Concept Activation Vectors"
_ICML.cc/2026/Conference — ICML 2026 regular_

### Official Review · Reviewer_xBVR · 2026-03-08

**Soundness:** 3
**Presentation:** 2
**Significance:** 3
**Originality:** 3
**Overall Recommendation:** 4
**Confidence:** 2

**Summary:**

This paper focuses on the key question of how many random samples are sufficient to achieve stable CAVs or TCAV scores, as these values would be different if randomly sampling the images. The authors prove that in the infinitely imbalanced regime, the penalized logistic regression CAVs are asymptotically normal. And it is proven that there is a subset that lies on or near the decision boundary, which leads to the sensitivity of variance. Based on these analyses, the authors give insightful recommendations on how to adopt models.

**Compliance With Llm Reviewing Policy:**

Affirmed.

**Final Justification:**

This rebuttal address my concerns clearly, expecially clarifies the connection between the finite set up in their experiments by providing related cites; identify the concept of borderline samples; and also provides supplementary experimental results of the up-to-date models. With these clarifications, I find the work to be both well-motivated and rigorous, with solid originality and practical significance. Although the rebuttal fully addressed my concerns, my original score already reflected a relatively positive assessment of the work. Therefore, I will maintain my score.

**Key Questions For Authors:**

please refer to the weakness.

**Limitations:**

yes

**Strengths And Weaknesses:**

Strengths:
1. This is the first paper to analyze the variability of CAVs, which give the different perspective from the prior works.
2. The experiment is comprehansively arrange on four real-world dataset incluing the image, text, tabular and multimodal, which significantly enhances the universality of the conclusions.

Weakness:
1. The authors prove that penalized logistic regression CAVs are asymptotically normal in the infinitely imbalanced regime, however, the dataset used for experiments does not seem infinitely imbalanced. It is not clear how the experiment relates to the theoretical analysis.
2. The authors point out that there are some samples lying on the boundary of decision. however, it is remained to be dissgussed that what kind of samples are boundary samples. If N increases, and the proportion of boundary samples decreases, will it contradict the current conclusion?
3. The experiment includes some mainstream models such as ResNet or BERT, but the latest models are not discussed. It would be more reliable to arrange experiments on the sota models due to complex semantic models maybe out of the assumption of surrounded mean. (And the visualization of Figure 43 does not seem to show the surrounding mean)

---

> ### Author Rebuttal · Authors · 2026-03-31
>
> We thank the reviewer for the encouraging assessment and for highlighting these important points. We address the three main concerns below.
>
> ## Q1: Relation Between Infinitely Imbalanced Theory and Finite-$N$ Experiments
>
> >The authors prove asymptotic normality in the infinitely imbalanced regime, but the experiments do not seem infinitely imbalanced. How do the experiments relate to the theory?
>
> The infinitely imbalanced regime ($n$ fixed, $N \to \infty$) is where a precise mathematical characterization becomes tractable: we can prove asymptotic normality of the CAV and derive the $O(1/N)$ variance scaling. This is not an artifact of our setup, but mirrors standard TCAV practice, where concept examples are scarce and random references plenty. The canonical TensorFlow implementation recommends 50-200 random images with 10-500 runs; PyTorch Captum uses 120 random images per set; the ACE extension recommends 50-500 per folder [Kim et al., 2018; Ghorbani et al., 2019]. Our experiments should therefore be read as finite-$N$ instantiations of this asymptotic picture. For smaller $N$, finite-sample effects (e.g., regularization dominance) can cause deviations, but these do not alter the leading-order behavior. Empirically, once $N$ is moderately large, variability trends match the theory across all modalities tested.
>
> ## Q2: What Are Borderline Samples, and Does Their Proportion Decrease with $N$?
>
> >It remains to be discussed what kind of samples are borderline. If $N$ increases and the proportion of borderline samples decreases, does this contradict the conclusion?
>
> A sample is borderline when its concept sensitivity $S(x, \beta_\infty) = \nabla g_{\ell,k}(h_\ell(x)) \cdot \beta_\infty$ is near zero, i.e. geometrically, its class-gradient is nearly orthogonal to the limiting CAV direction. These are the points whose sign in Eq. (2) flips under small CAV perturbations. As $N$ grows, the fluctuation scale shrinks as $O(1/\sqrt{N})$ (Corollary 2), so the effective borderline zone narrows: only points with $|S(x, \beta_\infty)| < c/\sqrt{N}$ remain unstable. This does not contradict our conclusion. Our claim is not that TCAV variance is constant for all $N$, but that in the practically relevant regime, borderline points dominate the variance and create the observed plateau. On a finite test set with smallest nonzero sensitivity $c_{\min}$, decay resumes once $1/\sqrt{N} < c_{\min}$. As in high-dimensional layers ($d = 512$ for ResNet, $768$ for CLIP), inner products between weakly coupled vectors concentrate near zero with $O(1/\sqrt{d})$ spread [Vershynin, 2018, Thm. 3.4.6]. Therefore, $c_{\min}$ can be practically very small for high dimensional models. For example, with $c_{\min} \sim 10^{-3}$, decay would require $N > 10^{6}$, far beyond any practical TCAV budget. This is also consistent with the observation that the stable behaviour is much more pronounced for high-dimensional models like ViT or CliP and not really present for the simple model implemented for the tabular dataset.
>
> ## Q3: Modern Models and the "Surrounded Mean" Assumption
>
> >The experiments cover ResNet and BERT but not SotA models. Also, the visualization of Figure 43 does not seem to show the surrounded mean.
>
> We ran additional experiments on LLaVA and SmolLM2-1.7B-Instruct. The same pattern holds: CAV variability decreases with _N_, while TCAV stability is governed by borderline points. Results are available at https://anonymous.4open.science/r/icml_var_of_cavs-B6EE/xBVR/additional_models.pdf. Figure $43$ was meant as a geometric illustration, not a visual proof; the assumption is verified numerically. Concretely, "surrounded mean" requires that for any direction in activation space, at least a fraction $\delta$ of concept samples $\omega$ lie beyond distance $\epsilon$ from the mean $\bar{x}$ to ensure the activation cloud is not collapsed or one-sided. Our new  plots confirm this: for LLaVA it holds across all layers; for SmolLM2, for all layers except the first. See https://anonymous.4open.science/r/icml_var_of_cavs-B6EE/xBVR/additional_assumptions.pdf. We will add these results in the revision as empirical evidence of robustness across newer model families, while keeping the theoretical claim moderate.
>
> **References**
>
> - Ghorbani, A., Wexler, J., Zou, J. & Kim, B. (2019). Towards automatic concept-based explanations. NeurIPS, 32.
> - Kim, B. et al. (2018). Interpretability beyond feature attribution: TCAV. ICML, 80, 2668–2677.
> - Vershynin, R. (2018). High-Dimensional Probability. Cambridge University Press.

---

> > ### Author Rebuttal · Reviewer_xBVR · 2026-03-31
> >
> > Thank you for the clarification. I will retain my score.

---

> > > ### Author Response · Authors · 2026-04-04
> > >
> > > We are glad that all concerns were fully addressed. If there are any residual concerns we haven't addressed, we are happy to discuss them further in the hope that the reviewer will reconsider raising their score accordingly.

---

### Official Review · Reviewer_LjJY · 2026-03-11

**Soundness:** 3
**Presentation:** 4
**Significance:** 2
**Originality:** 2
**Overall Recommendation:** 4
**Confidence:** 3

**Summary:**

This paper investigates the variability of Concept Activation Vectors (CAVs) and TCAV scores, which arises from the random sampling of negative reference examples during their computation. The authors provide a theoretical analysis proving that the variance of the CAV direction decays at a rate of $\mathcal{O}(1/N)$ as the number of random samples $N$ increases. However, they discover that the variance of the downstream TCAV score remains approximately constant $\mathcal{O}(1)$ regardless of $N$, due to samples lying near the decision boundary. The paper offers practical guidelines derived from these theoretical and empirical insights: using the Difference of Means estimator with a maximized $N$ for stabilizing CAV directions, and averaging over multiple runs ($s$) with modest $N$ for stabilizing TCAV scores. The findings are validated across image, tabular, text, and multimodal datasets.

**Compliance With Llm Reviewing Policy:**

Affirmed.

**Final Justification:**

The authors provided a strong rebuttal that directly addressed all of my major concerns. They convincingly demonstrated the statistical significance of Figure 2 with new per-layer plots, corrected the textual claims regarding Figure 4 to match the visual evidence, and committed to moving key non-vision experiments to the main text. They also agreed to fix the noted presentation issues. Because the authors have thoroughly and effectively resolved the weaknesses I highlighted in my initial review, I have increased my score and recommend this paper for acceptance.

**Key Questions For Authors:**

1. Can you provide a more formal theoretical justification or a toy model proof for Conjecture 1 regarding the $\mathcal{O}(1)$ variance of the TCAV scores?
2. What are your hypotheses for the underlying causes of the initial increase in variance observed with the logistic regression classifiers for small values of $N$ before the asymptotic $\mathcal{O}(1/N)$ decay takes over?
3. In lines 159-160 (left), because the random dataset is sampled uniformly, there could be elements containing the concept present in both subsets, effectively being considered both positive and negative. Does this introduce label noise, and how does this affect the stability guarantees in your theoretical bounds?
4. In Equation 2, the magnitude of the sensitivity is not taken into account (e.g., a sensitivity of 0.1 is treated the same as 0.9). Given your findings that TCAV scores retain an $\mathcal{O}(1)$ variance due to borderline points, do you hypothesize that a magnitude-aware aggregation metric would achieve the $\mathcal{O}(1/N)$ stability seen in the CAV directions?
5. In line 237 (right), you use the phrase "for large N". Could you clarify this phrasing? As shown in the appendix, the asymptotic covariance matrix $\Sigma$ itself does not appear to depend on $N$, even though the Normal approximation relies on a large $N$.

**Limitations:**

The authors have adequately addressed the limitations of their work in Section 6. They explicitly state that their analysis focuses solely on statistical stability and does not determine which method yields the most accurate or faithful concept representations for downstream tasks.

**Strengths And Weaknesses:**

### Strengths
1. **Relevance and Practicality**: The stability of concept-based explainability methods is a critical issue. The recommendations are actionable and offer a computationally efficient way for practitioners to appropriately allocate compute budgets to achieve stable CAVs and TCAV scores.
2. **Comprehensive Empirical Evaluation**: The authors validate their theoretical claims extensively across four distinct data modalities (vision, tabular, text, multimodal) and several estimators.
3. **Insightful Findings**: The divergence in variance behavior between the continuous CAV direction ($\mathcal{O}(1/N)$) and the discrete TCAV score ($\mathcal{O}(1)$) is an interesting and non-trivial finding.

### Weaknesses
1. **Statistical Significance of Figure 2**: Currently, this plot lacks statistical significance because it only shows the results for a single layer. To address this, the authors could average the results across several layers and display the mean and standard deviation (e.g., using shaded areas).  Since the magnitude of variability might differ across layers, all variabilities could be normalized by dividing by the variance when $N=10$. In this updated plot, we should ideally see: (i) the mean following a trend similar to the current one, and (ii) a very low standard deviation, visually confirming that Corollary 1 is universally valid.
2. **Overstated Claims vs. Visual Evidence**: The text claims the TCAV variance is "constant" with $N$, but Figure 4 shows a clear initial decay before plateauing. The wording should be refined to reflect this initial decay.
3. **Over-reliance on the Appendix**: Section 5 is heavily reliant on the appendix. It would be worth the effort to include some key experiments for non-vision modalities in the main text.
4. **Presentation and Formatting Issues**:
    1. The beginning of Section 4.1 redundantly redefines terms ($n$ fixed points, $N$ random points) already defined in Section 3.1.
    2. In lines 190-191 (right), the text abruptly introduces the intercept ($\alpha \in \mathbb{R}$) and coefficients ($\beta \in \mathbb{R}^d$) without first clearly stating that a logistic regression is being trained. Furthermore, the parameters $\alpha \in \mathbb{R}, \beta \in \mathbb{R}^d$ are unnecessarily restated in line 200 (right).
    3. The acronym CARs is never explicitly defined as Concept Activation Regions in the text.
    4. In lines 313-315 (right), the text incorrectly references Section 4.2 for the $\mathcal{O}(1/s)$ scaling of multi-run TCAV scores, but Section 4.2 focuses on the $\mathcal{O}(1/N)$ scaling for the CAV direction.

---

> ### Author Rebuttal · Authors · 2026-03-31
>
> We thank the reviewers for their valuable feedback. We think that using their ideas we can significantly improve the paper. Let us now address some of the questions:
>
> ## Q1: Theoretical Justification of Conjecture 1
>
> >Can you provide a more formal theoretical justification or a toy model proof?
>
> A point $x$ is "borderline" when $S(x, \beta_\infty) = \nabla g_{\ell,k}(h_\ell(x)) \cdot \beta_\infty \approx 0$. By Corollary 2, fluctuations in $S(x, \beta_N)$ scale as $O(1/\sqrt{N})$, so sign flips only occur when $|S(x, \beta_\infty)| < c/\sqrt{N}$. This yields two regimes:
>
> **Case 1:** Exactly zero sensitivity. Points with $S(x, \beta_\infty) = 0$ flip with probability $\approx 1/2$ for all $N$, contributing an $N$-independent term to $\operatorname{Var}(\text{TCAV})$ and sustaining the $O(1)$ claim.
>
> **Case 2:** Near zero sensitivity. This effect also happens for $S(x, \beta_\infty) \approx 0$. If the distribution of sensitivities has positive density near zero, i.e., $f(0) > 0$,  a non-negligible fraction of points always sits inside the borderline zone $|S(x, β∞)| < c/\sqrt{N}$. Even though this zone shrinks with $N$, it traps roughly $|X_k| · f(0) · O(1/\sqrt{N})$ points at any given sample size, each of which can flip sign. Thus, variance decays only as $O(1/\sqrt{N})$.
>
> High-dimensional geometry reinforces this: in $\mathbb{R}^d$ inner products between weakly coupled vectors concentrate near zero with $O(1/\sqrt{d})$ spread [Vershynin, 2018, Thm. 3.4.6]. Therefore, $c_{\min}$ can be practically very small for high dimensional models. For example, with $c_{\min} \sim 10^{-3}$, decay would require $N > 10^{6}$, far beyond any practical budget. This is consistent with our experiments: the $O(1)$ behavior is pronounced for high-dimensional models like ResNet-50 or CLIP, and largely absent for the low-dimensional tabular model.
>
> ## Q2: Initial Variance Increase
>
> >What are your hypotheses for the initial increase in variance before the $O(1/N)$ decay takes over?
>
> At very small $N$, the $L_2$ penalty dominates, shrinking $\beta$ toward zero. As $N$ grows, the data term "pulls" the solution away from this regularization-dominated regime. However, many near-equivalent hyperplanes exist and which one is selected is highly sample-sensitive. From $N \sim 10$ to $300$, the problem transitions from underdetermined to a regime where the random class "fills out" the activation space, after which the $O(1/N)$ decay predicted by M-estimation theory takes over [Van der Vaart, 1998, Ch. 5].
>
> ## Q3: Label Noise
>
> >Concept samples could appear in both subsets. How does this affect stability guarantees?
>
> The random dataset may indeed contain concept-positive elements, introducing label noise. However, our theory is formulated for a general reference distribution $F_0$ and does **not** require a perfectly concept-free negative pool or perfect linear separability. Thus, as long as the stated assumptions still hold, the asymptotic stability result for the CAV remains unchanged. What contamination can change is the **limit CAV itself**.
>
> ## Q4: Magnitude-Aware Aggregation
>
> >Given that TCAV scores retain an $O(1)$ variance, could a magnitude-aware metric achieve the $O(1/N)$ stability seen in the CAV directions?
>
> This is an excellent point, and we think yes. For instance, the metric
>
> $$\mathrm{softTCAV}\tau = \frac{1}{|X_k|}\sum_{x \in X_k} \sigma\left(\frac{S_{C,k,\ell}(x)}{\tau}\right),$$
>
> is **$\approx 10-1000\times$ more stable than the original TCAV score** (see http://anonymous.4open.science/r/icml_var_of_cavs-B6EE/LjJY/soft_tcav.pdf). The canonical implementation recommends $50-200$ random images per run with $10-500$ random runs, which is very costly. Soft variants would make TCAV both faster and more stable. Designing variants that fully inherit the $O(1/N)$ stability of the CAVs while remaining semantically meaningful is exciting future work.
>
> ## Q5: Clarification of "For Large $N$"
>
> >The asymptotic covariance matrix $\Sigma$ does not appear to depend on $N$. Could you clarify the phrase "for large $N$"?
>
> We appreciate the close reading, though we believe the phrasing is consistent with standard usage: The phrase "for large $N$" refers not to $\Sigma$ itself but to the regime in which the distribution of the estimator is well approximated by the normal limit. This is a standard usage of the Central Limit Theorem. We illustrate this in https://anonymous.4open.science/r/icml_var_of_cavs-B6EE/LjJY/clt.pdf. Specifically, we prove that the estimator's variability decreases as $O(1/N)$ as $N \to \infty$, and our experiments confirm that the normal approximation is already accurate at $N \approx 50$.
>
> **References**
>
> - Kim et al., Interpretability beyond feature attribution: Quantitative testing with concept activation vectors (TCAV). ICML 2018.
> - Van der Vaart, Asymptotic Statistics, Cambridge University Press, 1998
> - Vershynin, High-Dimensional Probability, Cambridge University Press, 2018

---

> > ### Author Rebuttal · Reviewer_LjJY · 2026-04-03
> >
> > Dear Authors,
> >
> > Thank you for your rebuttal and for taking the time to answer my questions.
> >
> > However, your response did not address the concerns raised in the "Weaknesses" section of my review—specifically regarding the statistical significance of Figure 2, the overstated claims related to Figure 4, the over-reliance on the appendix, and the various presentation issues.
> >
> > Consequently, I am maintaining my score. I remain open to reconsidering my score if these specific weaknesses are correctly addressed.

---

> > > ### Author Response · Authors · 2026-04-04
> > >
> > > Thank you for your follow-up and the opportunity to clarify these remaining points.
> > >
> > > You are right that our initial rebuttal focused on the theoretical questions and did not address the weaknesses you raised. We apologize for the oversight. This was due to space constraints in the rebuttal, not a lack of consideration. Let us address each point now.
> > >
> > > ## Statistical Significance of Figure 2
> > > We agree this is a good suggestion. We have now produced per-layer variance decay plots for all layers of the models used in our experiments. These are available at https://anonymous.4open.science/r/icml_var_of_cavs-B6EE/LjJY/avg_over_layers.pdf.
> > >
> > > We display the averaged plots, each normalized by the variance at the smallest value of $N$.
> > > Across all layers, **the $O(1/N)$ decay predicted by Corollary $1$ is consistently observed**, confirming that the result is not an artifact of a particular layer choice. The original submission showed only representative layers because variance scales differ across layers, making a single-layer plot at its natural scale easier to interpret. These normalized per-layer plots offer a complementary view: they confirm that the trend holds universally across layers and thereby strengthen the statistical significance of the result. We will provide them in addition to the original figures in the revised paper.
> > >
> > > ## Overstated Claims Regarding Figure 4
> > > We agree that the current wording does not do justice to the initial transient decay visible in Figure $4$. We propose replacing the relevant passage with the following:
> > >
> > > >"While the TCAV score variance exhibits an initial decrease for small $N$, it quickly plateaus for moderate to large N. The initial decay likely reflects the transition out of the small-sample regime, where the regularization penalty dominates and many near-equivalent hyperplanes exist; as $N$ grows, the random class fills out the activation space and the classifier stabilizes (see Section 4.2). The subsequent plateau is driven by borderline points, i.e. samples whose concept sensitivity is near zero, whose sign in Eq. (2) flips under small CAV perturbations regardless of sample size. In high-dimensional layers, inner products concentrate near zero with $O(1/\sqrt{d})$ spread [Vershynin, 2018, Thm. 3.4.6], so borderline points can persist for any practical $N$, which is consistent with the plateau being especially pronounced for high-dimensional models like ViT or CLIP."
> > >
> > > ## Over-Reliance on the Appendix
> > > We acknowledge that Section 5 leans too heavily on supplementary material. If ICML proceeds as in previous years, accepted papers will be allowed one additional page in the camera-ready version. We will use this opportunity to **bring key non-vision experiments into the main text**, as you suggest. We believe this will make the paper substantially more self-contained.
> > >
> > > ## Presentation and Formatting Issues
> > > We did not detail these fixes in our initial rebuttal due to space constraints, but we want to assure the reviewer that we take these issues seriously and will address all of them in revision.
> > >
> > > We hope this clarifies our plans for revision and demonstrates that we are committed to addressing every concern raised.

---

### Official Review · Reviewer_KxdQ · 2026-03-12

**Soundness:** 3
**Presentation:** 3
**Significance:** 2
**Originality:** 3
**Overall Recommendation:** 4
**Confidence:** 3

**Summary:**

This paper studies the statistical variability of Concept Activation Vectors (CAVs) and the resulting stability of TCAV explanations under randomness in the reference set. The authors theoretically show that, in an infinitely imbalanced regime with fixed concept samples and increasing random references, logistic-regression-based CAV estimators are asymptotically normal with covariance scaling as O(1/N). They further analyze how this variance propagates to concept sensitivity and TCAV scores, showing that while sensitivity variance decreases with N, TCAV scores may remain unstable due to sign-boundary effects. Experiments across multiple modalities and CAV estimators support the theoretical findings and lead to practical recommendations for allocating compute when running TCAV.

**Compliance With Llm Reviewing Policy:**

Affirmed.

**Final Justification:**

My main concerns are now largely resolved, and I appreciate the clearer positioning of the theoretical claims; while the contribution remains somewhat specialized.

**Key Questions For Authors:**

1.	Assumptions underlying the theoretical analysis.
The asymptotic results rely on a specific regime where the number of random reference samples grows while the concept samples remain fixed, and also assume consistency of the logistic regression estimator. Could the authors clarify under what practical conditions these assumptions are expected to hold for typical TCAV usage? In particular, how sensitive are the theoretical conclusions to violations of these assumptions (e.g., when concept sets are small or noisy)?

2.	Generality beyond logistic regression CAV estimators.
The main theoretical results focus on logistic-regression-based CAV estimators. However, the paper experimentally considers several other estimators (e.g., hinge-loss linear models, Difference-of-Means). To what extent do the theoretical insights extend to these alternative estimators?

3.	Role of concept set size and quality.
The analysis primarily studies the effect of increasing the number of random reference samples N, while keeping the concept samples fixed. In practice, the size and quality of the concept set can also significantly influence TCAV results. Could the authors discuss or experimentally evaluate how varying the concept set size affects CAV variability and TCAV stability?

4.	Sensitivity to implementation choices.
TCAV results are often sensitive to design choices such as the layer from which representations are extracted, regularization strength in the CAV estimator, or feature normalization. Could the authors provide additional analysis or discussion on how these factors interact with the variance scaling behavior reported in the paper?

**Limitations:**

Yes.

**Strengths And Weaknesses:**

**Strengths:**
+ Provides a theoretical perspective on CAV variability.
The paper presents an asymptotic analysis showing that logistic-regression-based CAV estimators become asymptotically normal and their covariance scales as O(1/N) as the number of random reference samples increases. This offers a useful statistical interpretation of how reference set size affects CAV stability.

+ Analyzes variance propagation to downstream metrics.
Beyond CAV estimation, the paper studies how variance propagates to concept sensitivity scores and TCAV scores, highlighting that TCAV scores may remain unstable due to sign-boundary effects. This distinction between CAV stability and TCAV stability provides useful insight for practitioners.

+ Provides practical recommendations for TCAV usage.
The paper discusses how to allocate compute resources (e.g., increasing reference samples vs. increasing the number of runs) to improve stability. These recommendations may be useful for practitioners applying TCAV in real-world settings.


**Weaknesses:**
+ Theoretical analysis relies on restrictive assumptions.
The main theoretical results are derived under a specific asymptotic regime (infinitely imbalanced logistic regression with fixed concept examples and increasing random references). While this setting enables tractable analysis, it is not entirely clear how closely it reflects typical TCAV usage in practice, where concept sets and model representations may vary or be noisy.
+ Some theoretical claims are conditional or partially conjectural.
Certain key results depend on assumptions such as the consistency of the logistic regression estimator, which are assumed rather than fully established in the analysis. Additionally, the variance reduction result for multi-run averaging is presented as a conjecture rather than a formal proof, which weakens the theoretical completeness of the paper.
+ Experimental analysis focuses primarily on confirming scaling trends.
While the experiments span multiple modalities, they mainly validate the predicted variance scaling behavior. The empirical study could be strengthened by deeper investigation of factors that may influence stability in practice, such as different concept set sizes, regularization choices, feature normalization, or layer selection.
+ Practical recommendations may depend on implementation details.
Some of the conclusions (e.g., about the relative benefits of increasing N versus increasing the number of runs) may vary depending on the choice of CAV estimator, optimization procedure, or model representation. The current experiments provide useful evidence but may not fully establish the generality of these recommendations.
+ Limited advancement beyond existing TCAV methodology.
The paper improves understanding of variability in TCAV but does not fundamentally change the methodology or address broader challenges in concept-based interpretability, such as concept quality, disentanglement, or faithfulness. As a result, the overall impact may be somewhat specialized.

---

> ### Author Rebuttal · Authors · 2026-03-31
>
> We thank the reviewer for the careful reading and constructive feedback. We are glad the reviewer finds the theoretical perspective, the variance-propagation analysis, and the practical guidance valuable. Below we address the main questions.
>
> ## Q1: Assumptions underlying the theoretical analysis
>
> >Under what practical conditions do the assumptions hold, and how sensitive are the conclusions to small or noisy concept sets?
>
> The theorem is intentionally stated in the regime most relevant for TCAV: a fixed, curated concept set and an increasing number of random reference samples. This is both mathematically tractable and aligned with practice, where concept examples are scarce while random references are easy to scale. In the paper, the experiments already operate in the practical finite-$N$ regime ($N \in [10,300]$), and the predicted stabilization appears well before any extreme asymptotic limit, across images, tabular data, text, and multimodal models.
>
> Small or noisy concept sets do not remove the reference-sampling effect, but they can substantially worsen the constants, conditioning, and finite-sample behavior; in the extreme, they can yield a stable but semantically uninformative direction. This is a concept-quality issue rather than a failure of the variance analysis. We will make this distinction clearer in the revision.
>
> ## Q2: Generalization beyond logistic-regression CAV estimators
>
> >To what extent do the insights generalize beyond logistic-regression CAV estimators?
>
> While the main theorem in the core text is developed for logistic regression, we would like to emphasize that **the paper already goes  beyond that case.** In the appendix, we provide analogous theoretical analyses for hinge-loss linear models, Difference-of-Means, and Concept Activation Regions, and the experiments evaluate all four methods across modalities.
>
> Empirically, the same qualitative picture holds throughout: CAV variability decreases with sample size, concept sensitivities inherit that stabilization, and TCAV-style scores can remain less stable because they threshold at the sign boundary. All results can be found in the appendix.
>
> ## Q3: Role of concept-set size and concept quality
>
> >What is the role of concept set size and quality?
>
> This is an important point. The current theorem focuses on the practically dominant control variable in TCAV: the number of random references, as that is usually what practitioners can scale most easily. We agree, however, that concept-set size also matters. Although the theorem is stated for a fixed concept set and increasing number of random references, the same reasoning can be adapted to the complementary regime where the reference set is fixed and the number of concept examples grows.
>
> In additional experiments where we fixed the reference set and varied the number $n$ of concept examples, we observed lower variability for larger concept sets, with curves over the tested range consistent with an inverse-$n$ trend (see https://anonymous.4open.science/r/icml_var_of_cavs-B6EE/KxdQ/concept_set_size_variation.pdf). This is expected as our theoretical results directly apply also for the number of concept samples. On the other hand, poor concept quality primarily changes the limiting direction and the constant factors rather than the existence of stabilization itself.
>
> ## Q4: Sensitivity to implementation choices
>
> >How sensitive are the conclusions to implementation choices such as layer, regularization, normalization, or optimization details?
>
> In additional ablations, we varied layer, regularization strength, feature normalization, optimizer, and initialization. The main qualitative conclusions remain unchanged: CAV variability decreases with sample size, and TCAV stability benefits more from multi-run averaging than from simply increasing $N$. The main differences are in the constants.
>
> The clearest pattern is that later layers often exhibit lower variance, consistent with more clustered or separable representations. Stronger regularization generally lowers variance. Feature normalization also matters because it rescales activation dimensions before fitting the CAV and therefore changes which directions dominate the learned separator. By contrast, in our ablations optimizer choice and initialization have little impact in our experiments. We summarize the results in https://anonymous.4open.science/r/icml_var_of_cavs-B6EE/KxdQ/implementation_ablation.pdf and will add these ablations in the paper.
>
> We hope these clarifications address the reviewer’s concerns and strengthen the practical relevance of the paper.

---

> > ### Author Rebuttal · Reviewer_KxdQ · 2026-04-03
> >
> > Thank you for the detailed rebuttal. The response addresses several of my practical concerns. In particular, I appreciate the clarification of why the fixed-concept / increasing-reference regime is relevant for TCAV practice, the discussion of how the conclusions extend beyond logistic-regression CAVs, and the additional experiments on concept-set size and implementation choices (layer, regularization, normalization, optimizer, initialization). These additions strengthen the paper and help address my earlier questions about the practical stability of the reported trends.
> >
> > However, my concerns are only partially resolved. One remaining issue is the theoretical completeness of the analysis. My original concern was not only about the practical relevance of the asymptotic regime, but also about assumptions such as the consistency of the logistic-regression estimator and the partly conjectural status of the multi-run averaging result. The rebuttal clarifies the intended practical regime and provides additional empirical evidence, but it does not directly state the conditions under which consistency is expected to hold in this setting, nor does it clarify whether the multi-run averaging claim should be viewed as a theorem or an empirical/conjectural observation.
> >
> > Follow-up question: could the authors state more explicitly the assumptions under which consistency is expected to hold for the logistic-regression CAV estimator in their setting, and clarify the precise status of the multi-run averaging result (formal theorem vs. conjecture / empirical observation)?

---

> > > ### Author Response · Authors · 2026-04-04
> > >
> > > We thank the reviewer for the continued engagement and the opportunity to clarify these two remaining points.
> > >
> > > ## 1. Consistency of the logistic-regression CAV estimator.
> > > The ingredients are already present in the current draft (Assumption 1 in Section 3.1 and Lemmas 1-3 in Appendix B.1), but we agree that the consistency statement itself is not stated explicitly enough.
> > > We provide an **empirical verification** of consistency (see https://anonymous.4open.science/r/icml_var_of_cavs-B6EE/KxdQ/beta_star_convergence.pdf).
> > >
> > > Concretely, we approximate $\beta_\infty$ by a reference fit $\hat \beta_\infty$ obtained at the largest feasible sample size, and plot $||\hat\beta_N-\hat\beta_\infty ||$ as a function of $N$. In our experiments this distance decreases as $N$ grows, which is in agreement with the convergence of the slope estimator $\beta_N$. We will put this supplementary figure into the revised paper.
> > >
> > > Formally, the correct object is $\widehat Q_N(\beta):=\inf_{\alpha\in\mathbb R}\widehat L_N(\alpha,\beta)$.
> > > Because the intercept $\alpha_N$ diverges in the infinitely imbalanced regime [3], we do not apply a textbook M-estimation theorem directly to the full parameter vector $(\alpha_N,\beta_N)$. Instead, $\beta_N$ is the minimizer of $\widehat Q_N$, and consistency **follows under standard extremum-estimation conditions** [2, Thm. 2.1.]:
> > > - concept and reference examples are sampled i.i.d.;
> > > - the population criterion $Q$ has a unique well-separated minimizer $\beta_0$, which is the role of Assumption 1;
> > > - $\widehat Q_N(\beta)$  converges uniformly in probability to $Q_0(\beta)$: $\sup_{\|\beta\|\le R}\big|\widehat Q_N(\beta)-Q(\beta)\big|\xrightarrow{p}0$;
> > > - the estimator space (where $\beta_N$ lives) is compact, which is supplied here by the $L^2$-regularization bound in Lemma 2.
> > >
> > > With these conditions, Thm. 2.1 of [2] yields $\beta_N \xrightarrow{p} \beta_0$. We will therefore add a proposition about consistency and proof immediately after Assumption 4 in Section 3.2, rather than leaving this only implicit in Appendix B.1.
> > >
> > > ## 2. Status of the multi-run averaging result.
> > > This is not a theorem, and we will revise the wording to make that completely explicit. Conjecture 1 should be read as a conjecture and empirical observation. The exact identity
> > >  $\mathrm{Var}(\overline T_s)=\mathrm{Var}(T_1)/s$ for independent runs is immediate. The nontrivial part is whether $\mathrm{Var}(T_1)$ remains $O(1)$ over the practically relevant range of $N$, so that averaging across $s$ runs produces a meaningful $O(1/s)$ reduction.
> > >
> > > Our proposed hypothesis is that this happens when the evaluation set contains a nontrivial mass of borderline or near-borderline points, i.e., points with $S(x,\beta_0)\approx 0$. If $|S(x,\beta_0)|$ is bounded away from zero, then the sign $\mathbf{1}\{S(x,\beta_N)>0\}$ stabilizes and such points stop contributing to the variance. We do not currently prove that every model/evaluation set must contain enough such points, so we do not present Conjecture 1 as a general theorem.
> > >
> > > **What we do claim is empirical:** in the settings we study, the observed averaging behavior is consistent with this mechanism. To support that point more directly, we provide additional experiments in the anonymous repository and will put them in the supplementary material. These experiments are consistent with the intuition that near-borderline evaluation points are what sustain the single-run variance and make averaging effective in practice. See https://anonymous.4open.science/r/icml_var_of_cavs-B6EE/KxdQ/sensitivity_dist.pdf.
> > >
> > > So the answer to the reviewer’s question is:
> > > - **Consistency of the logistic-regression CAV parameter $\beta_N$** is expected under standard extremum-estimation assumptions, and we will state these assumptions and the corresponding proof in the revised paper along with empirical evidence, and
> > > - **Conjecture 1 is not a theorem**; it is a conjecture supported by the experiments, and we will revise the wording to make that status clear.
> > >
> > >
> > > ---
> > >
> > > ### References:
> > > - [1] van der Vaart, A. W. (1998). Asymptotic Statistics. Cambridge University Press.
> > > - [2] Newey, W. K. and McFadden, D. (1994). Large sample estimation and hypothesis testing. In Handbook of Econometrics, Vol. 4, Ch. 36. Elsevier.
> > > - [3] Owen, A. B. (2007). Infinitely imbalanced logistic regression. Journal of Machine Learning Research, 8(4).

---

### Official Review · Reviewer_4xWC · 2026-03-13

**Soundness:** 3
**Presentation:** 3
**Significance:** 2
**Originality:** 2
**Overall Recommendation:** 4
**Confidence:** 4

**Summary:**

The paper analyzes the stability of the TCAV method. Since Concept Activation Vectors (CAVs) are computed using randomly sampled reference examples, their estimates can vary across runs, potentially leading to inconsistent explanations. Therefore, the authors present a theoretical analysis of the variability of CAVs under an infinitely imbalanced logistic regression setting. They show that the variance of CAV estimates decreases as the number of random samples increases. The paper also provides practical recommendations on how to allocate random samples and runs to improve the stability of TCAV explanations.

**Compliance With Llm Reviewing Policy:**

Affirmed.

**Final Justification:**

The rebuttal addressed my concerns, and my initial score already reflected my view of the paper. Therefore, I will keep my score unchanged.

**Key Questions For Authors:**

1. The paper focuses on improving the stability of TCAV explanations. However, it remains unclear how this improved stability affects the quality of the resulting explanations. If the proposed approach generalizes across modalities, does this also imply that the explanations are more useful or meaningful for domain experts?
2. Related to Weakness 2, the theoretical analysis focuses on an infinitely imbalanced logistic regression setting. How sensitive are the results to this assumption in more practical settings where the class imbalance may be less extreme?
3. The experiments suggest similar stability patterns across architectures within each modality. Does Assumption 1 hold uniformly across architectures such as CNNs and ViTs, or does the empirical similarity in scaling behavior mask architecture-specific differences in the constant factor of the O(1/N) term? More broadly, do the theoretical guarantees and empirical findings generalize to modern large-scale models with substantially different representation geometries, such as LLaVA or decoder-only LLMs where CAVs are increasingly being applied?

**Limitations:**

Yes

**Strengths And Weaknesses:**

Strengths
1. The paper provides a principled theoretical analysis of the variability of CAVs, which has not been thoroughly studied in prior work, even though TCAV is widely used in the XAI community.
2. The experiments are conducted across four different modalities, which helps demonstrate the generalizability of the findings.
3. The paper provides practical guidelines that could be useful for practitioners applying TCAV in real-world settings.
4. The paper is well written and easy to follow.

Weaknesses
1. The paper does not evaluate whether the proposed recommendations for stabilizing CAVs improve the usefulness of TCAV explanations in downstream tasks. In particular, it remains unclear whether increased stability leads to more useful or reliable explanations, for example by aligning better with domain expert judgments.
2. The theoretical results rely on assumptions such as infinitely imbalanced logistic regression, which may not fully reflect practical settings where TCAV is applied.
3. The analysis mainly focuses on variability introduced by random reference sampling, but other potential sources of variability, such as model initialization or optimization randomness, are not explored.

---

> ### Author Rebuttal · Authors · 2026-03-31
>
> We thank the reviewer for the thoughtful and constructive feedback. We are encouraged that the reviewer finds the paper technically sound, clearly written, and practically relevant. Below we address the main questions.
>
> ## Q1: Stability vs. explanation usefulness
>
> >It remains unclear how this improved stability affects the quality of the resulting explanations. Does this imply that the explanations are more meaningful for domain experts?
>
> Not necessarily, and we do not claim that stability alone implies explanation quality. Our claim is narrower: stability is a prerequisite for meaningful interpretation. If the same concept/model/layer can yield materially different TCAV scores across runs, then the explanation is not reliable enough to interpret, compare, or validate. In that sense, our contribution is about the measurement reliability of TCAV rather than a complete downstream evaluation of explanation utility. This scope is consistent with the paper’s stated contributions and limitations.
>
> That said, improved stability is still practically important. The paper shows that CAV variance decreases as $O(1/N)$, and concept sensitivity inherits the same rate, while the hard-thresholded TCAV score can remain noisy because of borderline points. These results identify where instability comes from and provide concrete levers for reducing it, which we view as a necessary first step toward more useful explanations.
>
> ## Q2: Practical relevance of the infinitely imbalanced regime
>
> >The theoretical results rely on assumptions such as infinitely imbalanced logistic regression, which may not fully reflect practical settings where TCAV is applied. How sensitive are the results to this assumption in more practical settings where the class imbalance may be less extreme?
>
> We agree this deserves clarification. The infinitely imbalanced regime is not only a mathematical convenience; it matches a common TCAV setting in which concept examples are scarce while random reference examples are easy to sample. The paper explicitly motivates this by contrasting small annotated concept sets with large pools of random reference data, and the experiments already vary $N$ from $10$ to $300$ across image, tabular, text, and multimodal settings.
>
> Empirically, the predicted $O(1/N)$ trend is visible well before any extreme asymptotic regime. Across all four modalities, the scaling appears already for modest $N$. So our intended claim is not that only the limit $N \to \infty$, $n$ fixed matters, but that this regime captures the dominant source of variability in practice when $N$ is the main controllable quantity.
>
> ## Q3: Architecture dependence and modern large-scale models
>
> >The experiments suggest similar stability patterns across architectures within each modality. Does Assumption 1 hold uniformly across architectures such as CNNs and ViTs, or does the empirical similarity in scaling behavior mask architecture-specific differences in the constant factor of the $O(1/N)$ term? More broadly, do the theoretical guarantees and empirical findings generalize to modern large-scale models such as LLaVA or decoder-only LLMs?
>
> In the submitted paper, we already report that in the image setting the same qualitative scaling appears across ResNet-50, GoogLeNet, MobileNetV3, and ViT-B/16. Hereby, some architectures converge more slowly than others. Similarly, the experiments on the other three modalities show the same trend.
>
> Our interpretation is therefore: the rate is robust, but the constant in the $O(1/N)$ term can depend on the architecture, layer, and representation geometry.
>
> Following the reviewer’s suggestion, we also **ran additional experiments on a modern vision-language model (LLaVA) and a recent decoder-only language model (HuggingFaceTB/SmolLM2-1.7B-Instruct).** Regarding Assumption $1$: for LLaVA it holds across all layers; for SmolLM2, for all 24 layers except the first. See https://anonymous.4open.science/r/icml_var_of_cavs-B6EE/4xWC/additional_assumptions.pdf. Regardless, these show the same qualitative behavior: CAV variability decreases with the number of random samples, and the scaling is consistent with our other modalities; see https://anonymous.4open.science/r/icml_var_of_cavs-B6EE/4xWC/additional_models.pdf. We will add these results in the revision as empirical evidence of robustness across newer model families, while keeping the theoretical claim appropriately modest.
>
> **Regarding other sources of variability:** in additional ablations (see https://anonymous.4open.science/r/icml_var_of_cavs-B6EE/4xWC/implementation_ablation.pdf ), we varied layer, regularization strength, feature normalization, optimizer, and initialization. The $O(1/N)$ rate is robust across all settings. The main effect is on the constants: higher layers and stronger regularization lower variance, while optimizer choice and initialization have little impact.

---

> > ### Author Rebuttal · Reviewer_4xWC · 2026-04-02
> >
> > Thank you for the detailed response. The authors’ clarifications improved my understanding of the paper, particularly regarding the scope of the contribution. I also appreciate additional empirical evidence across architectures and modalities, including the new results on more recent models. While I agree that stability is an important prerequisite for interpretability, it still remains unclear to what extent improved stability translates into more useful or actionable explanations in downstream settings. Therefore, I will maintain my weak accept recommendation.

---

> > > ### Author Response · Authors · 2026-04-04
> > >
> > > We thank the reviewer for acknowledging the contribution of our work. We are glad that our answer clarified the scope of the contribution and that the additional empirical evidence across architectures and modalities was found convincing.
> > >
> > > To further expand our earlier response on the link between stability and actionable explainability, we would like to offer a concrete empirical illustration that we believe sharpens the point. We respectfully argue that stability is not merely a desirable property of explanations, but a necessary condition for their validity. Under the original protocol with a small number of random concept sets, the TCAV score can straddle the $0.5$ decision threshold (see https://anonymous.4open.science/r/icml_var_of_cavs-B6EE/4xWC/tabular_ci.pdf). In this regime, the statistical test is inconclusive: one cannot determine whether the concept is positively or negatively associated with the class. The explanation **is not just "less stable," it is unusable**.
> > >
> > > Increasing $n$ narrows the confidence interval, eventually placing it entirely above the threshold. The very same concept that was previously uninterpretable now yields a valid, actionable TCAV result. This demonstrates that instability can render an explanation method unable to produce any conclusion at all.
> > >
> > > We hope this concrete example illustrates that without sufficient stability, explanation methods may fail to produce any meaningful conclusion. We are happy to address any further questions the reviewer may have.

---

### Decision · Program_Chairs · 2026-04-30

**Decision:**

Accept (regular)

**Comment:**

Summary: This paper studies the variability of Concept Activation Vectors (CAVs), a classical method for concept-based interpretability (e.g., TCAV). The authors analyze how randomness in sampling (e.g., reference examples) affects the stability of learned CAV directions. The paper provides: (1) a theoretical analysis showing that, under an asymptotic regime (infinitely imbalanced logistic regression), (2) the variance of CAV estimates decreases as the number of random samples increases (approximately O(1/N)), (3) empirical validation across multiple modalities (image, text, tabular), demonstrating similar scaling behavior, (4) practical recommendations for stabilizing CAVs, such as increasing reference samples and averaging across runs. The main conclusion is that CAV instability is largely due to sampling noise and can be mitigated with sufficient averaging and data.

Strengths:
(1) Problem formulation. Reviewers agree that stability of interpretability methods is an important and practical issue.
(2) Theoretical contribution. The paper provides a principled analysis of CAV variability.
(3) Empirical validation across domains. Experiments span multiple modalities (image, text, tabular), different datasets and architectures.

Weakness:
(1) Limited novelty beyond variance analysis. The contribution is primarily focused on variance characterization. The paper does not propose fundamentally new interpretability methods or metrics.
(2) The paper only considers TCAV as the target of study, which limits the scope of this work.
(3) Several reviewers note missing analysis of model-related randomness (e.g., initialization, optimization), effects of concept set quality or size, sensitivity to implementation choices (e.g., normalization, regularization).

Overall, the paper provides a useful theoretical and empirical study of CAV variability. Its main strength lies in clarifying how sampling affects stability and offering actionable recommendations. The contribution is solid but somewhat narrow in scope. Reviewers view the paper as a technically sound but incremental contribution, with generally positive but moderate recommendations (e.g., weak accept).